# Simple Scaling of extreme precipitation in North America

Silvia Innocenti[1], Alain Mailhot[1], and Anne Frigon[2]

[1]Centre Eau-Terre-Environnement, INRS, 490 de la Couronne, Québec, Canada, G1K 9A9
[2]Consortium Ouranos, 550 Sherbrooke Ouest, Montrèal, Canada, H3A 1B9

*Correspondence to:* S. Innocenti (silvia.innocenti@ete.inrs.ca)

**Abstract.** Extreme precipitation is highly variable in space and time. It is therefore important to characterize precipitation intensity distributions at several temporal and spatial scales. This is a key issue in infrastructure design and risk analysis, for which Intensity-Duration-Frequency (IDF) curves are the standard tools used for describing the relationships among extreme rainfall intensities, their frequencies, and their durations. Simple Scaling (SS) models, characterizing the relationships among extreme probability distributions at several durations, represent a powerful means for improving IDF estimates. This study tested SS models for approximately 2700 stations in North America. Annual Maxima Series (AMS) over various duration intervals from 15 min to 7 days were considered. The range of validity, magnitude, and spatial variability of the estimated scaling exponents were investigated. Results provide additional guidance for the influence of both local geographical characteristics, such as topography, and regional climatic features on precipitation scaling. Generalized Extreme Value (GEV) distributions based on SS models were also examined. Results demonstrate an improvement of GEV parameter estimates, especially for the shape parameter, when data from different durations were pooled under the SS hypothesis.

## 1 Introduction

Extreme precipitation is highly variable in space and time as various physical processes are involved in its generation. Characterizing this spatial and temporal variability is crucial for infrastructure design and to evaluate and predict the impacts of natural hazards on ecosystems and communities. Available precipitation records are however sparse and cover short time periods, making a complete and adequate statistical characterization of extreme precipitation difficult. The resolution of available data, whether observed at meteorological stations or simulated by weather and climate models, often mismatches the resolution needed for applications (e.g., Blöschl and Sivapalan, 1995; Maraun et al., 2010; Willems et al., 2012), thus adding to the difficulty of achieving complete and adequate statistical characterizations of extreme precipitation.

The need for multi-scale analysis of precipitation has been widely recognized in the past (Rodriguez-Iturbe et al., 1984; Blöschl and Sivapalan, 1995; Hartmann et al., 2013; Westra et al., 2014, among others) and much effort has been put into the development of relationships among extreme precipitation characteristics at different scales. The conventional approach for characterizing scale transitions in time involves the construction of Intensity-Duration-Frequency (IDF) or the equivalent Depth-Duration-Frequency (DDF) curves (Bernard, 1932; Burlando and Rosso, 1996; Sivapalan and Blöschl, 1998; Koutsoyiannis et al., 1998; Asquith and Famiglietti, 2000; Overeem et al., 2008; Veneziano and Yoon, 2013). These curves are a standard tool for hydraulic design and risk analysis as they describe the relationships between the frequency of occurrence of

extreme rainfall intensities (depth) $X_d$ and various durations $d$ (e.g., CSA, 2012). Analysis is usually conducted by separately estimating the statistical distributions of $X_d$ at the different durations (see Koutsoyiannis et al., 1998; Papalexiou et al., 2013, for discussions about commonly used probability distributions). The parameters or the quantiles of these theoretical distributions are then empirically compared to describe the variations of extreme rainfall properties across temporal scales.

Despite its simplicity, this procedure presents several drawbacks. In particular, it does not guarantee the statistical consistency of precipitation distributions, independently estimated at the different durations, and it limits IDF extrapolation at non-observed scales or ungauged sites. Uncertainties of estimated quantiles are also presumably larger because precipitation distribution and IDF curve parameters are fitted separately.

Scaling models (Lovejoy and Mandelbrot, 1985; Gupta and Waymire, 1990; Veneziano et al., 2007) based on the concept of
scale invariance (Dubrulle et al., 1997), have been proposed to link rainfall features at different temporal and spatial scales. Scale invariance states that the statistical characteristics (e.g., moments or quantiles) of precipitation intensity observed at two different scales $d$ and $\lambda d$ can be related to each other by a power law of the form:

$$f(X_{\lambda d}) = \lambda^{-H} f(X_d) \tag{1}$$

where $f(.)$ is a function of $X$ with invariant shape when rescaling the variable $X$ by a multiplicative factor $\lambda$ and for some values of the exponent $H \in \mathbb{R}$. In the simplest case, a constant multiplicative factor adequately describes the scale change. The
corresponding mathematical models are known as *Simple Scaling* (SS) models (Gupta and Waymire, 1990). SS models are attractive because of the small number of parameters involved, as opposed to *Multiscaling* (MS) models which involve more than one multiplicative factor in Eq. (1) (e.g., Lovejoy and Schertzer, 1985; Gupta and Waymire, 1990; Burlando and Rosso, 1996; Veneziano and Furcolo, 2002; Veneziano and Langousis, 2010; Langousis et al., 2013). A single *scaling exponent $H$* is used to characterize the extreme rainfall distribution at all scales over which the scale invariance property holds. As a consequence,
a consistent and efficient estimation of extreme precipitation characteristics is possible, even at non-sampled temporal scales, and a parsimonious formulation of IDF curves based on analytical results is available (e.g., Menabde et al., 1999; Burlando and Rosso, 1996; De Michele et al., 2001; Ceresetti, 2011).

Theoretical and physical evidence of the scaling properties of precipitation intensity over a wide range of durations has been provided by several studies. MS has been demonstrated to be appropriate for modeling the temporal scaling features of the
precipitation process (i.e., not only the extreme distribution) and for the extremes in event-based representations of rainfall (stochastic rainfall modeling) (e.g., Veneziano and Furcolo, 2002; Veneziano and Iacobellis, 2002; Langousis et al., 2013, and references therein). These multifractal features of precipitation last within a finite range of temporal scales (approximatively between 1 hour and 1 week) and concern the temporal dependence structure of the process. They have been connected to the large fluctuations of the atmospheric and climate system governing precipitation which are likely to produce a "cascade of
random multiplicative effects" (Gupta and Waymire, 1990).

At the same time, many studies confirmed the validity of SS for approximating the precipitation distribution tails in IDF estimation (for examples of durations ranging from 5 min to 24 h see Menabde et al., 1999; Veneziano and Furcolo, 2002; Yu et al., 2004; Nhat et al., 2007; Bara et al., 2009; Ceresetti et al., 2010; Panthou et al., 2014). This type of scaling is substan-

tially different from the temporal scaling since it only refers to the power law shape of the marginal distribution of extreme rainfall. Application of the SS models to precipitation records showed that the scaling exponent estimates may depend on the considered range of durations (e.g., Borga et al., 2005; Nhat et al., 2007) and the climatological and geographical features of the study regions (e.g., Menabde et al., 1999; Bara et al., 2009; Borga et al., 2005; Ceresetti et al., 2010; Blanchet et al., 2016).

However, the application of the SS framework has been mainly restricted to specific regions and small observational datasets. A deeper analysis of the effects of geoclimatic factors on the SS approximation validity and on estimated scaling exponent is thus needed.

The present study aims to deepen the knowledge of the scale-invariant properties of extreme rainfall intensity by analyzing SS model estimates across North America using a large number of station series. The specific objectives of this study are: a) assess

the ability of SS models to reproduce extreme precipitation distribution; b) explore the variability of scaling exponent estimates over a broad set of temporal durations and identify possible effects of the dominant climate and pluviometric regimes on SS; c) evaluate the possible advantages of the introduction of the SS hypothesis in parametric models of extreme precipitation. Note that, although modifications in precipitation distributions are expected as a result of climate changes (e.g., Trenberth et al., 2003; Hartmann et al., 2013; Westra et al., 2014), the proposed approach implicitly relies on the assumption of station-

arity for extreme rainfall. This choice has been motivated by both the limited evidence for changes in rainfall intensities for North America extremes during last decades, and the difficulties of assessing distribution changes from short recorded series, especially for sub-daily extremes (Barbero et al., 2017, and references therein).

The article is structured as follows. In Sect. 2 the statistical basis of scaling models is presented, while data and their preliminary treatments are described in Sect. 3. Sect 4 presents the distribution-free estimation of SS models and their validation using

available series. Section 5 focuses on to the spatial variability of SS exponents and discusses the scaling exponent variation from a regional perspective. Finally, the SS estimation based on the Generalized Extreme Value (GEV) assumption is discussed in Sect. 6, followed by a discussion and conclusions [Sect. 7]. Table S1 of the supplementary material lists in alphabetic order the recurrent acronyms used in text.

## 2    Simple Scaling models for precipitation intensity

When the equality in Eq. (1) holds for the cumulative distribution function (cdf) of the precipitation intensity $X$ sampled at two different durations $d$ and $\lambda d$, the Simple Scaling (SS) can be expressed as (Gupta and Waymire, 1990; Menabde et al., 1999):

$$X_d \stackrel{dist}{=} \lambda^H X_{\lambda d}, \tag{2}$$

where $H \in \mathbb{R}$ and $\stackrel{dist}{=}$ means that the same probability distribution applies for $X_d$ and $X_{\lambda d}$, up to a dilatation or contraction of size $\lambda^H$. An important consequence of the SS assumption is that $X_d$ and $\lambda^H X_{\lambda d}$ have the same distribution. Hence, if $X_d$ and

$X_{\lambda d}$ have finite moments of order $q$, $E[X_d^q]$ and $E[X_{\lambda d}^q]$, these moments are thus linked by the following relationship (Gupta

and Waymire, 1990; Menabde et al., 1999):

$$E[X_d^q] = \lambda^{Hq} E[X_{\lambda d}^q]. \tag{3}$$

This last relationship is usually referred to as the *wide sense* simple scaling property (Gupta and Waymire, 1990) and signifies that simple scaling results in a simple translation of the log-moments between scales:

$$\ln\{E[X_d^q]\} = \ln\{E[X_{\lambda d}^q]\} + Hq\ln\lambda \tag{4}$$

Moreover, without loss of generality, $\lambda$ can always be expressed as the scale ratio $\lambda = d/d^*$ defined for a reference dura-
tion $d^*$ chosen, for simplicity, as $d^* = 1$. Therefore, the SS model can be estimated and validated over a set of durations $d_1 < d_2 < .. < d_D$ by simply checking the linearity in a log-log plot of the $X$ moments versus the observed durations $d_j$, $j = 1, 2, \ldots, D$ [see, for instance, Gupta and Waymire (1990); Burlando and Rosso (1996); Fig. 1 of Nhat et al. (2007); and Fig. 2 (a) of Panthou et al. (2014)]. If $H$ estimated for the first moment equals the exponents (slopes) for the other moments, the precipitation intensity $X$ can be considered scale invariant under SS in the interval of durations $d_1$ to $d_D$.

More sophisticated methods have also been proposed for detecting and estimating scale invariance [for instance, dimensional analysis, Lovejoy and Schertzer (1985); Tessier et al. (1993); Bendjoudi et al. (1997); Dubrulle et al. (1997); spectral analysis and wavelet estimation Olsson et al. (1999); Venugopal et al. (2006) Ceresetti (2011); and empirical probability distribution function (pdf) power law detection Hubert and Bendjoudi (1996); Sivakumar (2000); Ceresetti et al. (2010)]. However, estimation through the moment scaling analysis is by far the simplest and most intuitive tool to check the SS hypothesis for a large dataset. For this reason, the presented analyses are based on this method.

According to the literature, the values of the scaling exponents $H$ generally range between 0.4 and 0.8 for precipitation intensity considered at daily and shorter time scales (e.g., Burlando and Rosso, 1996; Menabde et al., 1999; Veneziano and Furcolo, 2002; Bara et al., 2009) (note that for the rainfall depth the scaling exponent $H_{depth} = 1 - H$ applies). Values from 0.3 to 0.9 have also been reported for some specific cases (e.g., Yu et al., 2004; Panthou et al., 2014, for scaling intervals defined within 1 h and 24 h).

Higher $H$ values have been generally observed for shorter-duration intervals, and regions dominated by convective precipitation (e.g., Borga et al., 2005; Nhat et al., 2007; Ceresetti et al., 2010; Panthou et al., 2014, and references therein). Nonetheless, some studies performing spatio-temporal scaling analysis reached a different conclusion. For instance, Eggert et al. (2015), analyzing extreme precipitation events from radar data for durations between 5 min and 6 h and spatial scales between 1 km and 50 km, indirectly showed that stratiform precipitation intensity generally displays higher temporal scaling exponents than convective intensity. For short-duration intervals (typically less than one hour), previous studies have also reported more spatially homogeneous $H$ estimates than for long-duration intervals (e.g., Alila, 2000; Borga et al., 2005, and references therein). This suggests that processes involved in the generation of local precipitation are comparable across different regions.

More generally, higher $H$ values are associated with larger variations in moment values as the scale is changed (i.e. a stronger scaling), while $H$ close to zero means that the $X_d$ distributions for different durations $d$ more closely match each other.

## 2.1 Simple Scaling GEV models

Annual Maximum Series (AMS) are widely used to select rainfall extremes from available precipitation series. Various theoretical arguments and experimental evidences support their use for extreme precipitation inference (e.g., Coles et al., 1999; Katz et al., 2002; Koutsoyiannis, 2004a; Papalexiou et al., 2013).

Based on the asymptotic results of the Extreme Value Theory (Coles, 2001), the AMS distribution of a random variable $X$ is well described by the Generalized Extreme Value (GEV) distribution family. If we represent the AMS by $(x_1, x_2, ..., x_n)$, the GEV cdf can be written as (Coles, 2001):

$$F(x) = \exp\left\{-\left[1 + \xi\left(\frac{x-\mu}{\sigma}\right)\right]^{-1/\xi}\right\} \tag{5}$$

where $\xi \neq 0$, $-\infty < x \leq \mu + \sigma/\xi$ if $\xi < 0$ (bounded tail), and $1/\mu + \sigma\xi \leq x < +\infty$ if $\xi > 0$ (heavy tail). If $\xi = 0$ (light-tailed
shape, Gumbel distribution), Eq. (5) reduces to:

$$F(x) = \exp\left\{-\exp-\left\{\frac{x-\mu}{\sigma}\right\}\right\} \tag{6}$$

where $-\infty < x < +\infty$. In Eq. (5) and (6), the parameters $\mu \in \mathbb{R}$, $\sigma > 0$ and $\xi$ respectively represent the location, scale, and shape parameters of the distribution. The shape parameter describes the characteristics of the distribution tails. Thus, high order quantile estimation is particularly affected by the value of $\xi$.

In applications, the GEV distribution is frequently constrained by the assumption that $\xi = 0$ (i.e., to the Gumbel distribution), due to the difficulty of estimating significant values of the shape parameter when the recorded series are short (e.g., Borga et al., 2005; Overeem et al., 2008; CSA, 2012). However, based on theoretical and empirical evidence, many authors have shown that this assumption is too restrictive for extreme precipitation, and may lead to important underestimations of the extreme quantiles (e.g., Koutsoyiannis, 2004a, b; Overeem et al., 2008; Papalexiou et al., 2013; Papalexiou and Koutsoyiannis,
2013). Instead, approaches aimed at increasing the sample size may be used to improve the estimation of the GEV distribution shape parameter (for instance, the Regional Frequency Analysis (RFA), Hosking and Wallis, 1997). Among these approaches, SS models constitute an appealing way to pool data from different samples (durations) and reduce uncertainties in GEV parameters.

For the GEV distribution it is straightforward to verify that, if $X \stackrel{dist}{=} GEV(\mu, \sigma, \xi)$ then $\lambda X \stackrel{dist}{=} GEV(\lambda\mu, \lambda\sigma, \xi)$ for any
$\lambda \in \mathbb{R}$. This means that the GEV family described by Eq. (5) and (6) satisfies Eq. (1) and thus complies with statistical scale invariance for any constant multiplicative transformation of $X$. Hence, when the scale invariance is further assumed for the change of observational scale from duration $d$ to $\lambda d$ [as in Eq. 2], the wide sense SS definition [Eq. (3)] gives:

$$\mu_d = d^H \mu_* \ , \sigma_d = d^H \sigma_* \ , \ \text{and} \ \xi_d = \xi_* \tag{7}$$

where $\mu_*$, $\sigma_*$, and $\xi_*$ represent the GEV parameters for a reference duration $d^*$ chosen, for simplicity, as $d^* = 1$, so that $\lambda = d$.

## 2.2 SS GEV estimation

Taking advantage of the scale invariant formulation of the GEV distribution, many authors have proposed simple scaling IDF and DDF models for extreme precipitation series (e.g., Yu et al., 2004; Borga et al., 2005; Bougadis and Adamowski, 2006; Bara et al., 2009; Ceresetti, 2011). In these cases, the scaling exponent and the GEV parameters are generally estimated in two separate steps: first, the $H$ value is empirically determined through a log-log linear regression, as described above; then, GEV parameters $\mu_*$, $\sigma_*$, and $\xi_*$ for the reference duration $d^*$ are estimated on the pooled sample of all available durations. In this case, classical estimation procedures, such as GEV Maximum-Likelihood (ML) (Coles, 2001) or Probability Weighted Moment (PWM) (Greenwood et al., 1979; Hosking et al., 1985), can be used.

In a few other cases, a Generalized Additive Model ML (GAM-ML) framework (Coles, 2001; Katz, 2013) has also been used to obtain the joint estimate of $H, \mu_*, \sigma_*$, and $\xi_*$ through the introduction of the duration as model covariate (e.g. Blanchet et al., 2016).

## 3   Data and study region

Four station datasets were used for the construction of intensity Annual Maxima Series (AMS) at different durations: the Daily Maxima Precipitation Data (DMPD) and the Hourly Canadian Precipitation Data (HCPD) datasets provided by Environment and Climate Change Canada (ECCC) and the MDDELCC [in french Ministère du Développement Durable, de l'Environnement et de la Lutte contre les Changements Climatiques] for Canada, and the Hourly Precipitation Data (HPD) and 15-Min Precipitation Data (15PD) datasets made available by the National Oceanic and Atmospheric Administration (NOAA) agency [http://www.ncdc.noaa.gov/data-access/land-based-station-data] for United States. The total number of stations was approximately 3400, with roughly 2200 locations having both DMPD and HCPD series, or both HPD and 15PD series. The majority of stations are located in the United States and in the southern and most densely populated areas of Canada. In northern regions the station network is sparse and the record length does not generally exceed 15 or 20 years. Moreover, for most of DMPD and HCPD stations, the annual recording period does not cover the winter season and available series generally include precipitation measured from May to October. For this reason, the *year* from which the annual maxima was sampled was limited to the recording season going from June to September for northern stations [stations located north of the $52^{nd}$ Parallel] and from June to September for the southern stations. As a result, 122 days a year were used for northern stations and 184 days a year for remaining stations.

Data were collected through a variety of instruments [e.g., standard, tipping-bucket, and Fischer-Porter rain gauges] and precipitation values were processed and quality-controlled using both automated and manual methods (CSA, 2012, *HPD and 15PD online documentation*). Most often, observations were recorded by tipping-bucket gauges with tip resolution from 0.1 mm to 2.54 mm (CSA, 2012; Devine and Mekis, 2008). 15 min series usually present the coarser instrument resolution, with a minimum non-zero value of 2.54 mm, observed for about 80.5% of 15PD stations. The effects of such a coarse instrument resolution on simple scaling estimates could be important leading to empirical $X_d$ cdfs becoming step-wise functions with a low number of steps. Some preliminary analyses aiming at evaluating these effects on SS estimates are presented in the sup-

plementary material [see Fig. S2 and S3]. However, the 15PD dataset is important considering the associated network density and its fine temporal resolution, and thus it has been retained for our study. The main characteristics of the available datasets are summarized in Table 1.

The scaling AMS datasets were constructed according to the following steps:

(i) Three duration sets were defined: a) 15 min to 6 h with a 15min step; b) 1 h to 24 h with a 1h step; c) 6 h to 168 h (7 days) with a 6h step. These duration sets are hereinafter referred to as Short-Duration (SD), Intermediate-Duration (ID), and Long-Duration (LD) datasets, respectively [see Figure 1 (a)].

(ii) Meteorological stations that were included in each final dataset were selected according to the following criteria: 1) precipitation series must have at least 85% of valid observations for each May to October (or June to September) period, otherwise

the corresponding year was considered as missing; 2) each station must have at least 15 valid years; 3) for each station, it was possible to compute AMS for all durations considered in the scaling dataset (e.g., HCPD and HPD stations were not included in the SD dataset because only hourly durations were available). Note that, in order to exclude outliers possibly associated with recording or measurement errors, extremely large observations were discarded and assimilated to missing data. In particular, as in some previous studies (e.g., Papalexiou and Koutsoyiannis, 2013; Papalexiou et al., 2013), an iterative procedure was

applied prior to step (ii)-1) to discard observations larger than 10 times the second largest value of the series.

(iii) A moving window was applied to 15PD, HCPD, and HPD series to estimate aggregated series at each duration. For DMPD series, a quality check was also implemented in order to guarantee that precipitation intensities recorded each day at different durations were consistent with each other. For instance, each pair of DMPD rainfall intensity [mm] $(x_{d_1}, x_{d_2})$ observed at durations $d_1 < d_2$ must respect the condition $x_{d_2}/x_{d_1} \geq d_1/d_2$ derived from the definitions of daily maximum rainfall intensity

and depth; otherwise all DMPD values recorded that day were discarded and assimilated to missing data.

(iv) For each selected station, annual maxima were extracted for each valid year and duration. For stations having both DMPD and HCPD series, or 15PD and HPD series, for each year, the annual maxima extracted from these two series were compared and the maximum value was retained as the annual maximum for that year.

Major characteristics of each scaling AMS dataset are reported in Table 2.

## 4    SS estimation through Moment Scaling Analysis (MSA)

Moment Scaling Analysis (MSA) for the SD, ID, and LD datasets was carried out to empirically validate the use of SS models for modeling AMS empirical distributions. Assessing the validity of the SS hypothesis for various duration intervals also aimed at determining the presence of different scaling regimes for precipitation intensity distributions.

In order to identify possible changes in the SS properties of AMS distributions, various *scaling intervals* were defined for the

MSA. In particular, all possible subsets with 6, 12, 18 and 24 contiguous durations were considered within each dataset. Figure 2 and Figure 3 show the 136 scaling intervals thereby defined: 40 scaling intervals for SD and ID, and 56 scaling intervals for

LD. For instance, the top left matrix of Fig. 2(a) presents the 6-duration scaling intervals 15 min - 1 h 30 min, 30min - 1 h 45 min, ..., 4 h 45 min - 6 h defined for the SD dataset [i.e. the 19 scaling intervals containing six contiguous durations defined with a 15min increment]. More schematically, Fig. 1(b) shows an example of the first five 6-duration scaling intervals for the ID dataset [i.e. 1 h - 6 h, 2 h - 7 h, ..., 5 h - 10 h, containing six contiguous durations defined with an increment of 1h]. This procedure was defined in order to evaluate the sensitivity of the SS estimates to changes in the first duration $d_1$ of the scaling interval and in the interval length [i.e. the number of durations included in the scaling interval].

For each scaling interval (for simplicity, their index has been omitted), the validity of the SS hypothesis was verified according to the following steps:

1. *MSA regression:* for each $q = 0.2, 0.4, \ldots, 2.8, 3$, the slopes $K_q$ of the log-log linear relationships between the empirical $q-$moments $\langle X_d^q \rangle$ of $X_{d_1}, X_{d_2}, \ldots, X_{d_D}$ and the corresponding durations $d_1, d_2, \ldots, d_D$ in the scaling interval $[d_1, d_D]$ were estimated by Ordinary Least Squares (OLS) [see Fig. 1 (c) for a graphic example]. Order $q \geq 3$ were not considered because of the possible biases affecting empirical high order moment estimates.

2. *Slope test:* to verify the SS assumption that the estimated $K_q$ exponents vary linearly with the moment order $q$, i.e. $K_q \approx Hq$, an OLS regression between the MSA slopes $K_q$ and $q$ was applied [see Fig. 1 (d)]. For the regression line $K_q = \hat{h}_0 + \hat{h}_1 \, q$, a Student's t-test was then used to test the null hypothesis $\boldsymbol{H}_0$: $\hat{h}_1 = K_1$. If $\boldsymbol{H}_0$ was not rejected at the significance level $\alpha = 0.05$, the SS assumption was considered appropriate for the scaling interval and the simple scaling exponent $H = K_1$ was retained.

3. *Goodness-of-Fit (GOF) test:* for each duration $d$, the goodness of fit of the $X_d$ distribution under SS was tested using the Anderson-Darling (AD) and the Kolmogorov-Smirnov (KS) tests. These tests aim at validating the appropriateness of the scale invariance property for approximating the $X_d$ cdf by the distribution of $X_{d,ss} = d^{-H} X_{d^*}$. To this end, each AMS, $\boldsymbol{x}_{d_j} = (x_{d_j,1}, x_{d_j,2}, \ldots, x_{d_j,i}, \ldots x_{d_j,n})$, recorded at duration $d_j$ was rescaled at the reference duration $d^*$ by inverting Eq. (2):

$$\boldsymbol{x^*}_{d_j} = \left( d_j^{\ H} x_{d_j,1}, d_j^{\ H} x_{d_j,2}, \, \ldots \, , d_j^{\ H} x_{d_j,i}, \ldots d_j^{\ H} x_{d_j,n} \right) \tag{8}$$

where $n$ represents the number of observations (years) in $\boldsymbol{x}_{d_j}$. Then, the pooled sample, $\boldsymbol{x}_{d^*}$, of the $D$ rescaled AMS, $\boldsymbol{x^*}_{d_j}$, was used to define $X_{d^*}$ under the SS assumption:

$$\boldsymbol{x}_{d^*} = \left( \boldsymbol{x^*}_{d_1}, \ldots, \boldsymbol{x^*}_{d_j}, \, \ldots \, , \boldsymbol{x^*}_{d_D} \cdot \right) \tag{9}$$

Since, in Eq. (9), $D$ represents the number of durations $d_j$ in the scaling interval, $n \times D$ rescaled observations were included in $\boldsymbol{x}_{d^*}$.

As in previous applications (e.g., Panthou et al., 2014), the AD and KS tests were then applied at significance level $\alpha = 0.05$ to compare the empirical distributions (Cunnane plotting formula, Cunnane, 1973) of the SS sample, $\boldsymbol{x}_{d,ss} = d^{-H} \boldsymbol{x}_{d^*}$, and the non-SS sample, $\boldsymbol{x}_d$. In fact, despite the low power of KS and AD tests for small sample tests, they represent the only suitable solution to the problem of comparing empirical cdfs when the data do not follow a normal distribution. Because both AD and KS are affected by the presence of ties in the samples (e.g., repeated values due to rounding or instrument resolution), a permutation test approach (Good, 2013) was used to estimate test p-values. According to this approach, data in $\boldsymbol{x}_d$ and $\boldsymbol{x}_{d,ss}$

were pooled and randomly reassigned to two samples having same sizes as the SS and non-SS samples. Then, the test statistic distribution under the null hypothesis of equality of the $X_{d,ss}$ and $X_d$ distributions was approximated by computing its value over a large set of random samples. Finally, the test p-value was obtained as the proportion of random samples presenting a test statistic value larger than the value observed for the original sample.

5    The SS model validity and the mean error resulting from approximating the $X_d$ distribution by the SS model were then evaluated in a cross-validation setting. For this analysis, each duration was iteratively excluded from each scaling interval and the scaling model re-estimated at each station by repeating steps 1 to 3 [MSA regression, Slope test, and GOF tests]. Predictive ability indices, such as the Mean Absolute Error (MAE) and the Root Mean Squared Error (RMSE) between empirical and SS distribution quantiles, were then estimated for highest quantiles for valid SS stations. In particular, to focus on return periods of practical interest for IDF estimation, only quantiles larger than the median were considered (i.e., only return periods greater than 2 years).

For each station $s$, the normalized RMSE, $\bar{\epsilon}_{x_{d,s}}$, was estimated:

$$\bar{\epsilon}_{x_{d,s}} = \frac{\epsilon_{x_{d,s}}}{\overline{x}_{d,s}} \tag{10}$$

where $\epsilon_{x_{d,s}}$ and $\overline{x}_{d,s}$ are, respectively, the RMSE and the mean value of all $X_d$ quantiles of order $p > 0.5$. Then, the average

over all stations of the normalized RMSE, $\bar{\bar{\epsilon}}_{x_d}$, was computed for each scaling interval and duration:

$$\bar{\bar{\epsilon}}_{x_d} = \frac{1}{n_s} \sum_{s=1}^{n_s} \bar{\epsilon}_{x_{d,s}} \tag{11}$$

where $n_s$ is the number of valid SS stations in the dataset. Note that $\bar{\bar{\epsilon}}_{x_d}$ is a measure of error, meaning that values of $\bar{\epsilon}_{x_{d,s}}$ closer to 0 correspond to a better fit than larger values.

## 4.1    Model estimation and validation

Figure 2 presents the results of steps 1 to 3 of the methodology for evaluating the SS validity. For all the three scaling datasets, no particular pattern was observed for slope test results, and at most $2\%$ of the stations within each scaling interval displaying a non linear evolution of the scaling exponent with the moment order. For this reason, Fig. 2(a)-(c) show, for each scaling interval and duration, the proportion of valid SS stations without differentiating for slope or GOF test results. As showed in the example in Fig.1(e), for each scaling interval, valid SS stations were defined as stations having not rejected both the Slope test

for the scaling interval and the GOF tests for each duration included in this scaling interval.

As expected, the proportion of valid SS stations decreased when the number of durations within the scaling interval increased and with decreasing $d_1$. This is particularly evident for short $d$ in SD and ID datasets. More GOF test rejections were observed for longer scaling intervals [not shown], due to the higher probability of observing large differences between $x_d$ and $x_{d,ss}$ quantiles when $x_{d,ss}$ had larger sample size and included data from more distant durations. However, several factors can

impact GOF test results when shorter $d_1$ are considered. First, GOF tests are particularly sensitive to the presence of very large values in short-duration samples. Second, when considering durations close to the temporal resolution of the recorded series

[i.e., 15 min in SD and 1 h in ID and LD], stronger underestimations could affect the measure of precipitation because intense rainfall events are more likely to be split between two consecutive time steps. Finally, preliminary analyses [Fig. S2 and S3 in the supplementary material] showed that the largest GOF test rejections could also be connected to the coarse instrument resolution of 15PD series, which, similar to the temporal resolution effect, induces larger measurement errors in the shortest

duration series. Note that comparable resolution issues were previously reported by some authors while estimating fractal and intermittency properties of rainfall processes (e.g., Veneziano and Iacobellis, 2002; Mascaro et al., 2013) and IDF (e.g., Blanchet et al., 2016).

Valid SS station proportions between 0.99 and 1 were always observed for GOF tests in ID and LD datasets, except for some durations shorter than 3 h (ID dataset) or 6 h (LD dataset). When considering both GOF and Slope test, with the exception

of some durations $\leq 1$ hour, the proportion of stations satisfying SS was higher than 0.9, and the majority of scaling intervals [65%, 90%, and 98% of the scaling intervals in SD, ID, and LD, respectively] included at least 95% of valid SS stations. For each scaling interval, only valid SS stations were considered in the rest of the analysis.

These findings were also confirmed by cross-validation experiments. The proportion of valid SS stations resulting from cross-validation Slope and GOF tests were similar, event if slightly lower, to proportions displayed in Fig. 2 [see Fig. S4 of the

supplementary material].

Figure 3 presents, for each scaling interval and duration, the station average, $\overline{\epsilon}_{x_d}$, of the normalized RMSE. These graphics show that mean relative errors on intensity quantiles did not generally exceed 5% of the precipitation estimates for 6-duration scaling intervals [Fig. 3, first col.]. Greater errors were observed for durations at the border of the scaling intervals. Not surprisingly, this result underlines that, in a cross-validation setting, both the MSA estimation of $H$ and the $X_{d,ss}$ approximation are

less sensitive to the exclusion of an inner duration of the scaling interval than to the exclusion of $d_1$ or $d_D$. Conversely, the extrapolation under SS of the $X_d$ distribution is generally less accurate for durations at the boundaries or outside the scaling interval used to estimate $H$. Moreover, as for the valid SS station proportion, the performances of the model deteriorated with decreasing $d_1$ and with increasing scaling interval length, especially for durations at the border of the scaling intervals. However, for more that 70% of 12-, 18-, and 24-duration scaling intervals, $\overline{\overline{\epsilon}}_{x_d} \leq 0.1$ for each duration included in the scaling

interval. $\overline{\overline{\epsilon}}_{x_d} \geq 0.25$ were observed for 15 min in 12-duration or longer scaling intervals, pointing out the weaknesses of the model in approximating short duration extremes when the scaling interval included durations $\geq 3$ h.

## 4.2   Estimated scaling exponents and their variability

In order to evaluate the sensitivity of SS to the considered scaling interval, the variability of $H$ with $d_1$ has been analyzed. Then, the spatial distribution of the scaling exponents for each scaling interval was studied to assess the uncertainty in $H$ estimation

and the dependence of SS exponents on local geoclimatic characteristics.

Investigating the variability of the scaling exponent with the scaling interval is particularly important since, if SS is assumed to be valid between some range of durations, one should expect that $H$ remains almost unchanged over the various scaling intervals included in this range. For this reason, the variation $\Delta_{H_{(j)}}$ of the scaling exponents computed for overlapping scaling

intervals having the same $d_1$ but different lengths was analyzed. For each station and $d_1$, $\Delta_{H_{(j)}}$ was defined as:

$$\Delta_{H_{(j)}} = H_{(j)} - H_{(6)} \tag{12}$$

where $j = 12, 18,$ or $24$ represents the number of durations considered in the specified scaling interval, $H_{(j)}$ is the corresponding scaling exponent, and $H_{(6)}$ is the scaling exponent estimated for the 6-duration scaling interval having the same $d_1$. If SS
is appropriate over a range of durations, $\Delta_{H_{(j)}}$ is expected to be small for scaling intervals defined within this range.

Figures 4(ii)-(iv) show for all relevant scaling intervals, the median, Interquantile Range (IQR), and quantiles of order 0.1 and 0.9 of the $\Delta_{H_{(j)}}$ distribution over valid SS stations. Adding new durations to the scaling intervals the median $\Delta_{H_{(j)}}$, as well as its IQR, increased for all $d_1$. Nonetheless the median scaling exponent variation was generally smaller than 0.05, except for a relatively small proportion of stations. Equally important, $|\Delta_{H_{(j)}}|$ was generally centered on 0 and for all $d_1 \geq 1$ h more than
50% of stations had $|\Delta_{H_{(12)}}| \leq 0.025$ (SD dataset) and $|\Delta_{H_{(18)}}| \leq 0.03$ (ID dataset) [Fig. 4 (ii)-(iii)].

For some stations, a dramatic difference could exist in IDF estimations obtained with the different definitions of the scaling interval. For instance, for the 24-duration scaling interval "1h - 24h" (ID dataset), the median $\Delta_{H_{(24)}}$ was equal to 0.047 [Fig. 4(iv) b)]. For the interval "15min - 6h" (SD dataset), $\Delta_{H_{(24)}}$ was even larger, with a median scaling exponent variation approximately equal to 0.087 and with 25% of stations having $\Delta_{H_{(24)}} \geq 0.11$ [Fig. 4(iv) a)]. Finally, changes in $H$ values were also
important when comparing 6- and 12-duration scaling intervals when $d_1 \leq 1$ h (SD and ID datasets) and in LD dataset [Fig. 4 (ii)].

The median, Interquantile Range (IQR), and quantiles of order 0.1 and 0.9 of the $H$ distribution across stations, are presented in Fig. 4(i) for each 6-duration scaling interval. The smallest median $H$ values were observed for $d_1 \leq 30$ min in Fig. 4 (a-i), and for the longest $d_1$s in Fig. 4 (c-i). Scaling intervals beginning at 15 and 30 min also displayed the smallest variability across
stations. Although fewer stations were available for these intervals (only 15PD stations were used and the number of valid SS stations was smaller), this result is consistent with previous reports in the literature demonstrating that $H$ values are spatially more homogeneous for short durations.

A larger dispersion of $H$ values was observed when $d_1$ ranged between approximately 1 h and 5 h, in particular in the SD dataset, for which the $10^{th}$-$90^{th}$ percentile difference almost covered the entire range of observed $H$ values [Fig. 4 (i)]. This
result could be partially explained by the use of scaling intervals having equally spaced durations. This implies that the mean distance between the logarithms of durations in the scaling interval decreases as $d_1$ increases. Hence, the OLS estimator of $H$ used in the MSA regression may have larger variance for longer $d_1$, especially when scaling intervals include few durations. Larger uncertainty may thus have an impact on the $H$ estimation for the longest $d_1$ scaling intervals of SD. However, as showed in next sections, $H$ spatial distribution may also explain the greater variability of the scaling exponent for $d_1$ greater than a few
30   hours.

Largest median $H$ were observed for $d_1$ greater than 10 hours [Fig. 4 (b-i)] and lower than 2 days [Fig. 4 (c-i)], with approximately half of the stations having $H \geq 0.8$. This means that a stronger scaling (i.e., larger $H$ values) is needed to relate extreme precipitation distributions at approximately 12-hours to distributions at daily and longer scales. It may therefore be expected that the stations characterized by $H$ closer to 1 are located in geographical areas where differences in precipitation distributions

are important among temporal scales included in these scaling intervals.

Examples of the spatial distributions of the scaling exponent are given in Fig. 5 and 6 for the first and last $d_1$ for each interval length and dataset, respectively. Since only one 24-duration scaling interval was defined for both the SD and ID datasets, only scaling intervals containing 6, 12, and 24 (Fig. 5) or 18 (Fig. 6) durations are presented. This avoids the redundancy of showing

twice the "15min - 6h" (SD dataset) and "1h - 24h" (ID dataset) scaling intervals.

Generally, the scaling exponent displayed a strong spatial coherence and varied smoothly in space, although a more scattered distribution of $H$ characterizes maps in Fig. 6. In this last figure, the local variability of $H$ may be attributed to the larger estimation uncertainties affecting longer $d_1$ scaling intervals, as previously mentioned. Meaningful spatial variability and clear spatial patterns emerged for $d_1 \geq 1$ h. In fact, for stations located in the interior and southern areas of the continent, a shift

from weaker scaling regimes (smaller $H$) to higher $H$ values was observed as $d_1$ increases [e.g., second and third rows of Fig. 5]. On the contrary, a smoother evolution of $H$ over the scaling intervals characterized the northern coastal areas, especially in north-western regions, and the Rockies, where $H > 0.75$ values were rarely observed even for greater $d_1$ values.

## 5 Regional analysis

Regional differences in scaling exponents were investigated. Only the results for the 6-duration scaling intervals are presented,

similar results having been obtained for longer scaling intervals [see the supplementary material, Fig. S6 and S7 for 12- and 18-duration scaling intervals]. Stations were pooled into six climatic regions based on the classification suggested by Bukovsky (2012) [see Fig. 7]. Stations outside the domain covered by the Bukovsky regions were attributed to the nearest region. Regions with less than 10 stations were not considered (regions without colored borders in Fig. 7); regions A1 ($W\_Tun$) and A2 ($NW\_Pac$) were kept separated since only 14 stations were available in region A1 ($W\_Tun$) for ID and LD datasets.

To provide deeper insights about regional features of precipitation associated with specific scaling regimes two variables related to the precipitation events observed within AMS were also analyzed: the mean number of events per year, $\bar{N}_{eve}$, and the mean wet time per event, $\bar{T}_{wet}$, contributing to AMS within each scaling interval. For a given year and station, annual maxima associated to different durations of a given scaling interval were considered to belong to the same precipitation event if the time intervals over which they occurred overlapped. The mean wet time per event contributing to AMS, $\bar{T}_{wet}$, was defined as

the mean number of hours with non-zero precipitation within each event. Details on the calculation of $\bar{N}_{eve}$, $\bar{T}_{wet}$, and the corresponding results are presented in the supplementary material [Sect. S2 and Fig. S5 and S6].

### 5.1 Regional variation of the scaling exponents.

Figure 8 shows the distribution of $H$ within each region. Three types of curves can be identified. First, curves in Fig. 8 (a) to (c) have a characteristic smooth S shape. Conversely, Fig. 8 (d) displays a rapid increase of $H$ for scaling intervals defined in

ID and LD datasets until $d_1 = 2$ days, preceded and followed by two plateaus: one plateau for the longest $d_1$ with remarkably high $H$ values, and one for the shortest $d_1$ with small $H$ values. Finally, an inverse-U-shaped curve can be seen in Fig. 8 (e) and (f), with globally high $H$ values already reached at sub-daily durations in dry regions (E).

For $d_1 \leq 24$ h, Fig. 8 (a) displays lower values of $H$ than Fig. 8 (e)-(f), meaning that smaller variation in AMS moments are observed in A1 and A2 when the scale is changed. This difference can be partially explained by the weaker impact of convection processes in generating very short duration extremes in rNorth-West coastal regions with respect to southern areas (regions E and F). For northern regions, in fact, the transition between short and long duration precipitation regimes may be smoothed out by cold temperatures which moderate short-duration convective activity, especially for $W\_Tun$ (region A1). The topography characterizing the northern pacific coast may then explain the smoothing effect for the curve of region $NW\_Pac$ (A2). In this case, in fact, the precipitation rates at daily and longer scales are enhanced by the orographic effect acting on synoptic weather systems coming from the Pacific Ocean (Wallis et al., 2007).

Similarly, mountainous regions in C [Fig. 8 (c)] displayed the smallest variations of $H$ over $d_1$, indicating that analogous scaling regimes characterize both short- and long-duration scaling intervals. Again, this may be related to the important orographic effects of precipitation in these regions that are involved in the generation of extremes for both sub-daily and multi-daily time scales. The mean number of events per year in regions A and C was higher than in regions E-F, in particular for SD scaling intervals, and displayed steeper decreases with increasing $d_1$ [Fig. S5 (a) and (c) in the supplementary material].

Main differences between regions B and A were the stronger scaling regimes observed in B, which were mainly due to contributions from stations located in the south-eastern part of the $E\_Bor$ region (not shown). For scaling intervals in the ID dataset, region B was also characterized by the highest mean number of events per year, with most of the stations presenting $\bar{N}_{eve} > 2$ for $d_1 = 1$ h and $d_1 = 2$ h and sharp decreases of $\bar{N}_{eve}$ with increasing $d_1$ [Fig. S5 (b) in the supplementary material]. Moreover, a remarkably large range of $\bar{N}_{eve}$ was observed for $1$ h $\leq d_1 \leq 6$ h, suggesting that B may be highly heterogeneous.

Two distinct scaling regimes can be observed for $SW\_Pac$ (region D) at, respectively, $d_1 \leq 3$ h (SD dataset) and $d_1 \geq 2$ days (ID dataset) [region D in Fig. 8 (d)]. These plateaus may be interpreted by recalling that $1 - H = H_{depth}$. On the one hand, the low and constant $H$ observed for $d_1 \leq 3$ h indicates that the average precipitation depth increases with duration at the same growth rate for all these intervals. On the other hand, $H$ approximately equal to 0.9 at daily and longer durations demonstrates that the average precipitation depth associated with long-duration annual maxima remained roughly unchanged when the duration increased from 1.5 to 7 days ($\lambda^{H_{depth}} \approx 1$ in Eq. (3)). This, along with the fact that the scaling exponent increased almost monotonically for $1$ h $\leq d_1 \leq 24$ h (ID and LD datasets), suggests that extremes at durations shorter than $\sim 3$ h (SD dataset) drive annual maxima precipitation rates at longer scales, with the rapid and continuous decay in mean intensity caused by the increasing size of the temporal scale of observation.

For $SW\_Pac$ (region D), the relative absence of long-lasting weather systems able to produce important extremes for long durations, was confirmed by the analysis of $\bar{N}_{eve}$ and $\bar{T}_{wet}$ [see Fig. S5 and S6 of the supplementary material]. In fact, the mean number of events per year was relatively high for short durations (the median $\bar{N}_{eve}$ is equal to 1.82 for $d_1 = 15$ min and to 1.4 for $d_1 = 1$h), while it rapidly decreased below 1.1 events per year for $d_1 \geq 6$ h (ID dataset) and for $d_1 \geq 18$ h (LD dataset). With the exception of $d_1 = 6$ h (LD dataset), at least 90% of $SW\_Pac$ stations had $\bar{N}_{eve} \leq 1.25$ for all $d_1 > 3$ h. In other regions, median $\bar{N}_{eve}$ were never smaller than 1.1 for the SD and ID datasets, except for $d_1 \geq 12$h in region E.

These results suggests that both the distinctive topography of the west coast and the characteristic large-scale circulation of the south-west areas of the continent are crucial factors determining the transition between the two scaling regimes in region D.

Median $H$ values displayed inverse-U shapes for the remaining regions with very small IQR, despite the high number of valid SS stations: a slow transition from lower to higher $H$ is observed approximately between 1 h and 12 h (region E) or 30 h (region F). The strongest scaling regimes were observed for 1 h $\leq d_1 \leq$ 2 days in arid western regions [Fig. 8 (e)], while median $H$ values greater than 0.8 were only observed for approximately 6 h $\leq d_1 \leq$ 2 days in more humid areas [8 (f)]. In

both region E and F, very short-duration extremes are typically driven by convective processes, while a transition to different precipitation regimes may be expected between 1 h and a few hours. However, $H$ shows a smoother increase in Fig. 7 (f) with respect to Fig. 7(e). This may indicate that in eastern areas [region F] sub-daily duration extremes are more likely associated to embedded convective and stratiform systems, or to mesoscale convective systems, which are less active in western dry areas of region E (Kunkel et al., 2012). On the contrary, differences between short- and long-duration extreme precipitation intensity

seem stronger for south-western dry regions [Fig. 8 (e)], where less intense summer extremes are expected compared to eastern areas [see supplementary material, Fig. S1]. In particular, $H$ tended to scatter in a range of higher values for approximately 1 h $\leq d_1 \leq$ 12 h indicating that precipitation intensity moments strongly decrease as the duration increases.

In summary, these results suggest a regional effect on precipitation scaling of both local geographical characteristics, such as topography or coastal effects, and general circulation patterns. In general, the weakest scaling regimes were observed for short

$d_1$ and along the west coast of the continent and seem to be connected to scaling intervals and climatic areas characterized by homogeneous weather processes. Low $H$ values correspond in fact to small variations in AMS distribution moments. On the contrary, stronger scaling regimes were observed for longer $d_1$ in the other regions of the study area. This indicates that important changes occur in AMS moments across duration and, thus, in extreme precipitation features. According to these results, it would be important to take into account the climatological information included in the scaling exponent to improve

SS and IDF estimation. Even more important, these results give useful guidelines for modeling the spatial distribution of $H$, which could help for the definition of IDF relationships at non-sampled locations.

## 6   Simple Scaling GEV etimation

Results presented in this section are limited to a descriptive analysis of GEV parameter estimates for 6-duration scaling intervals. Similar results were generally obtained for 12-, 18-, and 24-duration intervals [see supplementary material, Fig. S10

to S16]. An assessment of the potential improvements carried out by Simple Scaling GEV (SS GEV) models with respect to non-SS GEV models is also presented.

In our study, the Probability Weighted Moment (PWM) procedure was applied to estimate SS-GEV parameters $\mu_*$, $\sigma_*$, and $\xi_*$ [Eq. (7)] from $\boldsymbol{x}_{d^*}$ [Eq. (9)]. For each duration $d$, PWM were also used to estimate non-SS parameters $\mu_d$, $\sigma_d$, and $\xi_d$ from each of the non-SS samples $\boldsymbol{x}_d$. Preliminary comparisons of various estimation methods [PWM, classical ML estimators, and

GAM-ML; see Sect. 2.2], showed that PWM slightly outperformed the other methods.

Quantiles estimated from the SS and the non-SS GEV were compared with empirical quantiles. Global performance measures, such as RMSE, were computed to evaluate the overall fit of the estimated GEV to the empirical $X_d$ distributions. In particular, mean errors between SS and non-SS quantile estimates and empirical quantiles were compared using the relative total RMSE

ratio, $R_{\overline{rmse}}$, defined as:

$$R_{\overline{rmse}} = \frac{[\overline{R}_{ss} - \overline{R}_{non-ss}]}{\overline{R}_{non-ss}} \tag{13}$$

where

$$\overline{R}_{mod} = \sum_{d=d_1}^{D} \frac{\epsilon_{d,mod}}{\bar{x}_d} \tag{14}$$

represents the normalized mean square difference between model and empirical quantiles of order $p > 0.5$ for all the durations included in the scaling interval. See Eq. 10 for the definition of $\epsilon_{d,mod}$ for each station.

## 6.1   Estimated SS GEV parameters

Figure 9 presents the distributions over valid SS stations of the SS GEV parameters rescaled at $d_* = 1$ h [Fig. 9 (a) and (b)] and $d_* = 24$ h [Fig. 9 (c)].

For the SD dataset, even for scaling intervals which did not include the reference duration $d^*$, the $\mu_*$ and $\sigma_*$ distributions appeared to be similar to the non-SS $\mu_d$ and $\sigma_d$ distributions [Figure 9, first row]. Similarly, for 6 h $\leq d_1 \leq$ 2 days in the LD dataset, the SS location and scale parameter distributions are in relatively close agreement with the corresponding non-SS parameter distributions. Conversely, for the ID dataset, both $\mu_*$ and $\sigma_*$ distributions are more positively skewed than the corresponding non-SS distributions. Finally, for $d_1 \geq$ 2 days in the LD dataset, $\mu_*$ and $\sigma_*$ had distributions shifted toward lower

values than $\mu_{24h}$ and $\sigma_{24h}$. Moreover, the relative differences $\Delta_\mu = (\mu_* - \mu_d)/\mu_d$ and $\Delta_\sigma = (\sigma_* - \sigma_d)/\sigma_d$ were estimated for each station, duration, and scaling interval. Two important results came out of this analysis [see Figures S11 and S12 of the supplementary material]. On the one hand, median values of $\Delta_\mu$ and $\Delta_\sigma$ were generally smaller than $\pm 5\%$ and $\pm 10\%$, respectively. On the other hand, $\Delta_\sigma$ showed large positive values when $\xi_d = 0$ (i.e. Gumbel distributions), while small $\Delta_\sigma < 0$ were estimated when $\xi_d \neq 0$ [not shown for conciseness]. These results are interesting since the estimation of the scale parameter

$\sigma$ of a GEV distribution may be biased when the shape parameter is spuriously set to zero ($\xi = 0$). Hence, while non-SS $\mu_d$ values can be considered to be accurate estimates of the $X_d$ location parameter, small uncertainties should be expected for the scale parameter only when the $\xi_d$ value is correctly assessed. In addition, $\mu_*$ and $\sigma_*$ displayed a strong spatial coherence. Their spatial distributions were characterized by an obvious North-West to South-East gradient [Fig. 10 shows examples for the scaling intervals 15min - 1.5h, 1h - 6h, and 6h - 36h].

Notable differences between SS GEV and non-SS GEV estimates were observed for the shape parameter [Fig. 9, third col., and Fig. 11]. Firstly, for cases having shape parameters strictly different from zero [third column of Fig. 9], $\xi_*$ absolute values were smaller than non-SS $\xi_d$ absolute values. Secondly, the distributions of $\xi_*$ across stations were generally more peaked around their median value than the corresponding non-SS distributions. Finally, for the non SS model the majority of stations had shape parameter $\xi_d$ non-significantly different from zero, while the fraction of SS GEV shape parameters $\xi_* \neq 0$ was always

greater than 39% (Hosking et al., 1985, asymptotic test for PWM GEV estimators applied at level 0.05;). In particular, for each duration, non-SS models estimated light-tailed distributions (i.e., $\xi_d = 0$) for more than 85% of the stations, except that for $d =$

15 min and $d = 30$ min [Fig. 11, first col.]. Conversely, for all scaling intervals with $d_1 > 15$ min, SS GEV shape parameters were significantly different from zero for 40% to 45% of valid SS stations [Fig. 11, second col.]. Moreover, when using scaling intervals of 12 durations or more, the proportion of $\xi_* > 0$ was always important [greater than 35% for all 18- and 24-duration scaling intervals; see the supplementary material, Fig. S10].

The previous results suggest that pooling data from several durations may effectively reduce the sampling effects impacting the estimation of $\xi$, allowing more evidence of non-zero shape parameters, and, in many cases, of heavy tailed ($\xi > 0$) AMS distributions. This conclusion is consistent with previous reports, namely that 100- to 150-year series are necessary to unambiguously assess the heavy-tailed character of precipitation distributions (e.g., Koutsoyiannis, 2004b; Ceresetti et al., 2010). These studies typically reported values of $\xi \approx 0.15$ (e.g., Koutsoyiannis, 2004b), which are close to $\xi_*$ values estimated in the
present analysis for cases with $\xi_* > 0$.

However, uncertainties on $\xi_*$ estimates remain important. Support for this comes from the spatial distribution of $\xi_*$, which was still highly heterogeneous, with local variability dominating at small scales [e.g., Fig. 10, third col.].

## 6.2 Improvement with respect to Non-SS models

The proportion of series for which the SS model RMSE, $\epsilon_{d,ss}$, was smaller than the non-SS GEV RMSE, $\epsilon_{d,non-ss}$, was
15 analyzed [see the supplementary material, Fig. S11]. For cases with non-zero $\xi_*$, more than 60% of stations had $\epsilon_{d,ss} <$ $\epsilon_{d,non-ss}$ over most scaling intervals and durations. The 6-duration scaling intervals "15 min - 1 h 30 min" (SD dataset) and "1 h - 6 sih" (ID dataset) showed the largest fractions of stations with increasing errors. On the contrary, increasing errors ($\epsilon_{d,ss} > \epsilon_{d,non-ss}$) were observed for all scaling intervals and durations for most stations (generally more than 70%) having $\xi_* = 0$.
Figure 12 presents the $R_{\overline{rmse}}$ distribution over valid SS stations. When the SS shape parameters were not significantly different from zero [Fig. 12, second col.], the relative increases in total RMSE were usually smaller than 0.1 in SD dataset and only scaling intervals with $d_1 < 1$ h had greater $R_{\overline{rmse}}$. For the ID and LD datasets, the medians of the total relative RMSE ratio distributions were smaller than 0.05 for $d_1 \geq 4$ h and $d_1 \geq 24$ h, respectively. Furthermore, more than 90% of stations had $R_{\overline{rmse}} < 0.125$ for $d_1 \geq 6$ h (ID dataset) and $d_1 \geq 30$ h (LD dataset). When $\xi_* \neq 0$, an increase of the mean error in high order
quantile estimates was observed for $d_1 = 15$ min (SD dataset) and $d_1 = 1$ h (ID dataset) for at least half of the stations [Fig. 12, first col.; note the different scale on the y-axis]. However, for all other $d_1$, negative $R_{\overline{rmse}}$ values were observed for the majority of stations for all scaling intervals, with a median reduction up to 30% of the mean error. Note that also for 12- and 18-duration scaling intervals the median $R_{\overline{rmse}}$ where generally negative for $d_1 > 1$ h and $\xi_* \neq 0$ [Fig. S14 and S15 of the supplementary material]. Conversely, $R_{\overline{rmse}}$ increased for the majority of stations in all 24-duration scaling intervals having $d_1 < 12$ h [Fig.
S17 of the supplementary material]. Note also that no particular spatial pattern characterized the $R_{\overline{rmse}}$ estimates.

## 7 Discussion and conclusion

This study investigated simple scaling properties of extreme precipitation intensity across Canada and the United States. The ability of SS models to reproduce extreme precipitation intensity distributions over a wide range of sub-daily to weekly durations was evaluated. The final objective was to identify duration intervals and geographical areas for which the SS model can be used for an efficient production of IDF curves.

The validity of SS models was empirically confirmed for the majority of the scaling intervals. In particular, based on the comparison of SS distributions to empirical quantiles, the hypothesis of a scale-invariant shape of the $X_d$ distribution held for all duration intervals spanning from 1 h to 7 days. Less convincing results were obtained for durations shorter than 1 h, especially for the longest scaling intervals (24-duration intervals). One possible explanation is that the coarse instrument resolution of the available 15 min series may strongly impact both the validation tools (for instance, GOF tests) and SS estimates. These results provide important operative indications concerning the inner and outer cut-off durations for AMS scaling and show the importance of a deeper analysis to evaluate the impact of dataset characteristics (e.g., their temporal and measurement resolutions, or the series length) on the scale invariant properties of extreme precipitation.

The majority of the estimated scaling exponents ranged between 0.35 and 0.95, showing a smooth evolution over the scaling intervals and a well-defined spatial structure. Six geographical regions, initially defined according to a climatological classification of North America into 20 regions, displayed different features in terms of scaling exponent values. Specifically, distinct median values of $H$ were observed for the various geographical regions, each characterized by a different precipitation regime. This is consistent with results reported in the literature for some specific regions and smaller observational datasets (e.g., Borga et al., 2005; Nhat et al., 2007; Ceresetti et al., 2010; Panthou et al., 2014, and references therein). Moreover, while small and smooth changes of $H$ over the scaling intervals were observed in regions containing the majority of stations, one region, $SW\_Pac$, displayed two dramatically distinct scaling regimes separated by a steep transition occurring between a few hours and 24 h. These results limit the applicability of SS models in $SW\_Pac$, and were connected to the local features of intense precipitation events by the analysis of the mean number of events per year and the mean wet time of these events.

Weak scaling regimes, characterized by relatively small $H$ values ($H$ close to 0.5), were generally observed for scaling intervals containing very short durations (e.g, less than 2 h) and for regions on the west coast of the continent [regions A1, A2, and D; see Fig. 8]. For these scaling intervals and regions, we can expect that extreme precipitation events observed at various durations will have similar statistical characteristics, being governed by homogeneous weather processes.

The interpretation of high $H$ values (e.g., $H > 0.8$), observed between 1 and several days, depending on the region, is more complex. These scaling regimes correspond to mean precipitation depth that varies little with duration. This suggests an important change in precipitation regimes occurring at some durations included in the scaling interval. One interesting example was region $SW\_Pac$ (region D) for scaling intervals of durations longer than 1 day . In this case, the analysis of the mean number of events per year sampled in AMS suggested that very few long-duration extreme events were produced by large-scale dynamic precipitation systems.

For scaling intervals of durations longer than 4 days, scaling exponents seemed to converge to approximately 0.7 for all regions,

except west coast regions (regions A1, A2, and D).

These results suggest that SS represents a reasonable working hypothesis for the development of more accurate IDF curves. This may have important implications for infrastructure design and risk assessment for natural ecosystems, which would benefit from a more accurate estimation of precipitation return levels. Besides, the spatial distribution of the scaling exponent and

its dependency on climatology should be taken into account when defining SS duration intervals for practical estimation of IDF. The accuracy of the SS approximation may in fact depend on the range of considered temporal scales. Equally critical, estimated $H$ values were found to gradually evolve with the considered scaling intervals. In this respect, interesting extensions of the analysis should consider methods for the quantification of the uncertainty in $H$ estimations as well as the possibility of modeling the scaling exponent as a function of both the observational duration and the AMS distribution quantile/moment

order, i.e. by the use of a multiscaling (MS) framework for IDFs. Equally important, the events sampled by the AMS also showed different statistical features within different geographical regions and some specific results [e.g., for the $SW\_Pac$ region] stimulate the interest for an analysis of the scaling property of extreme precipitation by the use of a temporal stochastic scaling approach.

The evaluation of SS model performances under the assumption of GEV distributions for AMS intensity was then performed.

Results indicate that the proposed SS GEV models may lead to a more reliable statistical inference of extreme precipitation intensity than that based on the conventional non-SS approach. In particular, a better assessment of the GEV shape parameter seems possible when pooling data from several durations under the scaling hypothesis. The use of the SS approximation may introduce biases in high quantile estimates when AMS distributions move drastically away from perfect scale invariance (short durations and/or longest scaling intervals). Nonetheless, decreases in the SS GEV $RMSE$ with respect to non-SS GEV models

for $d_1$ longer than a few hours and/or scaling intervals shorter than 24 durations indicate that quantile errors in IDF estimates can be generally reduced.

Caution is advised when interpreting these results due to the fact that high order empirical quantiles were used as reference estimates of true $X_d$ quantiles, which could be a misleading assumption especially when available AMS are short. Moreover, two important limitations of the presented SS approach must be stressed. Firstly, a more comprehensive assessment of the

scaling exponent uncertainty and of the influence of dataset characteristics on the estimation of AMS simple scaling is recommended for a reliable estimation of Simple Scaling IDF curves. Secondly, the proposed model relies on the implicit hypothesis of stationarity of AMS over the observed period while growing evidence supports the ongoing changes in extreme precipitation intensity, frequency, duration, and spatial patterns as a result of climate change (e.g., Hartmann et al., 2013; Westra et al., 2014; Donat et al., 2016). In particular, short duration extreme rainfall is expected to respond to global warming with a

different sensitivity to temperature than those expected at daily or longer time scales (e.g., Westra et al., 2014; Lenderink and Attema, 2015; Wasko and Sharma, 2017; Barbero et al., 2017) which implies a change in the temporal scaling properties of precipitation over time.

Hence, considering these limitations and our general results, any future extension of this study should investigate the possibility of introducing spatial information in scaling models as well as the characterization of possible evolution of the scaling

exponent in a warmer climate in order to identifying valuable approaches allowing non-stationarity of SS model parameters.

## 8 Data availability

The 15-Min Precipitation Data (15PD) and Hourly Precipitation Data (HPD) were freely obtained from NOAA/Climate Prediction Center (CPC) [http://www.ncdc.noaa.gov/data-access/land-based-station-data]. Houly Canadian Precipitation Data (HCPD) and Maximum Daily Precipitation Data (DMPC) for Canada were acquired from Environment and Climate Change Canada (ECCC) and from the MDDELCC of Québec [data available upon request by contacting *Info-Climat@mddelcc.gouv.qc.ca*].

*Acknowledgements.* Silvia Innocenti's scholarship was partly provided by the Consortium OURANOS. Financial support for this project was also provided by the Collaborative Research and Development Grants program from the Natural Sciences and Engineering Research Council of Canada. They also thank Guillaume Talbot for preliminary data processing and Dikra Khedhaouiria for useful discussions.

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

**Table 1.** List of available datasets and their main characteristics.

| Dataset | Region | N. of stations | Operational period[b] | Temporal resolution | Prevalent[c] resolution [mm] |
|---|---|---|---|---|---|
| Daily Maxima Prec. Data[a] (DMPC) | Canada | 370 | 1964-2007 | 1, 2, 6, 12 h | 0.1 (82.25%) |
| Hourly Canadian Prec. Data (HCPD) | Canada | 665 | 1967-2003 | 1 h | 0.1 (70%) |
| Hourly Prec. Data (HPD) | USA | 2531 | 1948-2013 | 1 h | 0.254 (82.5%) |
| 15-Min Prec. Data (15PD) | USA | 2029 | 1971-2013 | 15 min | 2.54 (80.42%) |

[a] Daily maxima depth series over a 24-hour window beginning at 8:00 AM.

[b] Main station network operational period corresponding to $25^{th}$ percentile of the first recording year and the $75^{th}$ percentile of the last recording year of the stations.

[c] Prevalent instrument resolution, estimated by the lowest non-zero value for each series, and corresponding percentage of stations with this resolution.

**Table 2.** Final datasets used in scaling analysis and corresponding AMS characteristics.

| Scaling dataset | Durations | N. of Stations | Mean series length [yr] | Max series length [yr] |
|---|---|---|---|---|
| SD[a] | 15min, 30min, ..., 6h | 1083 | 20 | 36 |
| ID | 1h, 2h, ...,24h | 2719 | 37.4 | 66 |
| LD | 6h, 12h, ..., 168h | 2719 | 37.4 | 66 |

[a] Only 15PD series.

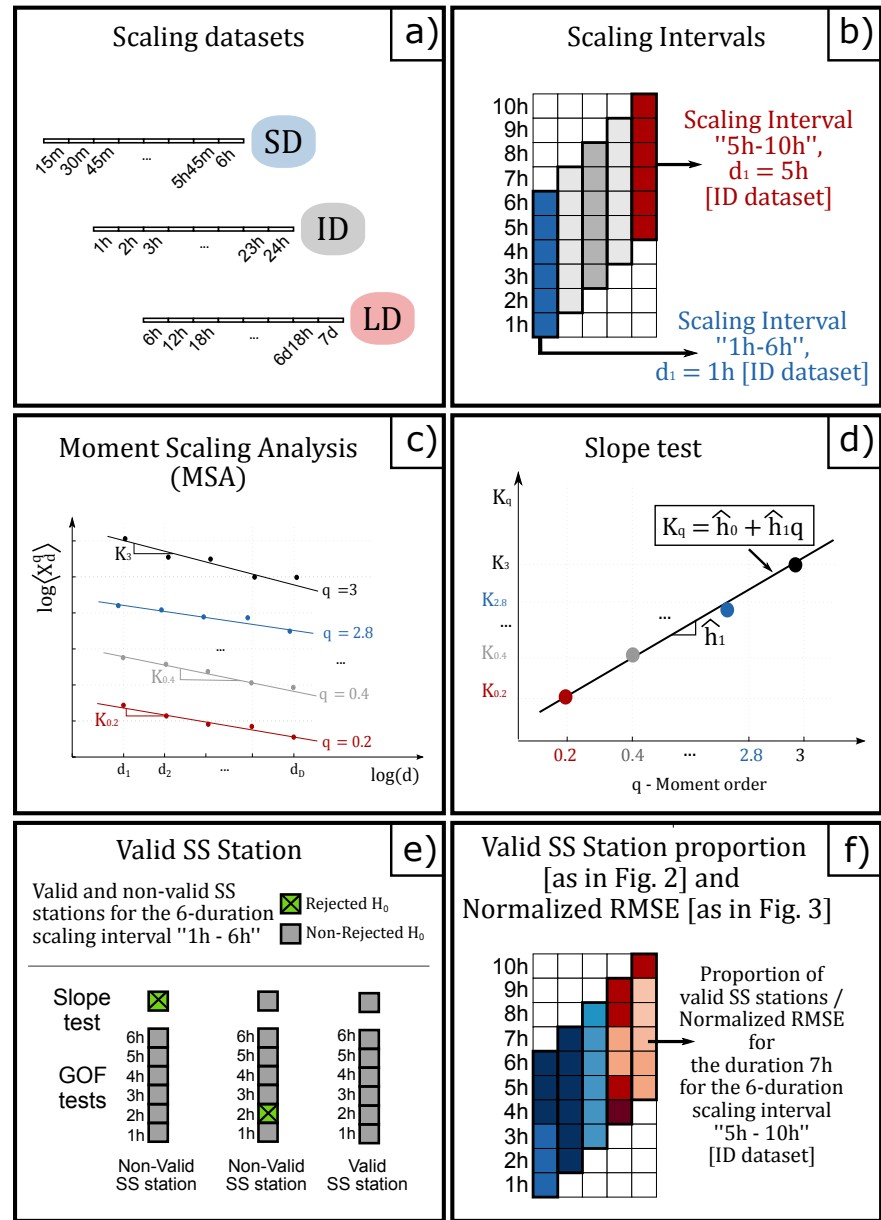

**Figure 1.** Methodology steps: a) Definition of the SD, ID, and LD scaling datasets. b) Identification of durations and scaling intervals within each matrix of Fig. 2 and 3; c) Moment Scaling Analysis (MSA) regression for the estimation of the slope coefficients $K_q$; d) Slope test: regression of $K_q$ on the moment order $q$ and Student's t-test for the null hypothesis $\boldsymbol{H}_0: \hat{h}_1 = K_1$; e) Examples of valid and non-valid SS stations according to the Slope and GOF tests; f) Example of valid SS station proportion values and Normalized RMSE values, $\overline{\overline{r}}_{x_d}$, as represented, in Fig. 2 and 3.

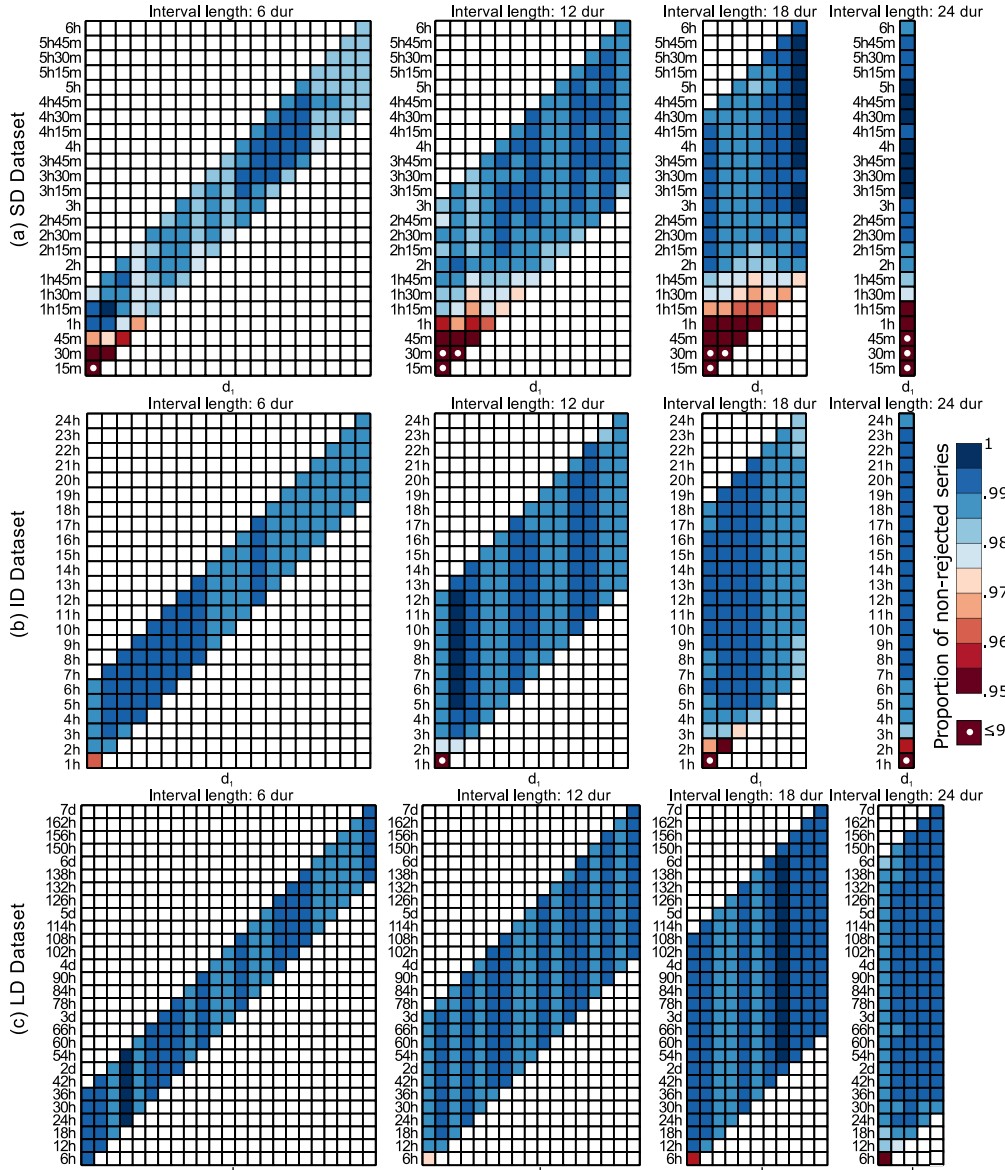

**Figure 2.** Proportion of stations satisfying both the Slope and GOF tests applied at the 0.95 confidence level, for each duration (vertical axis) and scaling interval (horizontal axis) for the SD, ID, and LD datasets [row a), b), and c) respectively]. White circles indicate proportions between 0.25 and 0.90. See Fig. 1 (b) and (f) for the identification of durations and scaling intervals within each matrix.

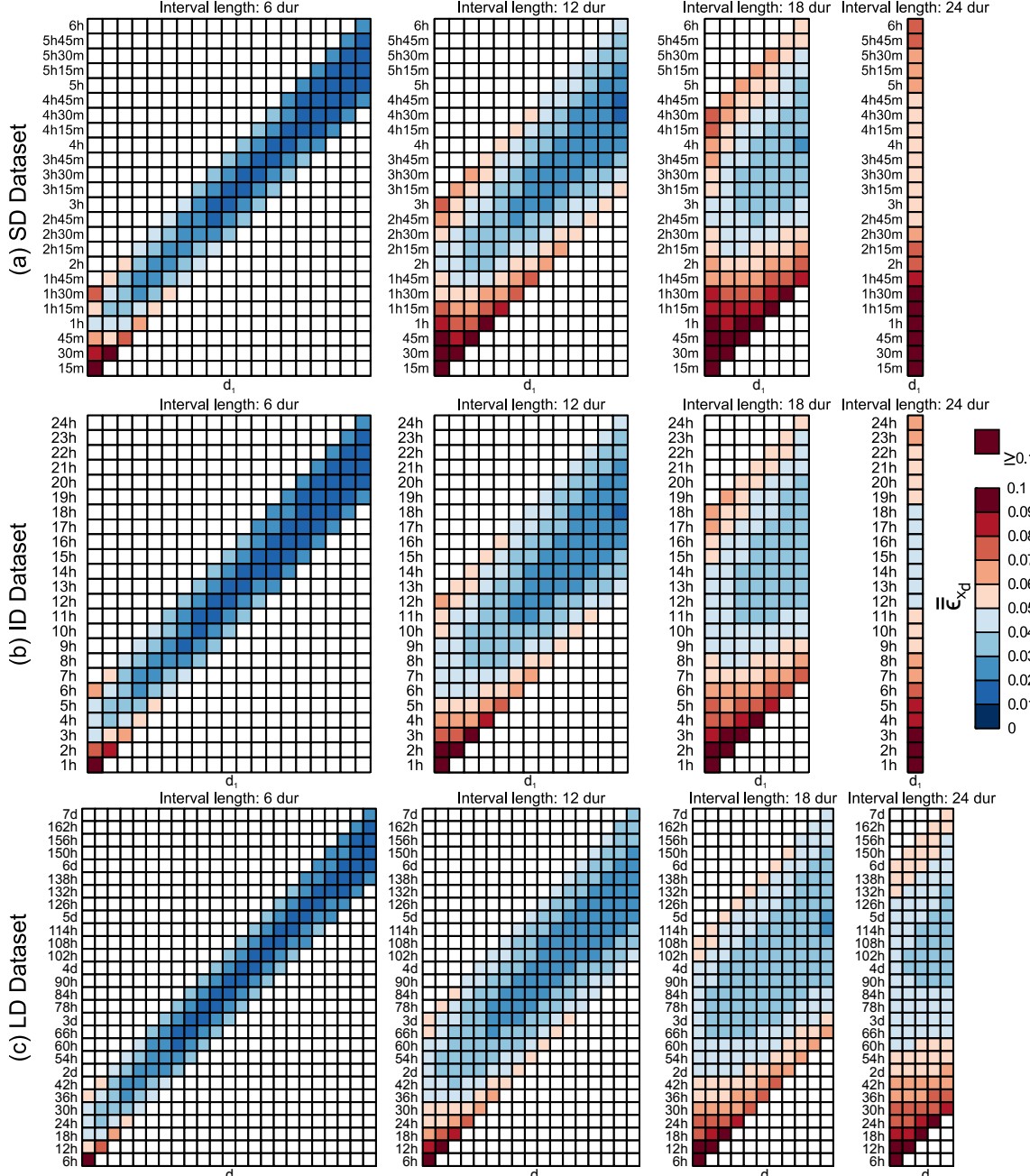

**Figure 3.** Cross-Validation Normalized RMSE averaged over all valid SS stations ($\bar{\bar{r}}_{x_d}$) for each duration (vertical axis) and scaling interval (horizontal axis) in the SD, ID, and LD datasets [row a), b), and c) respectively]. White circles indicate values between 0.15 and 0.3. See Fig. 1 (b) and (f) for the identification of durations and scaling intervals within each matrix.

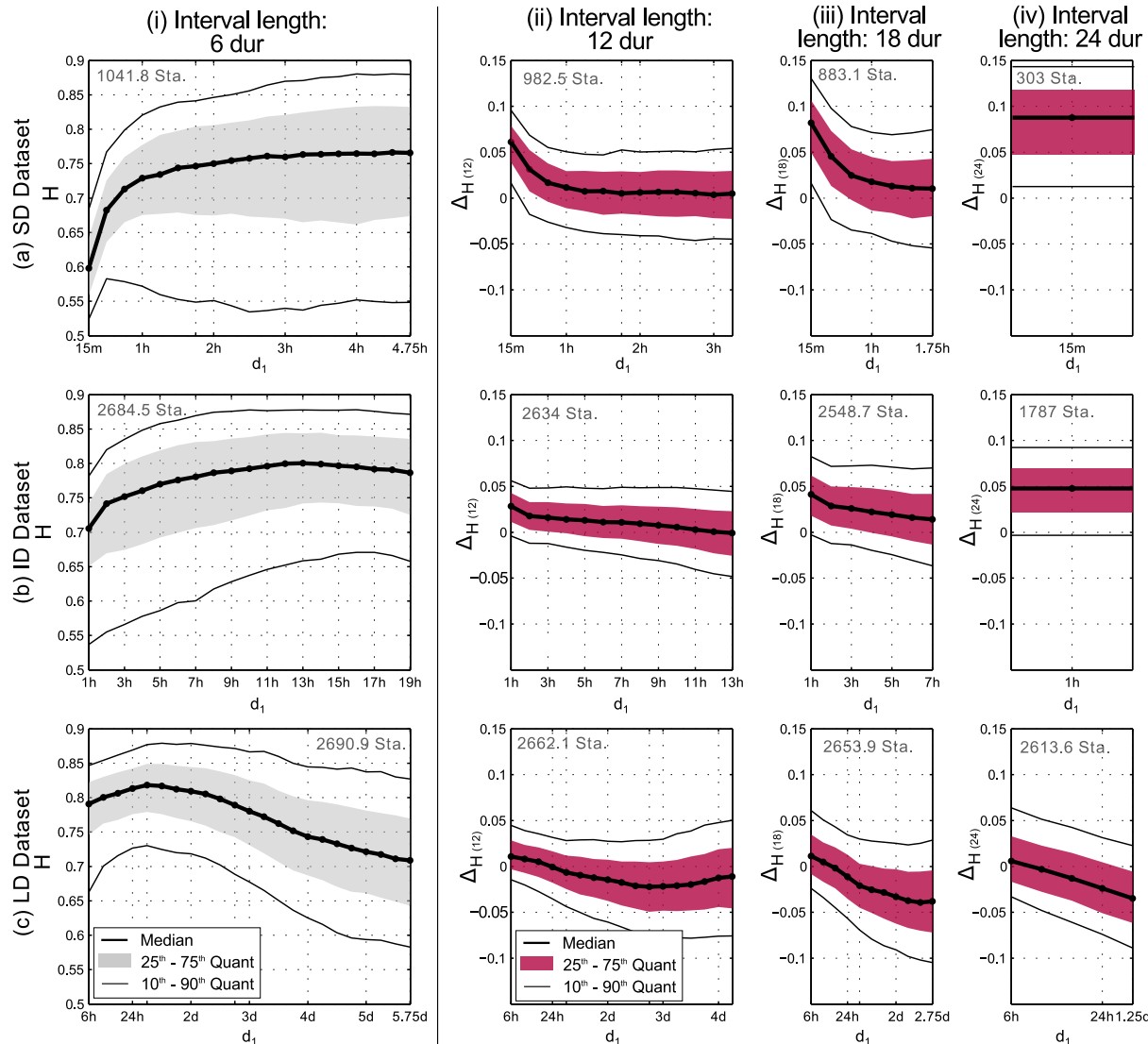

**Figure 4.** Col. (i): Median and relevant quantiles of the scaling exponent distribution over all valid SS stations for each 6-duration scaling interval. Col. (ii)-(iv): Median and relevant quantiles of the distribution of the scaling exponent deviation $\Delta_{H_{(j)}}$ [defined in Eq. (12)]. The average number of valid SS stations over the scaling intervals (identified by their first duration, $d_1$) is indicated at the top of each graph.

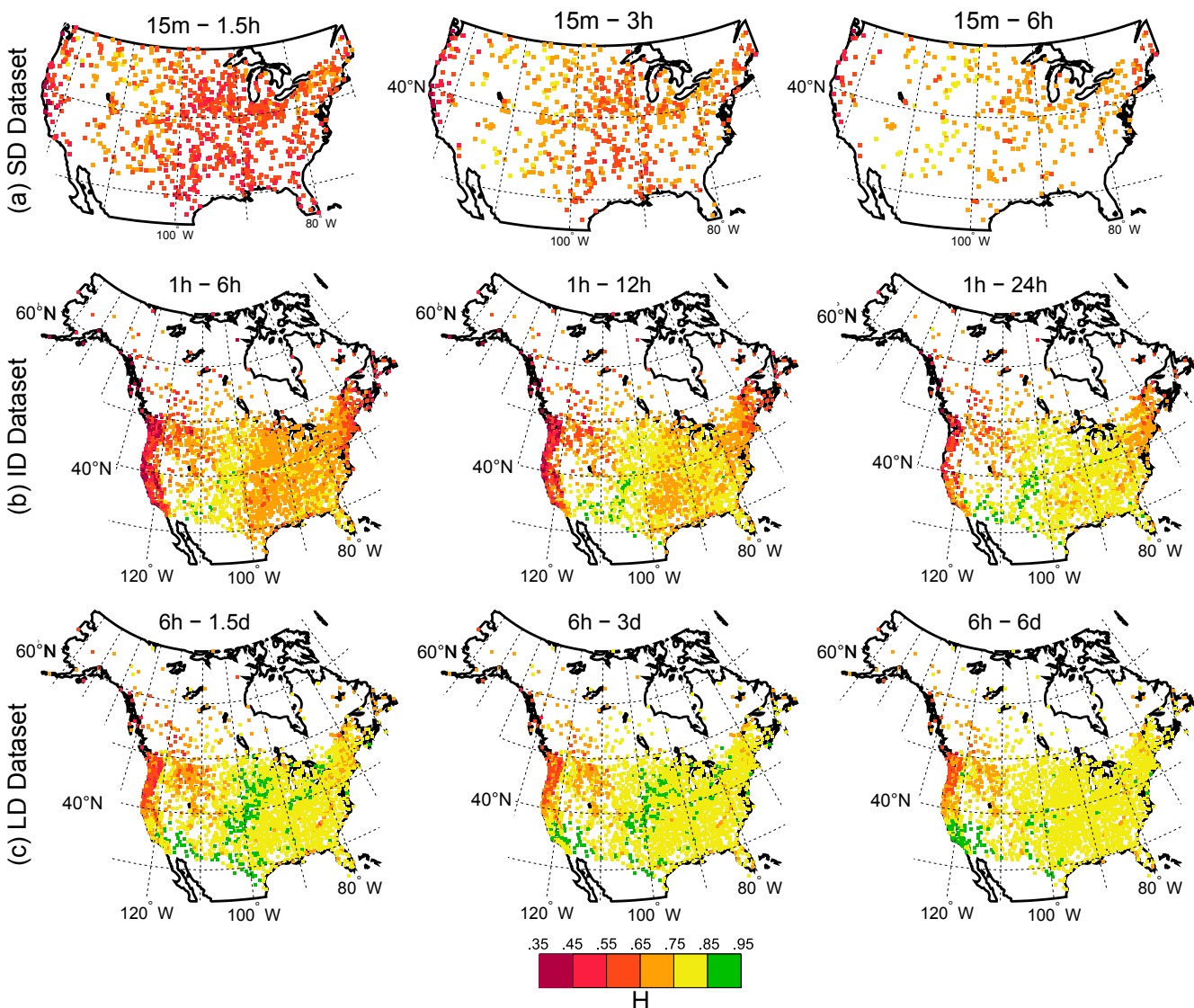

**Figure 5.** Spatial distribution of the scaling exponent for the first (i.e. with minimum $d_1$) 6-, 12-, and 24-duration scaling intervals (first, second, and third col., respectively) for SD, ID, and LD datasets (first, second, and third row, respectively). These scaling intervals correspond to the first column of matrices in Fig. 2 and 3.

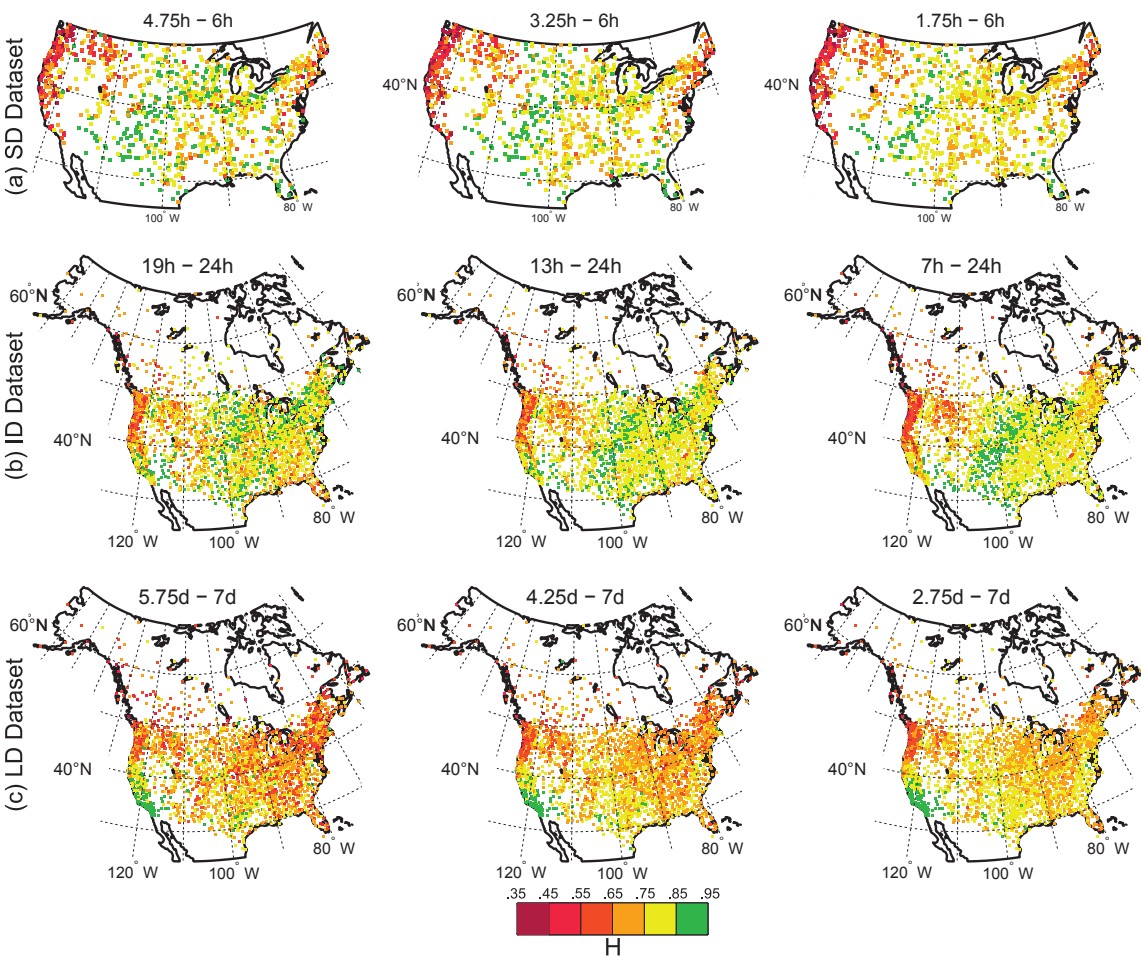

**Figure 6.** Spatial distribution of the scaling exponent for the last (i.e. with maximum $d_1$) 6-, 12-, and 18-duration scaling intervals (first, second, and third col., respectively) for SD, ID, and LD datasets (first, second, and third row, respectively). These scaling intervals correspond to the last column of matrices in Fig. 2 and 3.

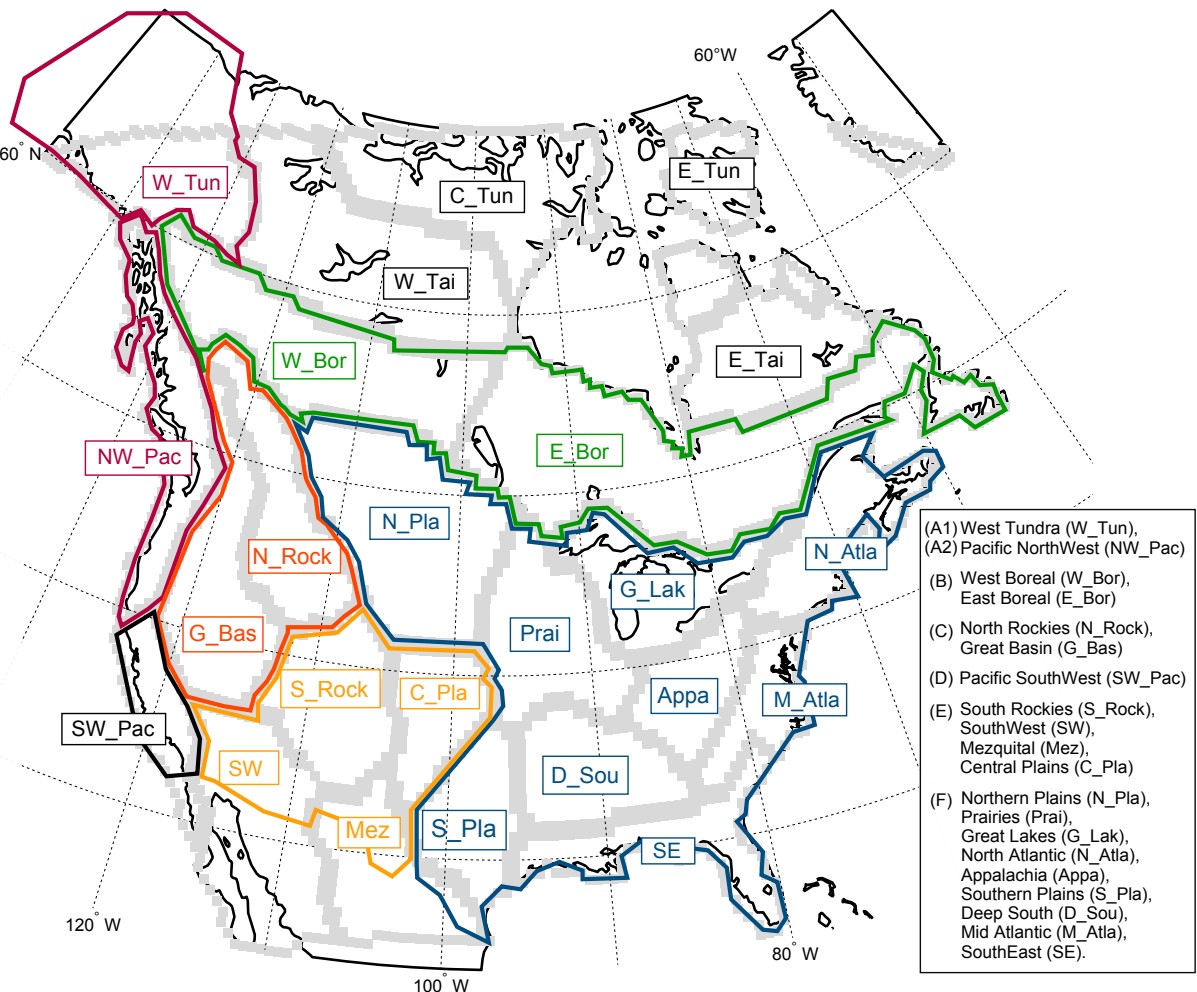

**Figure 7.** Climatic regions of Bukovsky (2012) [grey borders] and regions defined for this analysis [regions A1 to F in the legend; colored borders]. Abbreviations for each region are in parenthesis.

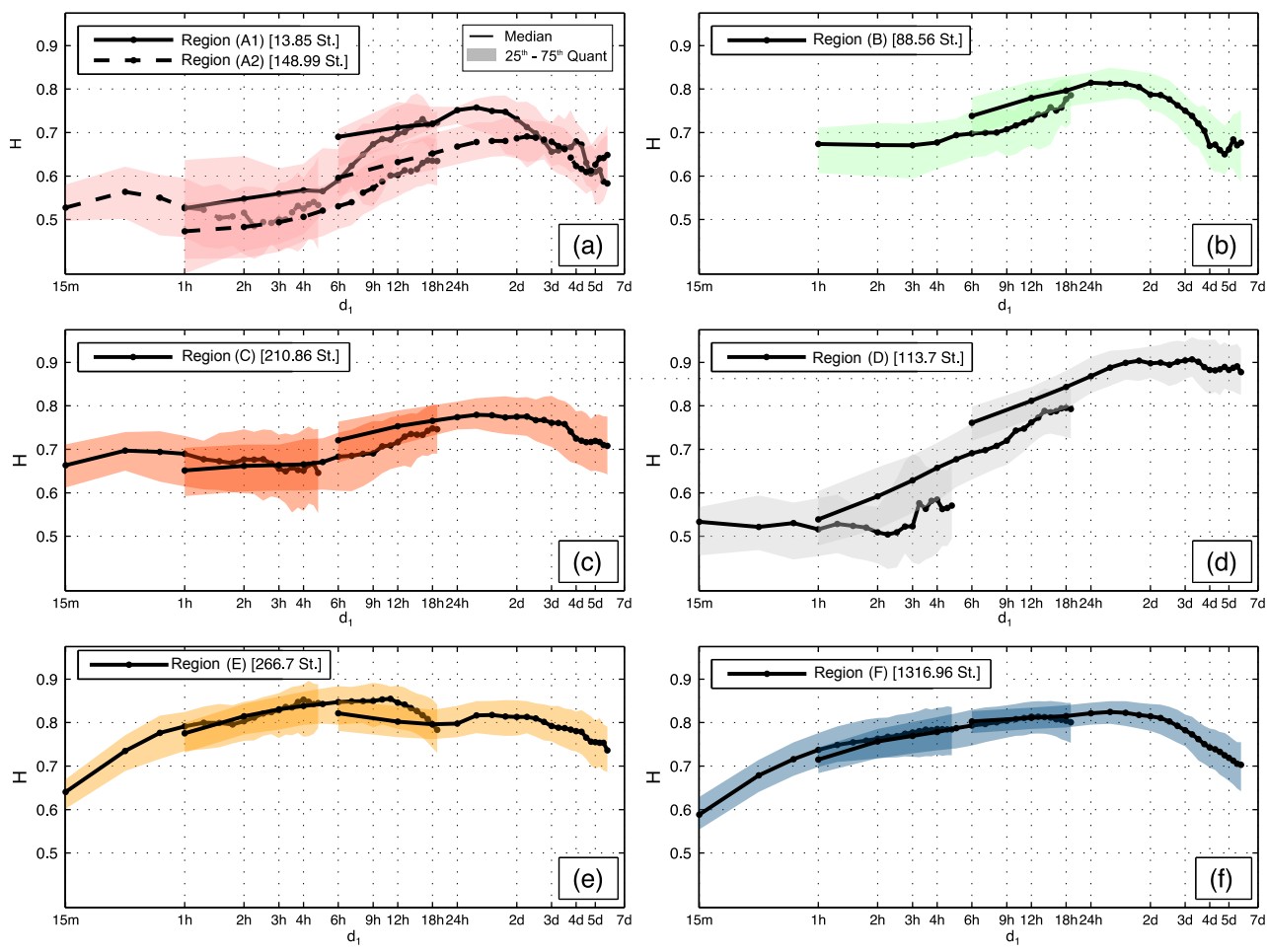

**Figure 8.** Median and Interquantile Range (IQR) of the scaling exponent distribution over valid SS stations within each region of Fig. 7 for 6-duration scaling intervals for the SD (left curve), ID (central curve), and LD (right curve) datasets. For each region, the mean number of valid SS stations over the scaling intervals is indicated in brackets in the legend. See Fig. 7 for region definition.

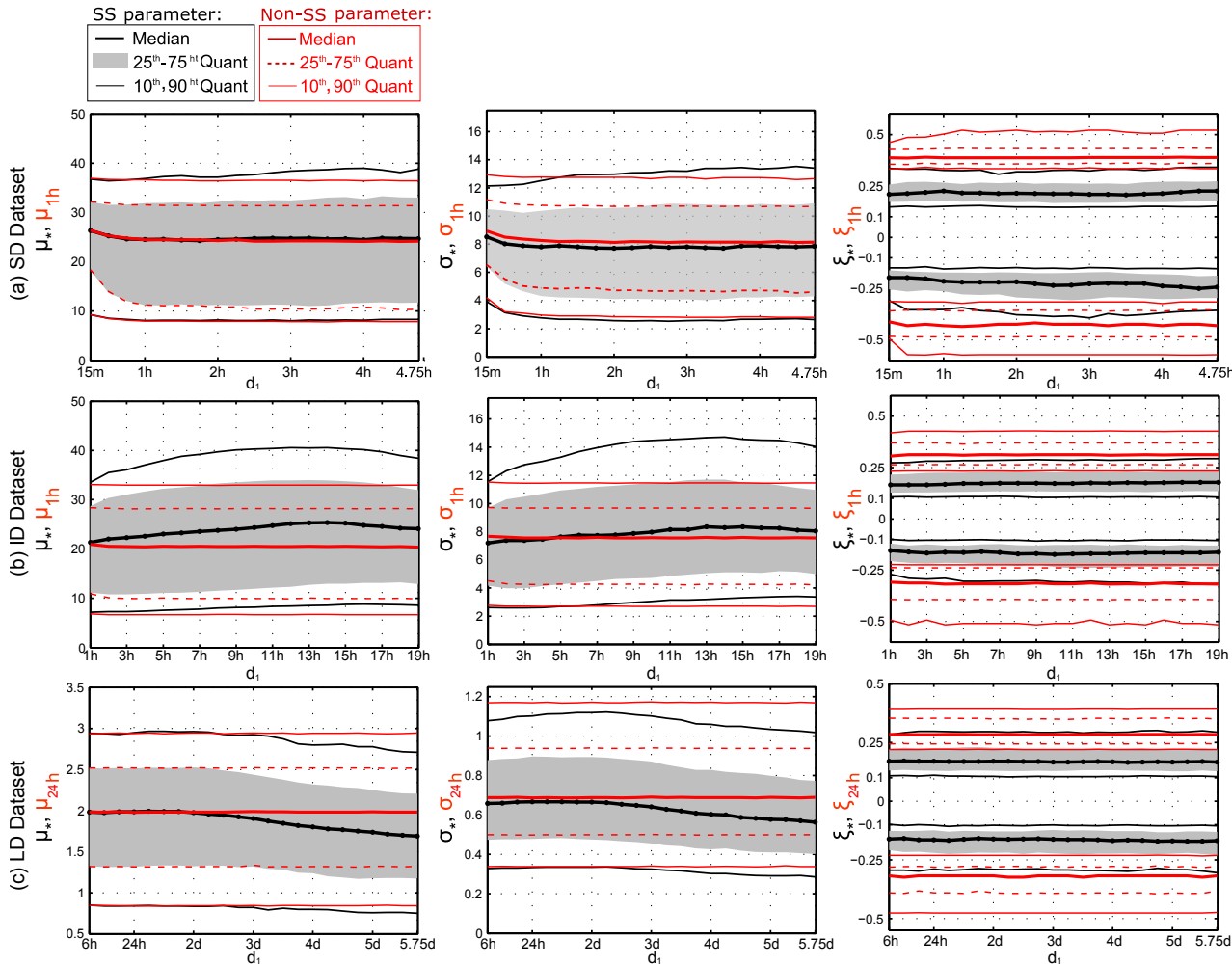

**Figure 9.** Distribution over valid SS stations of SS GEV parameters (gray and black lines) for 6-duration scaling intervals and non-SS GEV parameters (red solid and dashed lines) for reference durations. Location and scale parameters (first and second col., respectively) are scaled at $d_* = 1h$ (SD and ID datasets) and $d_* = 24h$ (LD dataset). Distributions for the shape parameter (third col.) are presented for $\xi > 0$ and $\xi < 0$, excluding cases where $\xi = 0$ (Gumbel distribution).

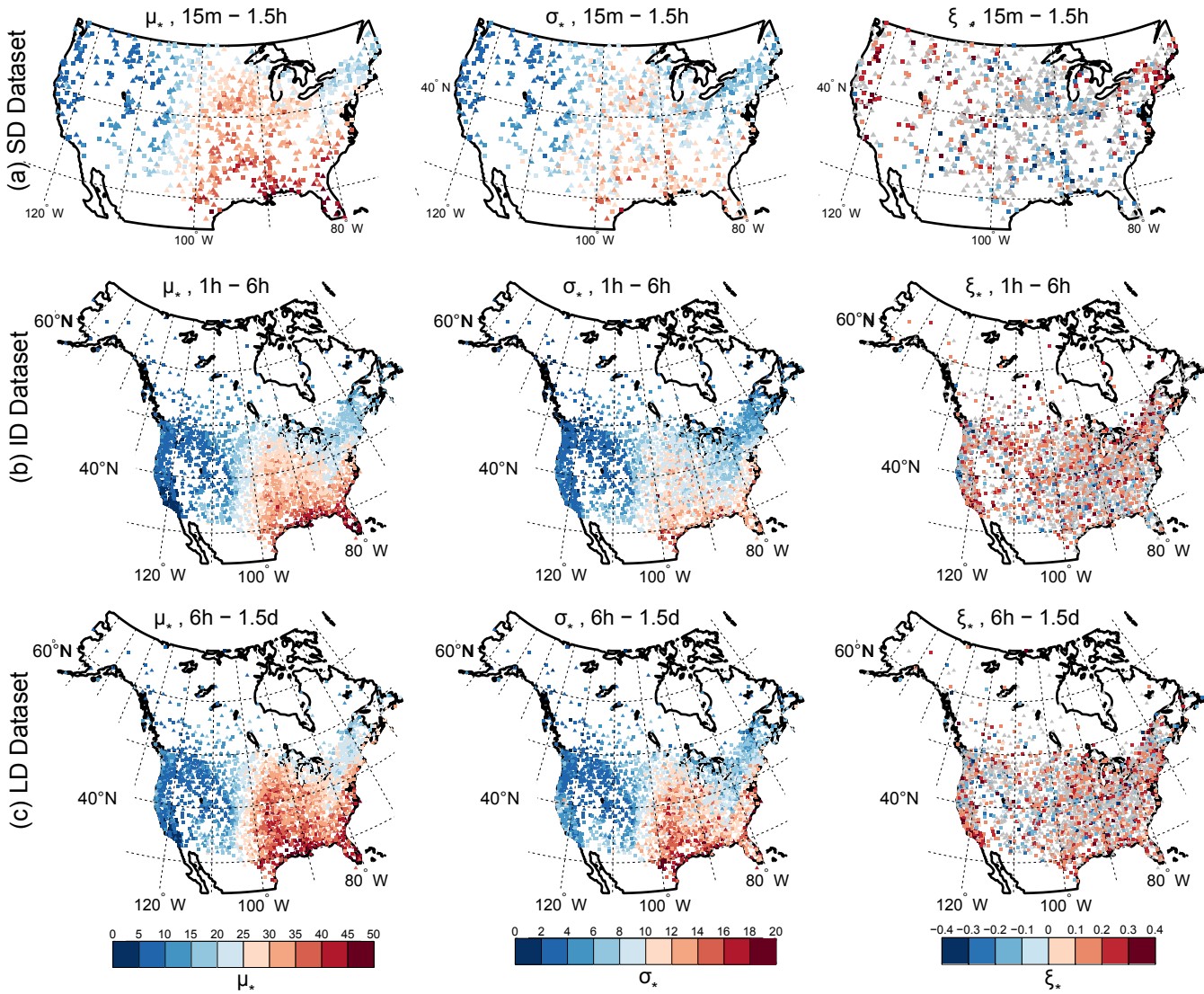

**Figure 10.** Spatial distribution over valid SS stations of SS GEV position (first col.), scale (second col.), and shape ($3^{rd}$ col.; gray symbols indicate Gumbel distributions, $\xi_* = 0$) parameters scaled at $d_* = 1h$ for the first 6-duration scaling interval (i.e. interval with minimum $d_1$) of: SD (a), ID (b), and LD (c) datasets.

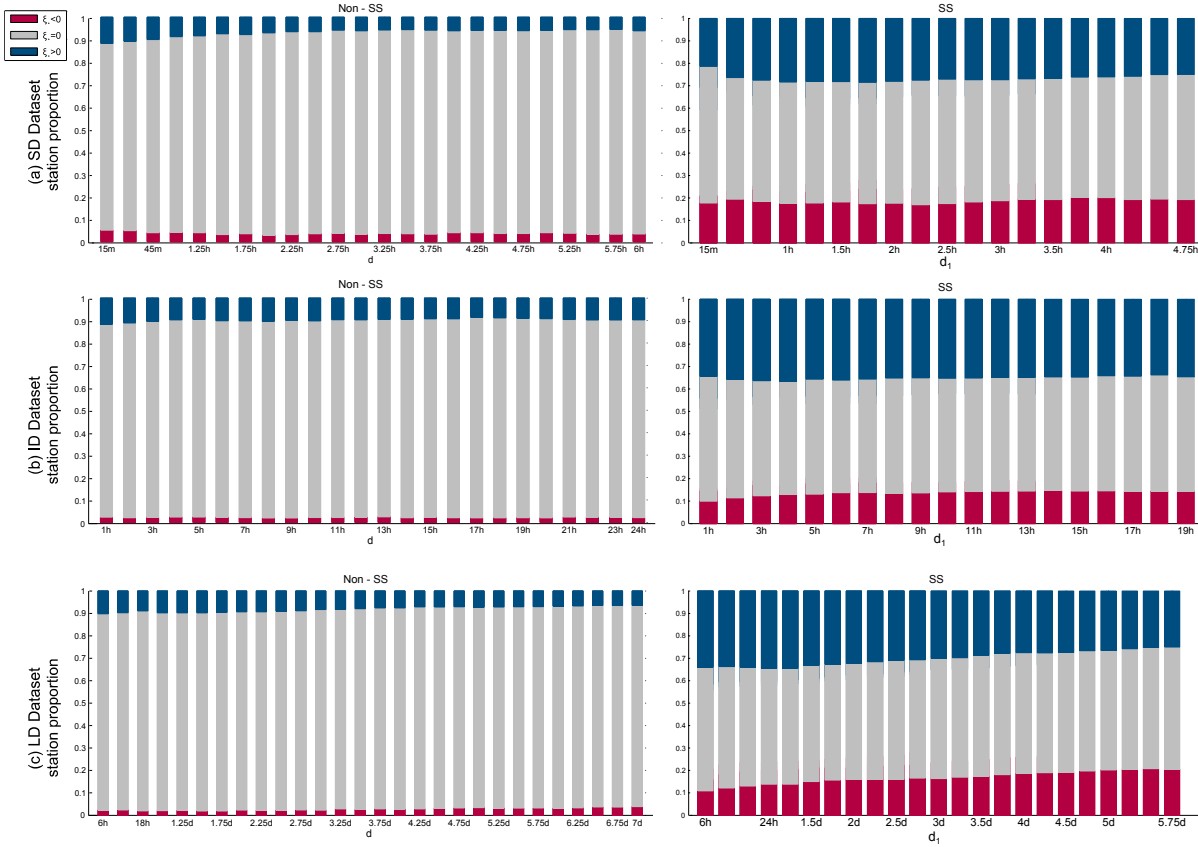

**Figure 11.** Stacked histograms of the fractions of valid SS stations with $\xi < 0$ (in red), $\xi = 0$ (in grey), and $\xi > 0$ (in blue) resulting from the Hosking test applied at the 0.95 confidence level for each duration (non-SS GEV, first col.) and each 6-duration scaling interval (SS GEV, second col.) for: SD (a), ID (b), and LD (c) datasets.

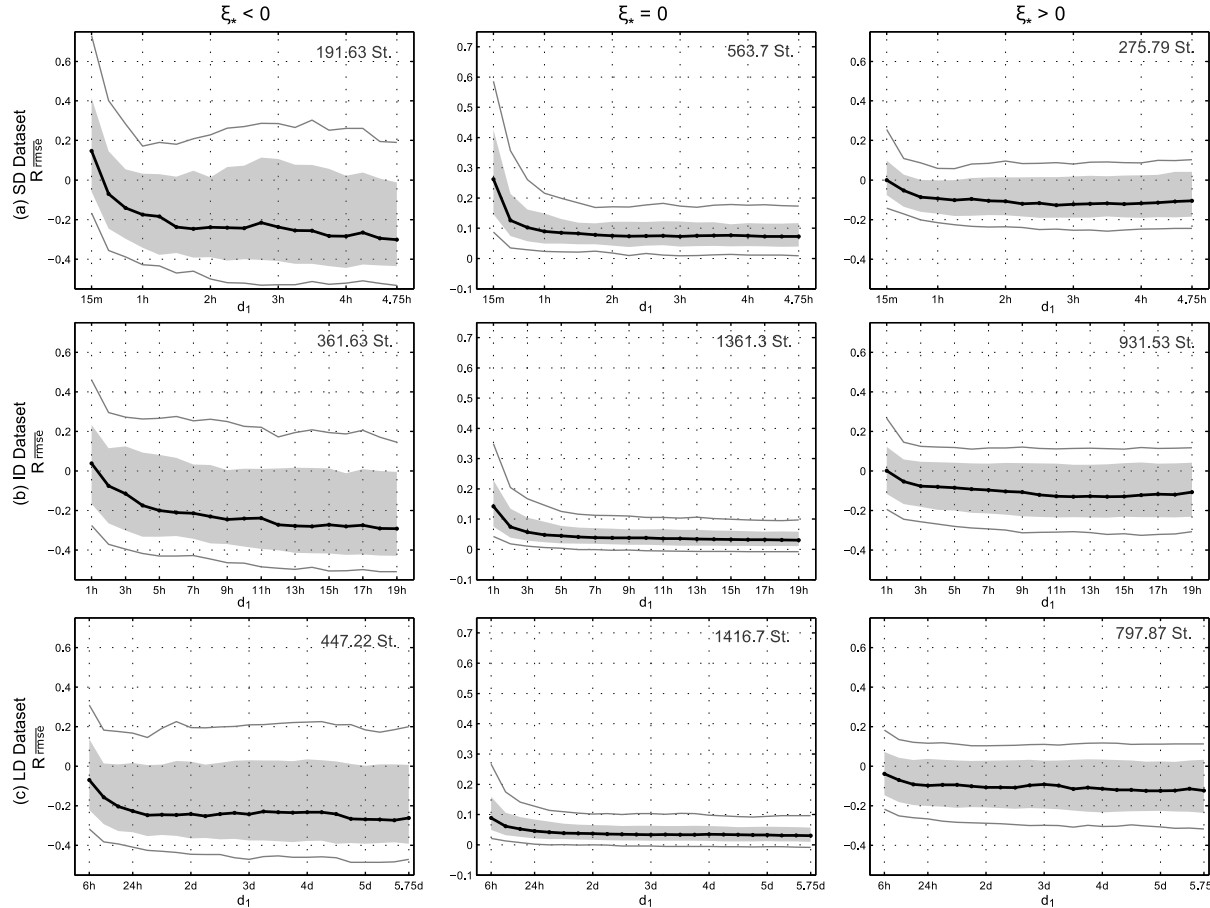

**Figure 12.** Distribution of the relative total RMSE ratio, $R_{\overline{rmse}}$, for $\xi_* < 0$ (first col.), $\xi_* = 0$ (second col.), and $\xi_* > 0$ (third col.) for 6-duration scaling intervals in SD (a), ID (b), and LD (c) datasets. The average number of valid SS station over the scaling intervals is indicated in the right-top corner of each graph.