# Peer review of "Simple Scaling of extreme precipitation in North America"

_Hydrology and Earth System Sciences, 2016_

## Short Comment (SC1) · 10 Dec 2016

I consider this paper as very interesting

At page 2, lines 23-24, where there is written "Theoretical and physical evidence of the scaling properties of precipitation intensity over a wide range of durations has been provided by several studies.", authors could add the following paper as a reference:

De Luca, D.L. (2014). Analysis and modelling of rainfall fields at different resolutions in southern Italy. Hydrological Sciences Journal, 59 (8), pp. 1536-1558. DOI: 10.1080/02626667.2014.926013

Best Regards

---

## Referee Comment (RC1) · Anonymous Referee #1 · 22 Dec 2016

Overall comments:

The article is overall well written, and the scientific subject is very topic and well addressed. The literature review and scientific context given in the introduction and Section 2 are very well written! The data description is also very clearly outlined. I had difficulties understanding (some technical details of) Section 4, and consequently (some of the results of) Sections 5 and 6. Therefore most of my comments aim to improve (my understanding) of this section. (I had also some perplexities about the chosen regions, which I address in the major comments too). I hope my comments lead to an improvement of the manuscript.

Major comments:

1) It is very difficult for me to comprehend Section 4 (mainly page 7 and Figures 1 and

2):

1a). the definition of scaling interval needs to be clearer: from page 7, lines 7-8, I understand that for each data-set (SD, ID and LD) you define several scaling intervals. These intervals have fixed durations equal to 6, 12,18 and 24 times the reference duration unit d* (which is 15', 1h and 6h, respectively, for each data-set SD, ID and LD). I assume the 6, 12, 18 and 24 durations are associated to a dilatation factor of $\lambda$ = 6, 12, 18, 24 (the same as in equations 2,3,4). The several scaling intervals you consider have all the same length (duration) but are distinct for their initial time. If this is correct, the text at line 11 (and thereafter) need to be revised: I suggest replacing "its first duration" with "its initial time", because using the word "duration" for both (initiation and duration) indices is very confusing.

1b). I am trying to understand how you estimate H: I understand that you use Eq. 4, with a fixed q and a fixed $\lambda$ (say, 6h). You have to regress however on more than one estimated moment, so I assume you consider the aforementioned scaling intervals with the different initial times (e.g. for the ID data-set and 6h-duration you consider the 19 6h-long scaling intervals you show in Figure 1b, left panel). For each (of the 19) initial times, you select the annual max, and then an AMS, from which you compute the q-moment (so you have 19 q-moments for the LHS of equation 4). What do you use for the first term of the RHS of Equation 4? will you have 6 X 19 q-moments corresponding to the AMS of 1h accumulated precipitation within each 6h long scaling interval?

1c). Page 7, line 16 "and the corresponding durations" are different durations, or different initial times for a fixed duration $\lambda$?

1d). I am not sure I understand what are you testing with the slope test. My hypothesis is that you have evaluated the regression coefficient Kq = -Hq log( $\lambda$ ) (the equality is from equation 4). Since the final goal is to estimate H, you want to regress the Kq versus q (for all the q=0.2,0.4, .. ,3), with a fixed $\lambda$, to finally find H. Please explain this better.

1e). GOF test: Page 7, line 23, "for each duration d": this time seems to refer to a real duration. Whereas four lines later "for all durations dj in the scaling interval" suggests dj are initial times. I do not understand what are you testing with the GOF test: what is the "pooled sample of the rescaled AMS x'_dj for all durations dj in the scaling interval"? I got completely lost in Eq. 8 and 9 . . . (I came back to this page few times, in separate days, to make sure my lack of understanding was not due to a particular bad moment. This is why I am describing in details what I do not understand, I hope this helps to point out what needs to be rephrased).

1f). Page 8, lines 10-11: here duration seems again to refer to a time duration. So, in my understanding, you consider 1h, 2h, 4h, 5h and 6h; from these you evaluate H; then you evaluate the 3h AMS with the SS. Then you evaluate the RMSE for the high quantiles of this synthetic 3h AMS versus the high quantiles of the empirical 3h AMS. I am not sure this is correct, but this is my guess . . ..

I think it would be very useful if you could show a figure with one (or maybe two) concrete example of your regression, for a fixed q and $\lambda$, so that the reader can better understand what youu regress, and what are the statistical tests you perform to be confident in your results.

2) When you perform your analysis, in my understanding, you consider scaling intervals of a duration of 6h (as an example) spanning the whole diurnal cycle as intervals with equivalent statistical properties. However, physically, precipitation occurring in the afternoon (which in the summer is triggered by convection) can present very different temporal structure (and physical properties) than precipitation occurring early in the morning: can you pull together these data, or (given that your focus is on extremes, which in summer often is related to convective events), should you consider a stratification based on the diurnal cycle? (The inhomogeneity of your phenomena along the dyurnal cycle might be also the cause of test rejections for longer scaling intervals, as you state at page 8, line 28-29)

3) Figure 1: your lowest confidence (largest proportion of rejected stations) is associated at the "durations" in the beginning of the scaling interval: why? This seems a sampling problem (is this what you "expect" in your statement at page 8, lines 26-27)? All the possible causes you list from page 8, line 29 to page 9 line 6 are plausible, but should hold also for 15m durations within the scaling interval (not just in the beginning). Possibly, my lack of understanding is linked to my lack of understanding on how the calculation is performed (page 7).

4) Figure 2 (and related text, at page 9, lines 15-21): your largest relative errors are always associated at the durations in the beginning and at the end of the scaling interval: similarly to Figure 1, is it possible that this is a sampling problem? (Or maybe, for longer time scaling, the inhomogeneity of the weather phenomena along the diurnal cycle plays a role .. )

5) Figure 3: I find it very difficult to understand the results shown in Figure 3:

5a) It is not well set what is the scope of this section is (in fact, at my first reading, I had the feeling it was a aimless technical analysis ... which instead is not). Later, I came to this hypothesis: in my understanding, H should be a scale-invariant parameter. Therefore large changes in H (aka large $\Delta$H) are "bad", whereas if $\Delta$H is near zero the SS model is a good approximation. Is this correct? Can you please state this clearly in the beginning of this section. Then, the reader will be able to search for the wanted results while analyzing Figure 3.

5b) Technical question: (the distribution of) H is computed for each duration (e.g. 1h, 2h, 3h, . . . in Figure 3 i-b). From Equations 2,3,4 I understood that H is scale invariant; then there should be one H which enables to describe all time scales. Why this is not the case? (Similar for the following sections, e.g Figure 8). Again, it is possible that my lack of understanding is linked to my lack of understanding on how the calculation of H is performed (page 7).

5c) I suggest as new title for Section 4.2: "Variability of the estimated scaling exponents" (to mirror the results shown in Figure 3, which shows ∆H).

6) Section 5: from the maps you show in Figures 4 and 5, the only two clearly distinct homogeneous regions (at a first eye analysis) seem to be SW_pacif and NW_pacif. I can see (the signal is more mild) also the regions C and E. It seems to me that the South-East of the United Stated could also be split in two regions (e.g. from Fig 4 ID and LD, and figure 5 ID). The Boreal region (as you conlude yourself at page 12, line 16) is very etherogeneous, and I do not see any reason for clustering these stations together (sole common factor is the network sparseness . . . . ). Have you attempted a cluster analysis to define your own regions, rather than considering the Bukovsky regions?

7) Figure 7 and S4 are needed -in my view- solely for supporting the description of Fig.8d (page 12, lines 26-31). (The results related to the other panels are less interesting, inmy view). I suggest to move also Figure 7 in the supporting material, along with the text at page 11, lines 11-22.

Minor comments:

1) There is a typo at line 7 of the abstract: should be 15', and not 15h.

2) page 4, equation 2: I suggest to eliminate "D" and explicitely write "λd" instead (the less symbols you introduce, the more readable is the article).

3) page 5, line 1: eliminate "and the frequency . . . F(x)".

4) page 5, line 9: I suggest writing "Approaches aimed at increasing the sample size may be used . . . "

5) page 5, Equation 7: notation is too complex, and should be simplified.

6) Page 6, lines 6-9: the cause-effect is not entirely clear to me: why two different periods where chosen (JJAS for the north and MJJASO for the south), rather the same period (either JJAS or MJJASO) for all stations?

7) Page 6, line 13-14 (and thereafter): rather than saying "coarser resolution" I suggest writing "more discretized recording procedure" (or something similar). The effect of recording discretizations are a well known problem in statistics, which can be bypassed simply by adding a uniformly distributed random noise (ranging between 0 and 2.54) to your data.

8) Page 6, line 30: this condition is not clear to me, please explain it more explicitely.

9) Figures 1 and 2: I suggest using a notation as 2h30', rather than 2.5h. Similarly for the days, 2d12h (or 50h) rather than 2.5d.

10) page 7, line 10-12: "This procedure . . . evaluate the variability of the SS estimates . . . " could it be the sensitivity instead? This text is not clear.

11) page 7, line 20: student t-test (the t is associated to the test, not to the student).

12) page 8, lines 14-20: r is usually used ijn statistics for correlation: it is possible to use a different notation? Please specify that the normalized RMSE is zero for a good fit, and it gets larger and larger for a worse fit.

13) Page 9, line 23: the acronym for inter-quartile range is, traditionally, IQR.

14) Figures 7,8 would be more easy to read if you put a title on each panel with the name of the region.

15) Page 11 lines 6-9: join this paragraph to the previous (they both pertain to the physical explanation of the different panels of Figure 8).

16) Page 12, lines 2-3: this apply to the SW_pac region as well.

17) Page 13, lines 10-12: I agree that scaling regimes are weaker for short d1 than for longer d1, however you need to rephrase the sentence at line 12. In fact, despite "smaller", the scaling regimes for short duration exceed 0.5 (most of them 0.6). Therefore effect of the scaling factor ($\lambda^{-H}$) is not negligible on the AMS distribution moments.

---

## Referee Comment (RC2) · J. Blanchet (Referee) · 23 Dec 2016

OVERALL COMMENTS

The article is well written and mainly clear. There is a substantial amount of work and many interesting results. However

1) Although Sections 1 and 2 are very clear, I had at first reading some difficulties understanding the rest of the paper, mainly because I got confused with the concepts of "d1" and "interval length". For example, if I understood correctly, an interval length of 6 durations with d1=1h for SD corresponds to durations 1h, 1h15, ..., 2h30, whereas an interval length of 6 durations with d1=1h for ID corresponds to durations 1h, 2h, ..., 6h. This may be confusing, so the authors may want to clarify these concepts, maybe giving examples or a table with the different intervals.

2) The authors use databases with different measurement frequencies. So I expect, e.g., the 1h-annual maxima at a given location to be larger when they stem from accumulating 15min rainfall than hourly rainfall. Thus I'm concerned about all the comparisons mixing these different measurement frequencies: do we expect H for example to be the same for different measurement frequencies? Likewise for the GEV parameters. In a pretty related study, Blanchet et al 2016 addresses this issue.

DETAILED COMMENTS

- p.3 l.5 "a deeper analysis ... needed": Blanchet et al 2016 make such a regional analysis in South of France. The study region is much smaller but rainfall variability seems quite comparable.

- p.4 l.11 "Hintensity and Hdepth": not defined

- section 2.2: Blanchet et al. 2016 use a GEV-ML estimation in a single step.

- section 3: it may be clearer for the reader to call the 4 databases 15PD, H1PD, H2PD and DPD.

- p.6 l.20: so if I understand correctly SD comprises stations from 15PD only, ID from both 15PD and HPD, and LD from both 15PD, 1HPD and DPD (DPD only for the duration intervals >=1day). I'm correct? The authors may want to clarify it. In which case, the authors are analysing annual maxima with different measurement frequencies, without taking this at all into account. I wonder how the results/parameters you're comparing later are really comparable.

- p.6 l. 23-26: Papalexiou and Koutsoyannis 2013 and Blanchet et al. 2016 consider also the rank of the observed maxima to decide whether they should consider it or not in the analysis.

- p.8 l. 8: I don't understand what are the "SS" and "non-SS" samples - Figures 1 and 2: it took me time to understand these figures, partly because the x-axis are not labeled. Please add the labels (d1?).

- Figure 3: isn't there also an effect of measurement frequency in the plots for ID and LD?

- section 6: do I understand correctly that "non-SS" cases mean that the GEV parameters are estimated using the data from d* only? Please make it clearer.

- Figure 4: it might be clearer for comparison to use the same US map for the three rows (the first row is different so far). Also there might be here an effect of the measurement frequency for LD and ID, although the spatial patterns are pretty coherent.

- Figure 5: same as Fig. 4.

- p.13 l.26: So if I understand correctly, here you use the H estimated previously and estimation is just for the GEV parameters. Please make it clearer. Have you also tried to estimate all parameters at once (mu*, sigma*, xi*, H) with ML estimators as in Blanchet et al 2016 for example? Theoretically, this should reduce the bias.

- Figure 9: isn't there also an effect of measurement frequency in the plots for ID and LD?

- Figure 10: idem

- Figure 11: please add in the legend "with Hosking test at level 5%"

References

J. Blanchet, D. Ceresetti, G. Molinié, J.-D. Creutin, A regional GEV scale-invariant framework for Intensity–Duration–Frequency analysis. Journal of Hydrology, Volume 540, September 2016, Pages 82–95.

Papalexiou, S.M., Koutsoyiannis, D., 2013. Battle of extreme value distributions: a global survey on extreme daily rainfall. Water Resour. Res. 49 (1), 187–201.

---

## Author Comment (AC1) · 21 Jan 2017

Dear Reviewer,

Thank you or your useful revision and remarks. Please find enclosed the reply to your interactive comment on the manuscript **"Simple Scaling of Extreme Precipitation in North America"** published in HESSD. We provide below a detailed response to each of your comments (which are reported in blue in the following text) and a copy of the revised manuscript in "track changes" mode in which specific colors are used to link corrections and reviewers' comments. More specifically, in the "track changes" manuscript,

- $\text{blue}^{R1}$ is used to underline changes related to your comments,

- while $\text{red}^{R2}$ is used for changes related to the second referee's comments [see the HESS interactive discussion webpage],

- and green is used for other changes.

Line numbering **(in bold)** refers to the revised manuscript with "track changes" attached to this reply.

Sincerely,

Silvia Innocenti, on behalf of the co-authors.

**Authors' detailed response to $1^{st}$ referee's comments**

5    The article is overall well written, and the scientific subject is very topic and well addressed. The literature review and scientific context given in the introduction and Section 2 are very well written! The data description is also very clearly outlined. I had difficulties understanding (some technical details of) Section 4, and consequently (some of the results of) Sections 5 and 6. Therefore most of my comments aim to improve (my understanding) of this section. (I had also some perplexities about the chosen regions, which I address in the major comments too). I hope my comments lead to an improvement of the manuscript.

10    Major comments:

     1)  It is very difficult for me to comprehend Section 4 (mainly page 7 and Figures 1 and 2)

        1a)  The definition of scaling interval needs to be clearer: from page 7, lines 7-8, I understand that for each data-set (SD, ID and LD) you define several scaling intervals. These intervals have fixed durations equal to 6, 12,18 and 24 times the reference duration unit d* (which is 15', 1h and 6h, respectively, for each data-set SD, ID and LD). I assume the

15          6, 12, 18 and 24 durations are associated to a dilatation factor of = 6, 12, 18, 24 (the same as in equations 2,3,4). The several scaling intervals you consider have all the same length (duration) but are distinct for their initial time. If this is correct, the text at line 11 (and thereafter) need to be revised: I suggest replacing "its first duration" with "its initial time", because using the word "duration" for both (initiation and duration) indices is very confusing.

        We agree that the definition of scaling interval needed to be clarified, probably because the concept of reference

20         duration $d^*$, initial duration $d_1$, scale ratio $\lambda$, and interval length were confusing [see also comment 1) of J. Blanchet, second referee].

        For all theoretical developments presented in Sect. 1, 2, and 4, we considered the reference duration $d^* = 1$ [defined at **Line 27, Page 5**] to express the scale ratio $\lambda$ [i.e. the ratio between two durations defined at **Line 20, Page 3**]. Choosing $d_* = 1$, the scale ratio $\lambda = d/d^*$ can be simplified to $\lambda = d$ [as stated at **Line 27, Page 5** in the original

25         version of the paper, and also at **Lines 5 to 11, Page 4** in the revised manuscript]. Hence, we can express the Moment Scaling Analysis (MSA) regression coefficients and the GEV parameters as functions of $d$ only [see Eq.(4), Eq. (7), and reply to comment 1b)].

        In Sect. 4, for all the SD, ID, and LD datasets and all scaling intervals, $d^* = 1 \, h$ has then been used as reference duration for defining the SS samples $x_{d*}$ [i.e. the samples of all AMS observation rescaled at $1h$ using Eq. (8)].

30         To clarify the definition of $x_{d*}$, **Lines 17 to 12, Page 8** have been rephrased [please refer to our reply to comment 1e)].

        In Sect. 6, $d^* = 1 \, h$ has also been used as the reference duration for estimating the SS-GEV parameters $\mu_*$, $\sigma_*$, and $\xi_*$ from the SS sample $x_{d*}$. This has been described at **Lines 24 to 27, Page 15**. However, without loss of generality, one could have been chosen $d_* \neq 1h$ with the only effect of rescaling the SS-GEV parameter values

35         and without affecting the shapes of the estimated distributions. For this reason, while Fig. 8 (a) and (b) present the distribution of the SS-GEV parameters for $d^* = 1h$ for the SD and ID dataset, in Fig. 8 (d) we rescaled $\mu_*$, $\sigma_*$, and $\xi_*$ at $d_* = 24h$ for comparing SS- and Non-SS-GEV estimates, as described at **Line 10, Page 16**.

        The definition of the scaling intervals, on the contrary, does not depend on $\lambda$ nor $d^*$. Each scaling interval is defined by the following three characteristic:

40          1. *The initial duration* $d_1$, corresponding to the first/smallest duration included in the scaling interval [see definition at **Line 5, Page 4** and the examples in Fig. 1 d)]; note that $d_1$ is effectively a "duration" [as in AMS definition] and not an "initial time" [which sounds more like the time at which the interval begins]. Hence, the terminology "first duration" has been kept.

2. *The time-step* used to construct each dataset [i.e., time-increment between successive durations for which we constructed AMS]: 15min for the DS dataset, 1h for ID, and 6h for LD, respectively [see **Line 33, Page 6** and Table 2)].

3. *The interval length*, i.e. the number of consecutive durations [either 6, 12, 18 or 24 ] included in the scaling interval $[d_1, d_D]$ [see **Line 28, Page 7**]: in other words, the scaling intervals do not all have the same length, as you suggested. For this reason we used the terminology "interval length", instead of "interval duration", which could be confusing.

In order to clarify the terminology and notations, the following modifications have been made in Sect. 2 and in the definition of the scaling intervals at page 7:

- We eliminated the notation $D = \lambda d$, keeping $\lambda d$ only, as you also suggested in minor comment 2) [**Line 20, Page 3** and following paragraphs]. Accordingly we also modified Eq. (2)-(4).

- We modified the discussion of Eq. (4) to [**Lines 5 to 11, Page 4**]:

  *"Moreover, without loss of generality, $\lambda$ can always be expressed as the scale ratio $\lambda = d/d^*$ defined for a reference duration $d^*$ chosen, for simplicity, as $d^* = 1$. Therefore, the SS model can be estimated and validated over a set of durations $d_1 < d_2 < .. < d_D$ by simply checking the linearity in a log-log plot of the $X$ moments versus the observed durations $d_j$, $j = 1, 2, \ldots, D$ [see, for instance, Gupta and Waymire, 1990; Burlando and Rosso, 1996; Fig. 1 of Nhat et al, 2007; and Fig. 2 (a) of Panthou et al., 2014]. If $H$ estimated for the first moment equals the exponents (slopes) for the other moments, the precipitation intensity $X$ can be considered scale invariant under SS in the interval of durations $d_1$ to $d_D$."*

- We rearranged the definition of the scaling intervals by explicitly mentioning that $d_1$ corresponds to the first duration of the 6, 12, 18, or 24 durations included within each scaling interval, by improving the definition of "interval length", and by adding other examples of scaling intervals [**Line 27, Page 7 to Line 5, Page 8**]:

  *"In order to identify possible changes in the SS properties of AMS distributions, various scaling intervals were defined for the MSA. In particular, all possible subsets with 6, 12, 18 and 24 contiguous durations were considered within each dataset. Figure 1 and Figure 2 show the 136 scaling intervals thereby defined: 40 scaling intervals for SD and IS, and 56 scaling intervals for LD. For instance, the first matrix on the left of Fig. 1(a) presents the 6-duration scaling intervals 15min - 1.5h, 30min - 1.75h, …, 4.75h - 6h defined for the SD dataset [i.e. the 19 scaling intervals containing six contiguous durations defined with a 15min increment], while Fig. 1(d) shows an example of the first four 6-duration scaling intervals for the ID dataset [i.e., 1h-6h, 2h-7h, 3h-8h, and 4h-9h, containing six contiguous durations defined with an increment of 1h]. This procedure was defined in order to evaluate the sensitivity of the SS estimates to changes in the first duration $d_1$ of the scaling interval and in the interval length [i.e. the number of durations included in the scaling interval]."*

1b) I am trying to understand how you estimate H: I understand that you use Eq. 4, with a fixed q and a fixed (say, 6h). You have to regress however on more than one estimated moment, so I assume you consider the aforementioned scaling intervals with the different initial times (e.g. for the ID data-set and 6h-duration you consider the 19 6h-long scaling intervals you show in Figure 1b, left panel). For each (of the 19) initial times, you select the annual max, and then an AMS, from which you compute the q- moment (so you have 19 q-moments for the LHS of equation 4). What do you use for the first term of the RHS of Equation 4? will you have 6 × 19 q-moments corresponding to the AMS of 1h accumulated precipitation within each 6h long scaling interval?

We apologize for this lack of clarity but the estimation of the scaling exponent $H$ is not based on the methodology you described. In particular, each scaling interval does not represent a period of time but an interval of durations [see the reply to the previous comment] and the estimation of $H$ for one scaling interval is independent from the

estimation over other scaling intervals.

In fact, for each moment order $q$ and each scaling interval, the scaling exponent can be estimated through Eq. (4), by the use of a linear regression between $E[X_d^q]$ and the scale ratio $\lambda$ in a log-log plot [see **Lines 5 to 17, Page 4**]. Moreover, since $d_* = 1$ implies $\lambda = (d/d_*) = d$, as stated at **Line 27, Page 5**, the MSA regression can be computed for each scaling interval by simply using $\ln(d)$ as covariate for $\ln(E[X_d^q])$. In other words, for each $q$ we estimate the following linear regression model:

$$Y_j = \alpha + K_q Z_j$$

where $Y_j = \ln(E[X_{d_j}^q])$, $Z_j = \ln(d_j)$, and $j = 1, 2, \ldots, D$. In practice, the empirical $q-$moments $\left\langle X_{d_j}^q \right\rangle$ of $X_{d_1}, X_{d_2},$ $\ldots, X_{d_D}$ were used for defining $Y_j$. This is a standard procedure called Moment Scaling Analysis (MSA) which is used to estimate the scaling exponents $K_q$ (the linear regression slopes) for different moment orders, and, at the same time, to validate the linearity of the scaling exponents with $q$, i.e. to test if $K_q \approx Hq$ [see, for instance, Gupta and Waymire, 1990; Burlando and Rosso, 1996; Fig.1 of Nhat et al, 2007; and Fig. 2 (a) of Panthou et al, 2014].

In our application, we estimated the scaling exponents $K_q$ as the slope of the MSA regression for fifteen moment orders $q = 0.2, 0.4, \ldots, 2.8, 3$ as described at **Lines 6 to 11, Page 8**. Then we checked the linearity condition $K_q \approx Hq$ using the "slope test" [**Line 12, Page 8 to Line 16, Page 8**]: a second regression model, $K_q = \beta_0 + \beta_1 q$, was fitted between the fifteen estimated values $K_q$ and the moment orders $q$. Then, a Student's t-test was used to test the null hypothesis $\boldsymbol{H}_0: \beta_1 = K_1$, i.e to test if $\beta_1$ is equal to the scaling exponent estimated for $q = 1$ [i.e. for the mean of the AMS]. If $\boldsymbol{H}_0$ was not rejected at the significance level $\alpha = 0.05$, the SS assumption $K_q \approx Hq$ was considered appropriate for the specific scaling interval and the simple scaling exponent $H = K_1$ was retained.

To improve the description of the MSA procedure and the slope test, we modified **Lines 5 to 11, Page 4** [see reply to comment 1a)] and we added a few details from **Line 6, Page 8 to Line 16, Page 8**.

1c) Page 7, line 16 "and the corresponding durations" are different durations, or different initial times for a fixed duration ?

Please see our replies to previous comments: for each scaling interval including durations $[d_1, d_D]$, we constructed the AMS for durations $d_1, d_2, \ldots, d_D$ while $\lambda$ only represents the scale ratio $\lambda = d/d^*$ needed to express $X_d$ [i.e. the variable "extreme precipitation intensity observed over time interval of duration $d$"] as a function of $X_{d^*}$. Note also that, throughout the manuscript, the word "duration" always refers to the AMS temporal scale. We hope that the modifications made for addressing the previous comments have already clarified these points.

1d) I am not sure I understand what are you testing with the slope test. My hypothesis is that you have evaluated the regression coefficient $Kq = -Hq\log(\lambda)$ (the equality is from equation 4). Since the final goal is to estimate H, you want to regress the Kq versus q (for all the q=0.2,0.4, .. ,3), with a fixed $\lambda$, to finally find H. Please explain this better.

The slope test described at **Line 12, Page 8 to Line 16, Page 8** is used to validate/invalidate the SS hypothesis $K_q \approx Hq$; if the slope test did not reject the null hypothesis of linearity of the MSA regression slopes with $q$, $H = k_1$ was retained. Please also see our reply to comment 1b). However, modifications made in response to previous comments should have clarified these points.

1e) GOF test: Page 7, line 23, "for each duration d": this time seems to refer to a real duration. Whereas four lines later "for all durations dj in the scaling interval" suggests dj are initial times. I do not understand what are you testing with the GOF test: what is the "pooled sample of the rescaled AMS $x'_{d_j}$ for all durations dj in the scaling interval"?

I got completely lost in Eq. 8 and 9 ... (I came back to this page few times, in separate days, to make sure my lack of understanding was not due to a particular bad moment. This is why I am describing in details what I do not understand, I hope this helps to point out what needs to be rephrased).

Please, see replies to comments $1a)$ et $1c)$. Both expressions "for each duration d" and "for all durations $d_j$ in the scaling interval" were used to effectively refer to "durations" since AMS were constructed for each duration $d_1, d_2, \ldots, d_D$ included in a given scaling interval $[d_1, d_D]$ (identified by its first duration $d_1$).

Since the scale invariance property states that $X_{d^*} \stackrel{d}{=} d^H X_d$ [Eq. (2) with $\lambda = d/d^* = d$], it is possible to rescale $\boldsymbol{x}_{d_1}, \ldots, \boldsymbol{x}_{d_j}, \ldots, \boldsymbol{x}_{d_D}$ (each of these vectors representing the AMS observed for the duration $d_j$ included in the scaling interval $[d_1, d_D]$) to the reference $d^* = 1h$ by simply applying the rescaling factor $d^H$. Accordingly, if $D$ durations $d_j$, with $j = 1, 2, \ldots, D$, are included in the scaling interval $[d_1, d_D]$, the SS hypothesis implies that the rescaled AMS $\boldsymbol{x'}_{d_1}, \ldots, \boldsymbol{x'}_{d_j}, \ldots, \boldsymbol{x'}_{d_D}$ can be pooled in a single sample $\boldsymbol{x'}_{d^*}$, where:

- for each $d_j$, $\boldsymbol{x'}_{d_j}$ is defined according to Eq. (9) as the samples of observations $x_{d_j,1}, x_{d_j,2}, \ldots, x_{d_j,i}, \ldots x_{d_j,n}$ sampled at duration $d_j$ and rescaled to $d^*$ through the rescaling factor $d_j^H$:

$$\boldsymbol{x'}_{d_j} = \left( d_j^H x_{d_j,1}, d_j^H x_{d_j,2}, \ldots, d_j^H x_{d_j,i}, \ldots d_j^H x_{d_j,n} \right)$$

- and $\boldsymbol{x}_{d^*} = \left( \boldsymbol{x'}_{d_1}, \ldots, \boldsymbol{x'}_{d_j}, \ldots, \boldsymbol{x'}_{d_D} \right)$ [Eq. (8)] is the pooled sample of AMS for durations $d_1, d_2, \ldots, d_D$ rescaled to the reference duration $d^*$ [i.e., $\boldsymbol{x}_{d^*}$ contains $n \times D$ observations rescaled at $d^* = 1h$].

For this reason, the AD and KS tests have been applied to compare the cdf of $\boldsymbol{x}_{d,ss} = d^{-H} \boldsymbol{x}_{d^*}$ [i.e. the pooled sample $\boldsymbol{x}_{d^*}$ scaled back to the duration $d$] to the cdf of the observed $\boldsymbol{x}_d$ at its original scale $d$, as a second test for the validity of the SS hypothesis.

In order to clarify this point the paragraph describing GOF tests has been rewritten as [**Line 17, Page 8 to Line 12, Page 9**]:

" *Goodness-of-Fit (GOF) test: for each duration d, the goodness of fit of the $X_d$ distribution under SS was tested using the Anderson-Darling (AD) and the Kolmogorov-Smirnov (KS) tests. These tests aim at validating the appropriateness of the scale invariance property for approximating the $X_d$ cdf by the distribution of $X_{d,ss} = d^{-H} X_{d^*}$. To this end, the pooled sample*

$$\boldsymbol{x}_{d^*} = \left( \boldsymbol{x'}_{d_1}, \ldots, \boldsymbol{x'}_{d_j}, \ldots, \boldsymbol{x'}_{d_D} \right) \tag{8}$$

*of the D rescaled AMS, $\boldsymbol{x'}_{d_j}$, was used to define $X_{d^*}$ under the SS assumption, considering all the durations $d_j$, with $j = 1, \ldots, D$, in the scaling interval. Each rescaled sample $\boldsymbol{x'}_{d_j}$ of the annual maxima $x_{d_j,i}$, $i = 1, \ldots, n$, observed for the duration $d_j$ was obtained by simply inverting Eq. (2) for $d^*$ :*

$$\boldsymbol{x'}_{d_j} = \left( d_j^H x_{d_j,1}, d_j^H x_{d_j,2}, \ldots, d_j^H x_{d_j,i}, \ldots d_j^H x_{d_j,n} \right) \tag{9}$$

*where $n$ represents the number of observations (years) available for each duration. In this way, $n \times D$ rescaled observations were included in $\boldsymbol{x}_{d^*}$.*
*As in previous applications (e.g., Panthou et al., 2014), the AD and KS tests were then applied at significance level $\alpha = 0.05$ to compare the empirical distributions (Cunnane plotting formula, Cunnane, 1973) of the SS and non-SS samples, $\boldsymbol{x}_{d,ss} = d^{-H} \boldsymbol{x}_{d^*}$ and $\boldsymbol{x}_d$.* "

1f) Page 8, lines 10-11: here duration seems again to refer to a time duration. So, in my understanding, you consider 1h, 2h, 4h, 5h and 6h; from these you evaluate H; then you evaluate the 3h AMS with the SS. Then you evaluate the RMSE for the high quantiles of this synthetic 3h AMS versus the high quantiles of the empirical 3h AMS. I am

*not sure this is correct, but this is my guess .*

Yes, the cross-validations was constructed as described in your example.

*I think it would be very useful if you could show a figure with one (or maybe two) concrete example of your regression, for a fixed q and , so that the reader can better understand what you regress, and what are the statistical tests you perform to be confident in your results.*

We added a reference to two graphical representations of the MSA procedure and SS equality $K_q \approx Hq$ at **Line 17, Page 4**:

*"... see, for instance, Gupta and Waymire (1990), Burlando and Rosso (1996); Fig. 1 of Nhat et al, 2007; and Fig. 2 (a) of Panthou et al., 2014 ".*

However, considering that the MSA is the standard tool for estimating and validating the SS models, we decided not to increase the number of figures in the paper.

- Nhat, L. M., Y. Tachikawa, T. Sayama, and K. Takara (2007), *A simple scaling charateristics of rainfall in time and space to derive intensity duration frequency relationships*, Ann. J. Hydraul. Eng, 51, 73–78.
- Panthou, G., T. Vischel, T. Lebel, G. Quantin, and G. Molini (2014), *Characterising the spacetime structure of rainfall in the Sahel with a view to estimating IDAF curves*. Hydrol. Earth Syst. Sci., 18(12), 5093–5107, doi:10.5194/hess-18-5093-2014.

2) *When you perform your analysis, in my understanding, you consider scaling intervals of a duration of 6h (as an example) spanning the whole diurnal cycle as intervals with equivalent statistical properties. However, physically, precipitation occurring in the afternoon (which in the summer is triggered by convection) can present very different temporal structure (and physical properties) than precipitation occurring early in the morning: can you pull together these data, or (given that your focus is on extremes, which in summer often is related to convective events), should you consider a stratification based on the diurnal cycle? (The inhomogeneity of your phenomena along the dyurnal cycle might be also the cause of test rejections for longer scaling intervals, as you state at page 8, line 28-29)*

We agree that the temporal structure of the observed precipitation events can be highly affected by the time of the day at which the events occur. This could be the case, for instance, for regions and time of the year when convection is the main generating process of extreme rainfall events. Accordingly, this will also be more important for short duration extreme events (e.g. duration less than an hour) for which convection is the main driver. In this regard, the analysis of the statistical characteristics of the temporal processes of precipitation should at some point consider the diurnal and seasonal precipitations cycles. However, in order to analyze the impact of these cycles on extreme precipitation intensity, one should consider methods and datasets which are different than the ones considered in our study. For instance one would have to deal with the time of the year and the time of the day at which the precipitation events occurred, as you suggested. Our study focuses, instead, on Annual Maxima Series (AMS) that do not consider the time at which the precipitation occurred but only the temporal scale (duration) over which the precipitation has been observed. This correspond to the classical definition of AMS typically used for constructing IDF curves and for scaling analysis [e.g., *Burlando and Rosso, 1996; Koutsoyiannis et al., 1998; CSA, 2012; Panthou et al., 2014*]. Even if it would be interesting to analyze the occurring time and other characteristics of the events from which annual maxima have been extracted [Sect. 5 of our study, for instance, addresses some of these issues], our extreme precipitation analysis did not considered the event-based definition of extremes needed to analyze the diurnal and seasonal cycles of precipitation.

For these reasons, in our opinion it is not possible to connect the higher proportion of GOF test rejections for longer scaling intervals [i.e., scaling interval considering an higher number of durations] to the diurnal cycle of rainfall, as you suggested. Instead, we can affirm that, for longer scaling intervals, there exists an *" higher probability of observing large differences between $x_d$ and $x_{d,ss}$ quantiles when $x_{d,ss}$ had larger sample size and included data from more distant durations. "* [see **Line 12, Page 10**;].

- Koutsoyiannis, D., D. Kozonis, and A. Manetas (1998), A mathematical framework for studying rainfall intensity-duration-frequency relationships, Journal of Hydrology, 206(1-2), 118–135, doi:10.1016/S0022-1694(98)00097-3.
- Burlando, P., and R. Rosso (1996), Scaling and multiscaling models of DDF for storm precipitations, Journal of Hydrology, 187, 45–64.
- CSA (2012), Technical guide: Development, interpretation and use of rainfall intensity- duration-frequency (IDF) information: Guideline for Canadian water resources practi- tioners, Tech. Rep. PLUS 4013 - 2nd ed.
- Panthou, G., T. Vischel, T. Lebel, G. Quantin, and G. Molini (2014), *Characterising the spacetime structure of rainfall in the Sahel with a view to estimating IDAF curves*. Hydrol. Earth Syst. Sci., 18(12), 5093–5107, doi:10.5194/hess-18-5093-2014.

3) Figure 1: your lowest confidence (largest proportion of rejected stations) is associated at the "durations" in the beginning of the scaling interval: why? This seems a sampling problem (is this what you "expect" in your statement at page 8, lines 26-27)? All the possible causes you list from page 8, line 29 to page 9 line 6 are plausible, but should hold also for 15m durations within the scaling interval (not just in the begin- ning). Possibly, my lack of understanding is linked to my lack of understanding on how the calculation is performed (page 7).

We agree that largest proportions of rejected stations occurring for the shortest durations of the scaling intervals can be partially explained by a sampling effect. In particular, as you mentioned, our interpretation of this result underlined three possible causes [**Line 14, Page 10 to Line 23, Page 10**] :

1. GOF tests could be more often rejected due to a relatively more important presence of very large values in short-duration samples, i.e. a difference in statistical features of $X_d$ between short and long durations that makes shorter durations more prone to rejection of GOFs.

2. When considering durations close to the temporal resolution of the recorded series [which is,generally 15 min ou 1 h], the measure of precipitation may be underestimated because intense rainfall events may be more likely split between two consecutive time steps. Obviously, this underestimation affect shortest durations more than longer durations.

3. Largest GOF test rejections for shortest series seem also to be connected to the coarser measurement resolution of 15PD series, which, similarly to the temporal resolution effect at point 2, induces larger measurement errors for shortest duration AMS.

These explanations were probably unclear due to the confusing definition of the scaling intervals reported in the previous version of the paper. The corrections made to sections 4 [the improvement of the paragraphs describing the construction of the scaling intervals and the scaling exponent estimation] should have now clarified this point.

4) Figure 2 (and related text, at page 9, lines 15-21): your largest relative errors are always associated at the durations in the beginning and at the end of the scaling interval: similarly to Figure 1, is it possible that this is a sampling problem? (Or maybe, for longer time scaling, the inhomogeneity of the weather phenomena along the diurnal cycle plays a role .. )

Thanks for the interesting comment. To our opinion, this result is not really surprising considering that Fig. 2 shows the cross-validation estimations of the normalized RMSE averaged over all valid SS stations. In a cross-validation setting, we expect that SS estimations are more impacted by the exclusion of a duration at the border of a scaling interval than by the exclusion of an inner duration.

Consider, for instance, the cross-validation estimations on the 6-duration scaling interval 1h-6h for the ID dataset. One would expect a greater change in SS estimation when the duration 6h is excluded than when an inner duration, e.g. 3h, is excluded. This is related to two factors:

i) an OLS linear regression is used for estimating $H$: excluding the last point of the regression (i.e. the last duration of the scaling interval) may have a greater impact on the slope estimation than excluding an inner point.

ii) the SS sample $x_{d,ss}$ pools and rescales observations coming from AMS observed over various durations: approximating the $X_{3h}$ distribution with rescaled observations from both durations $\leq 3h$ and $\geq 3h$ uses, in average, more information than approximating $X_{6h}$ with rescaled observations coming from durations $\leq 6h$ only .

Moreover, as stated at **Line 4, Page 11**, the quality of the $SS$ approximation seems also to deteriorate with decreasing $d_1$ and with increasing scaling interval length due to an effective decrease of the model performances.

In order to complete the discussion of Fig. 2, we added a reference to the greater sensitivity of SS estimation to the exclusion of durations at the border of the scaling interval [**Lines 34 to 4, Page 10**]:

*"Larger errors were observed for durations at the border of the scaling intervals. Not surprisingly, this result underlines that, in a cross-validation setting, both the MSA estimation of H and the $X_{d,ss}$ approximation are less sensitive to the exclusion of an inner duration of the scaling interval than to the exclusion of $d_1$ or $d_D$. Conversely, the extrapolation under SS of the $X_d$ distribution is generally less accurate if $d$ is outside the range of durations used to estimate H. ".*

5) Figure 3: I find it very difficult to understand the results shown in Figure 3:

5a) It is not well set what is the scope of this section is (in fact, at my first reading, I had the feeling it was a aimless technical analysis ... which instead is not). Later, I came to this hypothesis: in my understanding, H should be a scale-invariant parameter. Therefore large changes in H (aka large ΔH) are "bad", whereas if ΔH is near zero the SS model is a good approximation. Is this correct? Can you please state this clearly in the beginning of this section. Then, the reader will be able to search for the wanted results while analyzing Figure 3.

In order to clarify the topic of the section and the aim of our analysis we added the following introductory paragraph at **Lines 10 to 12, Page 11**:

*" In order to evaluate the sensitivity of SS to the considered scaling interval, the variability of H with $d_1$ has been analyzed. Then, the spatial distribution of the scaling exponents for each scaling interval was studied to assess the uncertainty in H estimation and the dependence of SS exponents on local geoclimatic characteristics."*

Then, we added the following explanations to the definition of $\Delta_H$:

- *" Investigating the variability of the scaling exponent with the scaling interval is particularly important since, if SS is assumed to be valid between some range of durations, one should expect that H remains almost unchanged over the various scaling intervals included in this range. For this reason, the variation $\Delta_{H_{(j)}}$ of the scaling exponents computed for overlapping scaling intervals having the same $d_1$ but different lengths was analyzed. "* [**Lines 13 to 16, Page 11**]

- *"If SS is appropriate over a range of durations, $\Delta_{H_{(j)}}$ is expected to be small for scaling intervals defined within this range. "* [**Line 24, Page 11**]

5b) Technical question: (the distribution of) H is computed for each duration (e.g. 1h, 2h, 3h, ... in Figure 3 i-b). From Equations 2,3,4 I understood that H is scale invariant; then there should be one H which enables to describe all time scales. Why this is not the case? (Similar for the following sections, e.g Figure 8). Again, it is possible that my lack of understanding is linked to my lack of understanding on how the calculation of H is performed (page 7).

The distribution of $H$ over stations is presented in Fig 3(i) for each $d_1$, i.e. for each scaling interval having $d_1$ as first duration [please, see replies to comments 1a) and 1b)]. In fact, only one value of $H$ is estimated for all durations included in a scaling interval [as you suggested, H enables to describe all time scales with only one rescaled distribution], and this is not in contradiction with what is showed in Fig. 3. The misunderstanding is due to the fact that $d_1$ is used to identify a scaling interval and not any general duration $d$: once the dataset and the scaling interval length are fixed, the first duration $d_1$ is used to identify the scaling interval that includes the durations $d_1, d_2, \ldots, d_D$.

To simplify result interpretation, we added the label "$d_1$" to all x-axis of Fig. 3, 8 [corresponding to Fig. 9 in the previous version], and 11 [Fig. 12 in the previous version].

5c) I suggest as new title for Section 4.2: "Variability of the estimated scaling exponents" (to mirror the results shown in Figure 3, which shows $\Delta$H).

The title of the subsection 4.3 has been modified to *"Estimated scaling exponents and their variability"*.

6) Section 5: from the maps you show in Figures 4 and 5, the only two clearly distinct homogeneous regions (at a first eye analysis) seem to be SW_pacif and NW_pacif. I can see (the signal is more mild) also the regions C and E. It seems to me that the South-East of the United Stated could also be split in two regions (e.g. from Fig 4 ID and LD, and figure 5 ID). The Boreal region (as you conlude yourself at page 12, line 16) is very etherogeneous, and I do not see any reason for clustering these stations together (sole common factor is the network sparseness .... ). Have you attempted a cluster analysis to define your own regions, rather than considering the Bukovsky regions?

We agree that for some scaling intervals (e.g., intervals showed in Fig. 5) many regions may appear quite heterogeneous. However, the regional analysis presented in Sect. 5 seems to confirm that, with the exception of A1 and B, the distribution of $H$ within the considered regions is concentrated about its mean value for most of the scaling intervals . For instance, it seems clear from Fig. 8 that the variability of $H$ in regions E and F is fairly low, while these regions contain the greater proportions of available stations. On the contrary, the heterogeneity observed in regions A1 and B could be probably connected to the low station density in these two large areas [as mentioned at **Line 9, Page 14**]. In this respect, we agree that a finer definition/separation of northern regions would improve region homogeneity.

However, our primary interest was to define a simple partition of the study area into regions with distinct climatic characteristics. In this sense, other climatological classifications could have also been considered. In other words, our analysis did not seek for a rigorous identification of "SS regions" but intended to perform a preliminary regional analysis based on the climatoilogical features that could influence SS regimes.

We agree that it would be interesting to apply methods, such as clustering, to further analyzed $H$ spatial distribution. More refined methods would be necessary in oder to asses the homogeneity of the *Bukosky* (or other) regions in terms of the scaling exponent values and to precisely describe which climate and meteorological pocesses drive the local distribution of the $H$. However, the intent of our analysis was, at this point, mainly descriptive and aimed at validating (or not) the hypothesis that basic climatological characteristics may influence the spatial distribution of $H$.

7) Figure 7 and S4 are needed -in my view- solely for supporting the description of Fig.8d (page 12, lines 26-31). (The results related to the other panels are less interesting, in my view). I suggest to move also Figure 7 in the supporting material, along with the text at page 11, lines 11-22.

As suggested, the figure and the details of calculations of $N_{eve}$ have been moved in the supplementary material: see **page 13**.

Minor comments

1) There is a typo at line 7 of the abstract: should be 15', and not 15h.

Corrected.

2) Page 4, equation 2: I suggest to eliminate "D" and explicitly write "λd" instead (the less symbols you introduce, the more readable is the article).

Correction made. Please see the reply to major comment 1a).

3) Page 5, line 1: eliminate "and the frequency ... F(x)".

Done.

4) Page 5, line 9: I suggest writing "Approaches aimed at increasing the sample size may be used ... "

Done.

5) Page 5, Equation 7: notation is too complex, and should be simplified.

10 Thank you for the suggestion but we think that no superfluous symbol is used in Eq. (7), which directly follow from Eq. (2) and the scale invariance property of the GEV expressed at **Line 25, Page 5**:

since $X \overset{d}{=} GEV(\mu, \sigma, \xi)$ implies $\lambda X \overset{d}{=} GEV(\lambda\mu, \lambda\sigma, \xi)$,

if one consider $X_{d*} \overset{d}{=} GEV(\mu_{d*}, \sigma_{d*}, \xi)$, and Eq.(2) applies, i.e. $X_d = \lambda^{-H} X_{d*} = d^{-H} X_{d*}$,

then $X_d \overset{d}{=} GEV(d^{-H}\mu_{d*}, d^{-H}\sigma_{d*}, \xi)$.

15 See also Blanchet et al (2016), Eq. (7), page 84 [J. Blanchet, D. Ceresetti, G. Molinie, J.-D. Creutin, A regional GEV scale-invariant framework for Intensity–Duration–Frequency analysis. Journal of Hydrology, Volume 540, September 2016, Pages 82–95.].

6) Page 6, lines 6-9: the cause-effect is not entirely clear to me: why two different periods where chosen (JJAS for the north and MJJASO for the south), rather the same period (either JJAS or MJJASO) for all stations?

For each station in the study region [i.e. no matter the latitude], a year was considered valid if it had minimally 85% valid values
20 (otherwise it was considered as a missing year). Then, only valid years were considered for constructing AMS. However, many Canadian stations -especially those equipped with tipping bucket rain gage- do not record precipitation during winter period, namely from November to April for stations located south of the 52nd Parallel N and from October to May for stations located north of the 52nd Parallel N (CSA, 2012). This means that, without restricting the definition of "years" to the annual period during which stations are in operation, we could have not use the records from many northern stations. At the same time, the 'block maxima' definition of extreme
25 precipitation series need long annual periods (i.e. large blocks) to be consistent with the GEV approximation of AMS distribution. Therefore, a trade-off between "year length" and "number of valid stations" existed. We therefore chose a definition of "year" (block) based on the latitude: for stations located north of the 52nd Parallel N we defined the year as the period from June to September (i.e. 122 days a year were considered), while for stations located south of the 52nd Parallel N we used the period from May to October (i.e. 184 days a year were considered). Note that the same criteria ware used for Canadian and US stations.

- CSA: *Development, interpretation and use of rainfall intensity-duration-frequency (IDF) information: Guideline for Canadian water resources practitioners*, Tech. Rep. Canadian Standard Association, Tech. Rep. PLUS 4013, Mississauga, Ontario, 2nd ed., http://shop.csa.ca/en/canada/infrastructure-and-public-works/plus-4013-2nd-ed-pub-2012/invt/27030802012, 2012.

7) Page 6, line 13-14 (and thereafter): rather than saying "coarser resolution" I suggest writing "more discretized recording procedure"
35 (or something similar). The effect of recording discretizations are a well known problem in statistics, which can be bypassed simply by adding a uniformly distributed random noise (ranging between 0 and 2.54) to your data.

Thank you for the suggestion but for consistence with the expression "temporal resolution" we kept the terminology "instrument resolution" for indicating the minimum amount of precipitation detectable from rainfall gauges [Note that these are two "resolutions" have similar impacts on our results]. For clarity, however, we changed *"resolution"* to *"instrument resolution"* when needed [e.g.,
40 **Lines 24 to 28, Page 6**, **Line 4, Page 18**, and **Table 1**].

Moreover we agree that several statistical methods exist for dealing with highly discretized data. Although we did not use such methods, we evaluate how this discretization may affect our results [some complementary analyses are presented in the supplementary material, Sect. S2 and Fig. S2-S3] and we considered it when applying our statistical analysis [e.g., when applying GOF test, see **Line 14, Page 9**]. However, we choose not to modify the raw data since it could possibly have, in our opinion, an unpredictable impact on our results [a non-quantifiable impact on AMS scaling].

8) Page 6, line 30: this condition is not clear to me, please explain it more explicitely.

From the definition of daily maxima [note that the DM dataset has been renamed Daily Maxima Precipiation Data (DMPD) according to minor comment 4 of the second reviewer; see **Line 9, Page 6**], if the intensity value $x_{d_1} \geq 0$ has been recorded at duration $d_1$ and the intensity value $x_{d2} \geq 0$ has been recorded at duration $d_2$, with $d_1 \leq d_2$, the following conditions must be met:

   i) The rainfall intensity $x_{d_1}$ observed for the shorter duration $d_1$ must be larger or equal to the intensity $x_{d_2}$ observed over the longer duration $d_2$ (equality occurs if rainfall intensity would be constant during a period of time $d_2$): $x_{d_1} \geq x_{d2}$, i.e. $0 \leq \frac{x_{d_2}}{x_{d1}} \leq 1$.

   ii) The rainfall depth $d_1 x_{d_1}$ recorded during the time period $d_1$ must be smaller or equal to the rainfall depth $d_2 x_{d_2}$ recorded during time period $d_2$ (equality occur if rainfall is recorded only during the time period $d_1$): $d_1 x_{d_1} \leq d_2 x_{d_2}$, i.e. $\frac{d_1}{d_2} \leq \frac{x_{d_2}}{x_{d_1}}$

Combining these two condition we have: $0 \leq \frac{d_1}{d_2} \leq \frac{x_{d_2}}{x_{d_1}} \leq 1$. Therefore, if maximum intensities recorded over the various durations within a given day do not satisfy this condition among pairs of durations, the values were considered "suspicious" and the day were discarded (i.e. all the daily maxima observed over the various durations for that day were discarded).

To clarify this issue, the paragraph has been rewritten as [**Lines 13 to 16, Page 7**]:

*"For instance, each pair of DMPD intensity $(x_{d_1}, x_{d_2})$ observed at durations $d_1 < d_2$ must respect the conditions $x_{d_2}/x_{d_1} \leq 1$ and $d_1 x_{d_1} \leq d_2 x_{d_2}$ derived from the definitions of daily maxima rainfall intensity and depth; otherwise all DMPD values recorded that day were discarded and assimilated to missing data. "*

9) Figures 1 and 2: I suggest using a notation as 2h30', rather than 2.5h. Similarly for the days, 2d12h (or 50h) rather than 2.5d.

Done. Note that a notation like "2h30min" instead of " 2h30' " has been used to respect HESS standard for units and figures.

10) Page 7, line 10-12: "This procedure ... evaluate the variability of the SS estimates... " could it be the sensitivity instead? This text is not clear.

The text has been changed integrating your suggestions [see also our reply to major comment 1a)]:

*"This procedure was defined in order to evaluate the sensitivity of the SS estimates to changes in the first duration $d_1$ of the scaling interval and in the interval length [i.e. the number of durations considered]."*

11) Page 7, line 20: student t-test (the t is associated to the test, not to the student).

Corrected.

12) Page 8, lines 14-20: r is usually used in statistics for correlation: it is possible to use a different notation? Please specify that the normalized RMSE is zero for a good fit, and it gets larger and larger for a worse fit.

The symbol $\epsilon$ has been substituted to $r$ in the whole text [see, for instance, Eq. (10), (11), and Fig. 2]. Moreover, we integrated your second suggestion adding the following note at **Line 4, Page 10** :

*"Note that the normalized RMSE is a measure of error, meaning that values of $\overline{\epsilon}_{x_{d,s}}$ closer to 0 correspond a better fit than larger*

*values. "*

13) Page 9, line 23: the acronym for inter-quartile range is, traditionally, IQR.

Modified.

14) Figures 7,8 would be more easy to read if you put a title on each panel with the name of the region.

Thank for the suggestion but the list of the *Bukosky* regions considered in each panel was too long to be added to the legend. For this reason we named the regions with names (A1), (A2), (B), ..., (F).

For clarity, we added "Region" in each panel and the following sentence to the legend of Fig 7 [corresponding to Fig. 8 of the original version of the paper]:

*"See Fig. 6 for region definition."*

15) Page 11 lines 6-9: join this paragraph to the previous (they both pertain to the physical explanation of the different panels of Figure 8).

Done.

16) Page 12, lines 2-3: this apply to the SWpac region as well.

We agree that the particular topography characterizing the pacific coast may also impact the results of the curves in Fig. 7 (d). However the different synoptic regime characterizing the south-west areas of the continent seems to be the most important factor differentiating the curves observed for the northern and southern parts of the west coast [according to $N_{eve}$ and $T_{wet}$ results]. To underline this point, the following comment has been added at **Line 30, Page 14** :

*"These results suggests that both the distinctive topography of the west coast and the characteristic large-scale circulation of the south-west areas of the continent are crucial factors determining the transition between the two scaling regimes in region D. "*

17) Page 13, lines 10-12: I agree that scaling regimes are weaker for short d1 than for longer d1, however you need to rephrase the sentence at line 12. In fact, de- spite "smaller", the scaling regimes for short duration exceed 0.5 (most of them 0.6). Therefore effect of the scaling factor $(\lambda^{\hat{}} - H)$ is not negligible on the AMS distribution moments.

The sentence has been simply rephrased to [**Line 11, Page 15**] :

[revised manuscript text omitted]

---

## Author Comment (AC3) · 21 Jan 2017

Dear Reviewer,

Thank you or your useful revision and remarks. Please find enclosed our reply to your interactive comment on the manuscript **"Simple Scaling of Extreme Precipitation in North America"** published in HESSD. We provide below a detailed response to each of your comments (which are reported in red in the following text) and a copy of the revised manuscript in "track changes" mode in which specific colours are used to link corrections to reviewers' comments. In particular, in the "track changes" manuscript,

- red$^{R2}$ is used to underline changes related to your comments,

- while blue$^{R1}$ is used for changes related to the first referee's comments [see the HESS interactive discussion web-page],

- and green is used for other changes.

Line numbering **(in bold)** refers to the revised manuscript with "track changes" attached to this reply.

Sincerely,

Silvia Innocenti, on behalf of the co-authors.

**Authors' detailed response to $2^{nd}$ referee's comments**

5    The article is well written and mainly clear. There is a substantial amount of work and many interesting results. However

1) Although Sections 1 and 2 are very clear, I had at first reading some difficulties understanding the rest of the paper, mainly because I got confused with the concepts of "d1" and "interval length". For example, if I understood correctly, an interval length of 6 durations with d1=1h for SD corresponds to durations 1h, 1h15, . . ., 2h30, whereas an interval length of 6 durations with d1=1h for ID corresponds to durations 1h, 2h, . . ., 6h. This may be confusing, so the authors

10    may want to clarify these concepts, maybe giving examples or a table with the different intervals.

We apologize for this lack of clarity, which was also pointed out by the first referee [see our reply to comment 1a) of the first reviewer]. To improve the description of the set of scaling intervals and their characteristic $d_1$ (initial duration ), length (number of durations considered), and time-step (the time-increment separating contiguous durations within each dataset, equal to 15min in SD, 1h in ID, and 6h in LD) we modified the paragraph describing these characteristics to

15    [**Lines 27 to 5, Page 7**]:

*"In order to identify possible changes in the SS properties of AMS distributions, various scaling intervals were defined for the MSA. In particular, all possible subsets with 6, 12, 18 and 24 contiguous durations were considered within each dataset. Figure 1 and Figure 2 show the 136 scaling intervals thereby defined: 40 scaling intervals for SD and IS, and 56 scaling intervals for LD. For instance, the first matrix on the left of Fig. 1(a) presents the 6-duration scaling intervals*

20    *15 min - 1.5 h, 30min - 1.75h, . . ., 4.75 h - 6 h defined for the SD dataset [i.e. the 19 scaling intervals containing six contiguous durations defined with a 15min increment], while Fig. 1(d) shows an example of the first four 6-duration scaling intervals for the ID dataset [i.e. 1 h - 6 h, 2 h - 7 h, 3 h - 8 h, and 4 h - 9 h, containing six contiguous durations defined with an increment of 1h]. This procedure was defined in order to evaluate the sensitivity of the SS estimates to changes in the first duration $d_1$ of the scaling interval and in the interval length [i.e. the number of durations included*

25    *in the scaling interval]. "*

2) The authors use databases with different measurement frequencies. So I expect, e.g., the 1h-annual maxima at a given location to be larger when they stem from accumulating 15min rainfall than hourly rainfall. Thus I'm concerned about all the comparisons mixing these different measurement frequencies: do we expect H for example to be the same for different measurement frequencies? Likewise for the GEV parameters. In a pretty related study, Blanchet et al 2016

30    addresses this issue.

We agree that the temporal resolution of observed series, as well as the measurement resolution of rain gauges, may have an impact on the estimations of the AMS and of the scaling exponents.

Note however, that for many stations both DMPD and HCPD, or both 15PD and HPD series are available over a common period [note also that the names of the 4 datasets have been changed according to your minor comment 4)]. For these

35    stations, annual maxima measured at the two different temporal resolutions were compared and combined [for each year, the annual maximum value of the two AMS was retained] In this way, the impact of the temporal resolution should have been partially reduced. [note that, to improve the description of this preliminary step in the construction of our AMS **Lines 17 to 20, Page 7** have been rephrased. Please, see our reply to minor comment 5)].

At the same time, the double resolution effects [temporal and measurement resolution effects which could add and have

40    a greater impact on shortest durations] may still affect the estimation of AMS Simple Scaling. In fact, some of these issues have been raised and briefly discussed while analysing GOF test results [see **Lines 17 to 23, Page 10**]. Some complementary analyses have also been presented in the supplementary material, Sect. S2 and Fig. S2-S3. [Note that, for completeness, we added the reference to *Blanchet et al (2016)* at **Line 23, Page 10**].

However, despite the obvious interest of studying the effects of these factors, it would have been difficult, with the available datasets, to separate the influence of the temporal and measurement resolutions on extreme precipitation inference from the effect of other factors, such as, sampling errors associated to the series length.

Moreover, pooling the four datasets allow the construction of a rainfall dataset with an exceptional extent and density for North America. Hence, we decided to use these four datasets to construct the three AMS dataset SD, ID, and LD which constitute a remarkable data source for the study of Simple Scaling of extreme precipitation at a regional scale.

For completeness, however, the following comment has been added to the Conclusion [**Line 6, Page 18**]:

*"... and (these results) show the importance of a deeper analysis to evaluate the impact of dataset characteristics (e.g., their temporal and measurement resolutions, or the series length) on the scale invariant properties of extreme precipitation."*

**Detailed Comments**

1 p.3 l.5 "a deeper analysis . . . needed": Blanchet et al 2016 make such a regional analysis in South of France. The study region is much smaller but rainfall variability seems quite comparable.

The reference has been added at **Line 4, Page 3**.

2 p.4 l.11 "$H_{intensity}$ and $H_{depth}$": not defined

The sentence has been rephrased in order to add the explicit definition of $H_{depth}$ [**Line 20, Page 4**]:

*"(note that for the rainfall depth the scaling exponent $H_{depth} = 1 - H$ applies)".*

3 section 2.2: Blanchet et al. 2016 use a GEV-ML estimation in a single step.

The following paragraph has been moved from Sect. 6 to **Line 5, Page 6** [Sect. 2] in order to clarify that a one-step procedure can also be used for the estimation of the SS-GEV parameters:

*"In a few other cases, a Generalized Additive Model ML (GAM-ML) framework (Coles, 2001; Katz, 2013) has also been used to obtain the joint estimate of $H, \mu_*, \sigma_*,$ and $\xi_*$ through the introduction of the duration as model covariate (e.g., Blanchet et al, 2016). "*

Note that this procedure has been tested in our preliminary analyses, as mentioned at **Lines 27 to 30, Page 15**. Please, see also our reply to minor comment 12.

4 section 3: it may be clearer for the reader to call the 4 databases 15PD, H1PD, H2PD and DPD.

As suggested, we homogenized the dataset acronyms changing DM to DMPD and H to HCPD [**Line 9, Page 6** and following paragraphs]. However, we kept the acronyms "HPD" and "15PD" for US datasets since these are the official acronyms used by the NOAA [http://www.ncdc.noaa.gov/data-access/land-based-station-data].

5 p.6 l.20: so if I understand correctly SD comprises stations from 15PD only, ID from both 15PD and HPD, and LD from both 15PD, 1HPD and DPD (DPD only for the du- ration intervals >=1day). I'm correct? The authors may want to clarify it. In which case, the authors are analysing annual maxima with different measurement frequencies, without taking this at all into account. I wonder how the results/parameters you're comparing later are really comparable.

We apologize for this lack of clarity but the stations included within each of the SD, ID, and LD dataset were selected according to a procedure slightly different from the one you mentioned. In particular, while it is correct that the SD dataset uses 15PD data only [**Line 6, Page 7**], both ID and LD datasets use all relevant series to construct AMS for the sampled durations [Please, see also Tables 1 and 2, and reply to major comment 2)]. To improve the description of dataset construction **Lines 17 to 20, Page 7** have been modified to:

*(iv) For each selected station, annual maxima were extracted for each valid year and duration. For stations having both DMPD and HCPD series, or 15PD and HPD series, for each year, the annual maxima extracted from these two series were compared and the maximum value was retained as the annual maximum for that year.*

Moreover, we agree that the inhomogeneity of the series temporal resolution should be taken into account when interpreting our results, as well as other factors such as the different series length, measurement resolution, etc [Please, refer to the reply to major comment 2)]. However, it is *a priori* difficult to asses and separate the impacts of each of these inhomogeneities on the SS estimation. In particular, in our opinion, it is difficult to rigorously relate one or some of these inhomogeneities to any specific feature observed for the $H$ distribution or GEV parameters. Hence, we assumed that they will not globally nor significantly affect our results nor the main conclusions of our study. For completeness, however, a brief comment about this issue has been added in the Conclusion [see **Line 6, Page 18** and major comment 2)].

6 p.6 l. 23-26: Papalexiou and Koutsoyannis 2013 and Blanchet et al. 2016 consider also the rank of the observed maxima to decide whether they should consider it or not in the analysis.

As in Papalexiou and Koutsoyiannis (2013), the following criterion was used: for each series, observation that are at least one order of magnitude larger than the series second largest value were excluded. This procedure was repeated until the ratio between the two largest values of the series was less than an order of amplitude. This detail, as well as your suggested reference, has been added at **Line 7, Page 7**:

*"Note that, in order to exclude outliers possibly associated with recording or measurement errors, extremely large observations were discarded and assimilated to missing data. In particular, as in some previous studies (e.g., Papalexiou and Koutsoyiannis, 2013; Papalexiou et al., 2013), an iterative procedure was applied prior to step (ii)-1) to discard observations larger than 10 times the second largest value of the series. "*

7 p.8 l. 8: I don't understand what are the "SS" and "non-SS" samples.

The definitions of SS and non-SS samples have been added at **Line 27, Page 8**:

*"As in previous applications (e.g., Panthou et al., 2014), the AD and KS tests were then applied at significance level $\alpha = 0.05$ to compare the empirical distributions (Cunnane plotting formula, Cunnane, 1973) of the SS sample, $\boldsymbol{x}_{d,ss} = d^{-H}\boldsymbol{x}_{d^*}$, and the non-SS sample, $\boldsymbol{x}_d$."*

Then, we rephrased the sentence at **Line 16, Page 9** as:

*"According to this approach, data in $\boldsymbol{x}_d$ and $\boldsymbol{x}_{d,ss}$ were pooled and randomly reassigned to two samples having same sizes of the SS and non-SS samples. "*

8 Figures 1 and 2: it took me time to understand these figures, partly because the x-axis are not labeled. Please add the labels (d1?).

Done.

9 Figure 3: isn't there also an effect of measurement frequency in the plots for ID and LD?

We would appreciate the reviewer be more specific and explain how she concluded that an affect of measurement frequency can be seen in Figure 3. In our opinion, further analyses would be needed to evaluate this effect, as previously mentioned. Please, see also our replies to major comment 2) and minor comment 5).

10 section 6: do I understand correctly that "non-SS" cases mean that the GEV parameters are estimated using the data from d* only? Please make it clearer.

We apologize for this lack of clarity but the non-SS GEV parameters were not estimated using the data from d* only: for each duration $d$, non-SS GEV paramaters were estimated on the non-SS sample $x_d$ , independently from other durations . To clarify this point, **Lines 24 to 27, Page 15** have been modified to:

*"In our study, the PWM procedure was applied to estimate SS-GEV parameters $\mu_*$, $\sigma_*$, and $\xi_*$ [Eq. (7)] from $x_{d*}$ [Eq. (8)]. For each duration $d$, PWM were also used to estimate non-SS parameters $\mu_d$, $\sigma_d$, and $\xi_d$ from each of the non-SS samples $x_d$. "*

11 Figure 4: it might be clearer for comparison to use the same US map for the three rows (the first row is different so far). Also there might be here an effect of the measurement frequency for LD and ID, although the spatial patterns are pretty coherent.

When using the same map limits for all maps of Fig. 4 and 5, the nine maps becomes really small while a lot of blank space is present in the first row because there are no stations in Canada and Alaska. The figure is thus less clear when the same latitude and longitude limits are used for the nine maps. Moreover, since the focus is the comparison of the maps placed in the same row (i.e., in the same dataset), we prefer to keep the map limit for SD as they are now.

Concerning the eventual effect of the measurement frequency, please see replies to major comment 2) and minor comments 5) and 9). We agree that spatial patterns are fairly homogeneous suggesting that the series temporal resolution has a weak impact on the estimation of H.

12 Figure 5: same as Fig. 4.

Please, see the reply to the previous comment.

12 p.13 l.26: So if I understand correctly, here you use the H estimated previously and estimation is just for the GEV parameters. Please make it clearer. Have you also tried to estimate all parameters at once (mu*, sigma*, xi*, H) with ML estimators as in Blanchet et al 2016 for example? Theoretically, this should reduce the bias.

Yes, SS-GEV $\mu_*$, $\sigma_*$, and $\xi_*$ presented in our results are estimated applying PWM on the rescaled sample $x_{d*}$ [i.e. using a two-step procedure, as described at **Lines 1 to 5, Page 6**]. To address the reviewer's comment, we rephrased **Lines 24 to 27, Page 15** as reported in our reply to minor comment 10.

The one-step procedure of *Blanchet et al* (2016) has also been tested. To point this out, the relevant paragraph has been rephrased as [**Lines 27 to 30, Page 15**]:

*"Preliminary comparisons of various estimation methods [PWM, classical ML estimators, and one-step GAM-ML; see Sect. 2.2], showed that PWM slightly outperformed the other methods".*

13 Figure 9: isn't there also an effect of measurement frequency in the plots for ID and LD?

Please, see our reply to previous comments, in particular reply to major comment 2) and minor comments 5) and 11).

14 Figure 10: idem

5    Please, see our reply to the previous comment.

15 Figure 11: please add in the legend "with Hosking test at level 5%"

Done.

[revised manuscript text omitted]

---

## Referee Report (RR1)

MINOR COMMENTS

- p 3 l 23 : the fact that $X_d^q$ and $\lambda^{Hq} X_{\lambda d}^q$ have the same distribution comes directly from (2). You don't need for that to have finite moments (please note by the way that exponent 'q' is missing in lambda). However (3) needs finite moments.

- p7 l 14 : I don't think the relation $x_{d2} \leq x_{d1}$ for $d1 < d2$ is always valid. For example, let consider the hourly series with values 10-2-10 mm/h. Then the maximum 2h-intensity is 12/2=6 mm/h, while the maximum 3h-intensity is 22/3>6 mm/h. So $x_{d2} > x_{d1}$ for d2=3h and d1=2h. Also $x_{d2}/x_{d1} < d1/d2$.

- p 7 l 29 : IS→ID ?

- p 7 l 30: the first matrix on the left of Fig. 1(a) → the top left matrix of Fig. 1(a)

- p 8 : Does the SS sample $x_{d,ss}$ comprises the non-SS sample $x_{d}$ ? I guess it should not (for independence testing) but it is not clear to me on (8)

- p 15 l 25 : obtained 12 → obtained for 12

---

## Referee Report (RR2)

**Recommendations: reject.**

This article is too long, very difficult to read, and has almost convinced me that Simple Scaling does work: e.g. Figure 3i and Figure 7, the scaling coefficient H is all over the places, whereas one would expected it to be constant over different durations, being a scale-invariant parameter. An other example is my last comment (just citing your text). Also, it is not clear to me why the authors chose the Bukovsky regions for their regional analysis, when they have a clerar clustering in their Figures 4 and 5 (and S1) which suggests a different regionalization. Moreover, the physical interpretation in Section 5 is not clear (how do you link the behaviour of H across different durations with the topography, as an example). Often the exposition does not follow a logical order. I doubt several results in Section 6. My overall comment is that the authors dwelve too much into the details, but they fail to convey the main message (at least to me … ). I regret to say that my recommendation is a rejection: I leave however the decision to the Editor, since I am not sure I have fully understood the study.

**My numerous comments:**

Equation 2 (and through out the article): I suggest changing the notation by replacing above the equality sign (=) the letter "d" with "pdf". This avoid confusion with "d" denoting the duration.

Page 5, the sentence at line 12-13-14 should be stated earlier, after line 9.

Page 5, lines 24-25: The equalities at line 24 pertains a dilatation of the (values of the) GEV distributions by a factor lambda, whereas Equation 2 pertains to a change of distribution due to a sampling on different durations, with a dilatation factor = lambda. The implication stated at line 25 (the GEV family satisfies Equation 2) is not a direct consequence of the equality at line 24. Rephrase (you might have to add a specific reference, or show explicitely the implication).

Equation 7: I would still prefer you replace d with lambda (since the protagonist here is the dilatation factor)! Here and throughout the article, wherever consistency is required.

Page 6, line 19: I appreciate your clarification wrt my previous comment. In light of that explanation, I suggest adding at the end of this sentence " .. from May to October in the south, and from June to September in the north. Specifically, the 'year' from which the annual maxima was sampled, was limited to the observed summer season, and was defined as ...  ".
Page 7, line 4: write " … at least 85% of valid observations for each summer season, otherwise … "

Page 6, line 23: substitute 'cheked' with 'quality controlled'.
Page 7, line 13: typo: "finstance".
Page 7, line 30: use the new notation 1h30 rather than 1.5h …

Page 7, line 23-24: you need to better state what MSA does. Line 23 is fine, "for intensity AMS" is too concise: you could write something like "for inferring annual maxima / extremes / GEV parameters for durations not sampled by using SS on sampled durations" or something similar.
Page 7, line 27: write *"scaling intervals"* in italic, so that the reader understand this is a definition (valid from hereafter throughout all the article).

Section 4:
I appreciate the efforts of the authors in better explaining this section (both in the text and responses to the previous revisions) and for providing very specific references. The section is indeed more clear. I

do have however still some difficulties in understanding, and I strongly believe that each article should be self-contained (to a certain degree), so here are my suggestions:

    a)  I do still believe that the article will gain in adding a figure showing i) the linear regression between log(duration) and log(moments) and ii) the (previously obtained) regession coefficients Kq and q (as Figure 4a in Panthou et al, 2014). These figures were fundamental for me to understand page 8, lines 8 to 16 (i.e. what you do in the MSA regression and in the slope test).

    b)  In the text regarding the MSA regression, you need to add a sentence which explains that the slope Kq that you are evaluating is equal to -Hq (it took me a while to figure this out), and that this is from Equation 4. Also, at line 8 write "for each q … ", so that it is clear that you have a Kq for each individual separate q.

    c)  In the slope test text (page 8, lines 12-16) you need first to say that you aim to find the scaling coefficient H, which is the slope of the line you obtain while regressing Kq versus q. Then you say that you use a OLS regression and estimate H, and you call such estimate Beta_1 (I would call it \hat{H}). Finally you can say that you perform a t-test on the regression. I suggest stating that the null hypothesis was \hat{H}=K_1, and that if this null hypothesis is not rejected then you set H = \hat{H} = K_1. I would avoid whiting the relation Kq=Beta_0+Beta_1 q (otherwise the reader will ask where Beta_0 is gone, for the equality of K_1= Beta_1).

    d)  The Goodness of Fit test (page 8 lines 19-26) can be explained more clearly, following the cronologiocal order of the calculations you perform: first, you consider the AMS for each duration dj, $AMSdj=\{x_{dj,1},x_{dj,2}, … x_{dj,n}\}$, and you rescale it as $dj^H \cdot AMSdj$, which can be considered as a sample of the reference duration (the choice of notation $x'_{dj}$ is confusing, since one would think it is a sample for the duration dj, whereas it is a sample for the reference duration). You then pool together these rescales samples for all dj. Then you invert equation 2 and obtain, from this large sample, a sample for each duration dj, under SS hypohesis.

    e)  I suggest performing the GOF test also in a cross validation way (e.g. for the ID dataset, when you test the duraton of 3h, you rescale to the duration of 1h all durations excluded the 3h; then you invert Eq.2 and you apply the test to the obtained -slightly smaller- samples).

    f)  Page 9, lines 21-23: this is not entirely clear, do you repeat all steps associated to the GOF part only (page 8, line 7 onwards), or also for the MSA regression and slope test? Only after reading page 11 line 1 I understood that you repeat all (points 1,2,3 at page 8). Make the sentence clearer.

    g)  Text from page 9 line 27 to page 10, line 4: similar to what suggested for the GOF test, describe what you calculate in order of calculation, i.e. define first the normalized RMSE for each station (Eq 11) and after the average over all stations (Eq 10).

Section 4.1:

Figure 1 shows results of slope test and GOF test together. However a reader is intrigued in disentageling the two. You have actually looked at the results seperately, and decided to put them together, with the knowledge that solely the GOF test has a signal. The reader, however, does not know this and remains in the doubt up unti reading at the end of page 10 (lines 25-27). I suggest you to move the sentence at lines 25-27 at the very beginning of the description of these results, after line 10. After this sentence, you should state that the differences in station rejections for the separate durations as shown in Figure 1 are due to the results of GOF test only. And then you keep describing Figure 1 as in your lines 11-25 (which I assume refers solely to the GOF test: this should be made more explicit). Eliminate the [not shown] at line 13 (I think you show this, with the SD dataset)

Figure 2: Page 11, lines 2-4: rewite this as "On the other hand, the extrapolation under SS of the Xd distribution is generally less accurate for durations at the boundaries of the scaling intervals (especially

for the short durations)". You do not show/perform extrapolations of Xd by estimating H with durations outside the scaling interval (as far as I can see), and I believe you meant to rewrite the sentence deleted at line 5.

Figure 2: I would be curious to see the slope and GOF test results also for the cross validation experiment (as in Figure 1, to be able to compare them). This actually is related also to my previous comments e) and f) for Section 4. In alternative, you could reproduce Figure 2 for the non-cross-validation calculation.

Section 4.2

After your introductory sentence (page 11, lines 10-12) I suggest describing first the results pertaining to ΔH (Figure 3 ii, iii, iv) and after the results pertaining to the spatial variability of H (Figure 3 i, and Figures 4 and 5). My suggestions for improvement are:
*   join the paragraph at lines 13-16 with the paragraph at lines 20-25 (page 11).
*   Eliminate (move) the sentence at page 11 lines 26-27 to page 12 line 11, where you will start the description of the spatial variability of H.
*   The text at lines 29-33 is difficult to follow, give (for each sentence) a precise reference to the figure panels.
*   You might want to discuss first the results at page 12 lines 3-10 (which are positive, showing near zero ΔH) and after the results at page 11, line 29 to page 12 line 2 (which are more detailed and negative).
*   Eliminate lines 13-17 and the related sentence at lines 30-31 of page 12 (this is too technical and does not add meat to the article, but rather distracts the reader)

Section 5

Figure 4 and 5 (and Figure S1) show clear spatial clustering in the behaviour of H (as commented in my previous revisions): I still think that the authors should apply a cluster analysis to their own data. Climatology and extremes are different, and extremes might not follow the Bukovsky regions. You can develop a spatial model for SS and IDF estimation (as you state at page 15, lines 16-18) solely considering a regionalization based on the extreme behaviour (rather than pre-set regions).

page 13, lines 9-23: I am not sure the two statistics described here add too much to the article (unless they help the physical interpretation of Section 5.1, which is obscure to me -see following comment-): in fact, I believe that their behaviour is expected, from their definition: shorter duration have a larger number of events, which decay the longer is d1 (S4); conversely, the mean wet time per event decay as d1 grows (averaged on longer duration) … I suggest moving all this to the supplementary material (also the related text at page 14 lines 7-8, 10-13, 23-32). The implications at page 14, line 13 is not clear.

Section 5.1

From page 13 line 30 to page 14 line 3: The link between the behaviour of H and the physical characteristics of the precipitation in the region is missing / not clear. Similarly, at page 14 lines 4-7, the implication is not clear at all.

Maybe you need to explain beforehand what does it means (physically) when H is small and when H is large (as at page 15, lines 13-15), when H increase and when H decreases with d1.

Page 14, lines 14-22: good interpretation of the behaviour in Region D.
Page 15 lines 2-5: clear, whereas you loose me at lines 6-9.

page 15, lines 10-11: I disagree with this sentence, I have not seen any evidence that the material illustrated here supports these results (you have not proven this).

Section 6

From page 15 line 30 to page 16 line 7: since you are assessing a distribution, why don't you use a KS statistics, rather than inventing an engeneered metric which compares the quantiles? Recall Equation 11 for clarity, please.

Page 16, line 13: for the ID and LD datasets, the behaviour of the SS parameters versus non-SS parameters is opposite, they cannot be both more right skewed for the SS estimation.

Page 16, line 16: it seems to me that the discerepancies between SS and non-SS parameters for the LD dataset and long durations (the $\Delta\mu$ and $\Delta\sigma$ in the supporting material) and quite big.

Page 16, lines 14-21: there is something strange about your results: you state that the estimation of the scale parameter has a small uncertainty when the shape parameter is correctly estimated. However, from Figure 8. I can see that the shape parameter is not correctly estimated (red and black curves do not match). What is the implication of the mismatch of the shape parameter for the non-SS and SS estimation on your results? The shape parameter is usually set to zero because it is all over the places, and often not-significantly different from zero (as you find in your Figure 9c: the estimated shape parameter is quite noisy)!!!

Page 16, lines 22-23: I agree with this sentence, nice spatial coherence.
Page 16, lines 25-26: why SS shape parameter is nearer to zero and exhibit less spread? Is this an effect of the assumptions of SS?
Page 16, line 27: very difficult sentence to read (essentially the shape parameter is zero).

Page 16, lines 27-31: these results are not intuitive. Figure 10 suggests that the shape parameter for the non-SS estimates is predominantely zero, whereas for the SS estimates there is a large proportions of positive and negative shape parameters. However, the right column of Figure 8 shows exactly the opposite (SS estimate are nearer to zero than non-SS estimates). I assume the difference is due to the very different width of confidence interval associated to the estimate of the shape parameter, for non-SS and SS samples. Are the sample sizes very different (I believe so … given that x* in equation 8 is obtained from a very large sample). Then maybe these results are artificial … (as you conclude afterwards, at page 17, lines 3-5: but this link is not explicit)!!!
Page 17, lines 7: eliminate (not clear what this refer too).

Overall, the text starting at page 16 line 27 and ending at page 17 line 10 should be all reorganized ina more coherent single paragraph.

Page 17, lines 12-15: Figure S13 shows that for shape parameter equal to 0 (the majority of the stations) the error of quantiles estimated by the non-SS is smaller than that for SS estimates.

---

## Author Response (AR2)

Dear Editor,

Please find enclosed the second revision of the manuscript **"Simple Scaling of Extreme Precipitation in North America"** by Innocenti et al. to be considered for publication in HESS.

According to comments from the first and third reviewers, we modified several paragraphs resulting in a clearer and more comprehensive description of methodology and results, in particular for Sect 4 to 6. Methodological choices for the regional analysis (Section 4.2) are discussed and reviewed in our reply to the first reviewer. In particular, we explained and motivated in details our methodology for the definition of geographical regions in North America in our reply to comment 15 a). Some additional analysis has been also presented in the reply to this specific comment. Finally, important considerations and further explanations concerning the results presented for Sect. 6 (SS GEV models) have been added.

We provide below a detailed response to each comment and a copy of the revised manuscript in "track changes" mode. The following specific colors are used to link corrections and reviewers' comments:

- blue$^{R1}$ is used to underline changes related to first referee comments,

- red$^{R2}$ is used for changes related to the second referee comments,

- purple$^{R3}$ is used for changes related to the third referee comments,

- and gray is used for all other changes.

Line and page numbering **(in bold)** refers to the revised manuscript in "track changes" mode attached to this reply.

Sincerely,

Silvia Innocenti, on behalf of the co-authors.

**Authors' response to 1$^{st}$ referee's comments**

This article is too long, very difficult to read, and has almost convinced me that Simple Scaling does work: e.g. Figure 3i and Figure 7, the scaling coefficient H is all over the places, whereas one would expected it to be constant over different durations, being a scale-invariant parameter. An other example is my last comment (just citing your text). Also, it is not clear to me why the authors chose the Bukovsky regions for their regional analysis, when they have a clerar clustering in their Figures 4 and 5 (and S1) which suggests a different regionalization. Moreover, the physical interpretation in Section 5 is not clear (how do you link the behaviour of H across different durations with the topography, as an example). Often the exposition does not follow a logical order. I doubt several results in Section 6. My overall comment is that the authors dwelve too much into the details, but they fail to convey the main message (at least to me ... ). I regret to say that my recommendation is a rejection: I leave however the decision to the Editor, since I am not sure I have fully understood the study.

**My numerous comments:**

1. Equation 2 (and through out the article): I suggest changing the notation by replacing above the equality sign (=) the letter "d" with "pdf". This avoid confusion with "d" denoting the duration.

   Thank you for the suggestion but the relationship in Eq. 2 is more general and does not only concern the pdf. To avoid confusion, we replaced $\overset{d}{=}$ by $\overset{dist}{=}$ in **Eq. 2**, at **Line 25, Page 3**, and at **Line 23, Page 5**.

2. Page 5, the sentence at line 12-13-14 should be stated earlier, after line 9.

   The paragraph has been moved to **Lines 6 to 10, Page 5**.

3. Page 5, lines 24-25: The equalities at line 24 pertains a dilatation of the (values of the) GEV distributions by a factor lambda, whereas Equation 2 pertains to a change of distribution due to a sampling on different durations, with a dilatation factor = lambda. The implication stated at line 25 (the GEV family satisfies Equation 2) is not a direct consequence of the equality at line 24. Rephrase (you might have to add a specific reference, or show explicitely the implication).

   We understand the reviewer perspective and we agree that we did not clearly explain these points in Section 2. In particular, we did not explicitly mention that SS models assume statistical scale-invariance with respect to the sampling scale [as stated, for instance, at **Lines 9 to 12, Page 2**].
   In order to explicitly mention this point in the text we rewrote **Lines 20 to 22, Page 3** as:

   *"When the equality in Eq. (1) holds for the cumulative distribution function (cdf) of the precipitation intensity $X$ sampled at two different durations $d$ and $\lambda d$, the Simple Scaling (SS) can be expressed as [Gupta and Waymire, 1990; Menabde et al, 1999]:*

   $$X_d \overset{dist}{=} \lambda^H X_{\lambda d}\text{"}$$

   We also agree that the paragraph describing the scale-invariant nature of the GEV distribution may be misleading since equations at **Line 23, Page 5** only imply the validity of Eq. (1) for the GEV cdf with respect to a constant multiplicative factor $\lambda$. Then, Eq. (2) follows if we consider the GEV cdf scale invariance being valid when changing the observational scale from $d$ to $\lambda d$. To be more precise, we thus rephrased **Lines 24 to 27, Page 5** as:

*"This means that the GEV family described by Eq. (5) and (6) satisfies Eq. (1) and thus complies with statistical scale invariance for any constant multiplicative transformation of $X$. When this scale invariance is further assumed for the change of observational scale from duration $d$ to $\lambda d$ [as in Eq. (2)], the wide sense SS definition [Eq. (3)] gives: ... [Eq. (7)] ".*

4. Equation 7: I would still prefer you replace d with lambda (since the protagonist here is the dilatation factor)! Here and throughout the article, wherever consistency is required.

We appreciate the reviewer's suggestion but, in our opinion, it is important, here, to highlight the fact that the AMS distribution for any duration $d$ can be always reparametrized in terms of the GEV distribution parameters of an unit scale used as reference duration ($d^* = 1$). This notation allows in fact to underline that GEV parameters $\mu_d$, $\sigma_d$, and $\xi_d$ can be thought almost as adimensional parameters for any arbitrary set of durations in the range of scale for which SS is valid.

In our application [Sect. 6], the use of $d_* = 1h$ and, thus, of $\lambda = d$ [**Line 3, Page 4** and **Line 3, Page 4**] allows to compare the parameters of different scaling intervals without the specification of a particular $\lambda$ for each interval, and to underline the physical interpretation of the SS GEV parameters in the different scaling intervals.

5. Page 6, line 19: I appreciate your clarification wrt my previous comment. In light of that explanation, I suggest adding at the end of this sentence " .. from May to October in the south, and from June to September in the north. Specifically, the 'year' from which the annual maxima was sampled, was limited to the observed summer season, and was defined as ... ".

To improve the text we modified **Lines 23 to 28, Page 6** to:

"For this reason, the *year* from which the annual maxima was sampled was limited to the recording season going from June to September for northern stations [stations located north of the $52^{nd}$ Parallel] and from June to September for the southern stations. As a result, 122 days a year were used for northern stations and 184 days a year for remaining stations."

6. Page 7, line 4: write " ... at least $85\%$ of valid observations for each summer season, otherwise ... "

Thank you for the suggestion. Since the May to October or June to September periods don't correspond to the summer season, we changed the sentence for the following [**Line 10, Page 7**]:

"at least $85\%$ of valid observations for each May to October (or June to September) period, otherwise the corresponding year was considered as missing."

7. Page 6, line 23: substitute 'cheked' with 'quality controlled'.

Done.

8. Page 7, line 13: typo: "finstance".

Corrected, thank you.

9. Page 7, line 30: use the new notation 1h30 rather than 1.5h ...

Done: the paragraph has been modified after the addition of Fig. 1. [see our reply to your comment 12 a)].

10. Page 7, line 23-24: you need to better state what MSA does. Line 23 is fine, "for intensity AMS" is too concise: you could write something like "for inferring annual maxima / extremes / GEV parameters for durations not sampled by using SS on sampled durations" or something similar.

The expression has been replaced by *"for modeling AMS empirical distributions"* [**Line 30, Page 7**].

For seek of precision, we did not add "for durations not sampled by using SS on sampled durations" since SS models can be used for modeling sampled duration distributions, and not only for non-sampled ones. Accordingly, in fact, we estimated SS and evaluated its performances in both "calibration" and "cross-validation" mode [see also reply to comment 12 e) - (Section 4)].

11. Page 7, line 27: write "scaling intervals" in italic, so that the reader understand this is a definition (valid from hereafter throughout all the article).

Done.

12. **Section 4:**

a) I do still believe that the article will gain in adding a figure showing i) the linear regression between log(duration) and log(moments) and ii) the (previously obtained) regession coefficients $K_q$ and q (as Figure 4a in Panthou et al, 2014). These figures were fundamental for me to understand page 8, lines 8 to 16 (i.e. what you do in the MSA regression and in the slope test).

Following your suggestion we added a new figure [Fig. 2 on **Page** 27] which explains the various steps of the methodology used for non-parametric SS estimation [Section 4]. In particular, the new Fig. 1 contains six panels representing the following steps:

Panel a): Definition of the SD, ID and LD scaling datasets.

Panel b): Example of the definition of five scaling intervals for the ID dataset. This panel is intended to help interpreting Fig.2 and 3.

Panel c): MSA regression: estimation of $K_q$ slope coefficients.

Panel d): Slope test: testing linearity of coefficient $K_q$ on $q$.

Panel e): Definition of Valid SS stations.

Panel f): Example of Valid SS station proportion and Normalized RMSE ($\overline{\overline{r}}_{x_d}$) as presented, respectively, in Fig. 2 and 3 [Fig. 1 and 2 in the previous version of the manuscript].

The following references were also added in the text:

– at **Line 9, Page 7**: *"[see Figure 1(a)]"*;

– at **Line 7, Page 8**: *"More schematically, Fig. 1(b) shows an example of the first five 6-duration scaling intervals for the ID dataset [i.e. 1h - 6h, 2h - 7h, …, 5h - 10h, containing six contiguous durations defined with an increment of 1h]."* ;

- at **Line 17, Page 8**: *"[see Fig. 1(c) for a graphic example]"*;
- at **Line 20, Page 8**: *"[see Fig. 1 (d)]"*;
- in the **captions of Fig. 2 and 3**: *"See Fig. 1 (b) and (f) for the identification of durations and scaling intervals within each matrix.".*

b) In the text regarding the MSA regression, you need to add a sentence which explains that the slope Kq that you are evaluating is equal to -Hq (it took me a while to figure this out), and that this is from Equation 4. Also, at line 8 write "for each q ... ", so that it is clear that you have a $K_q$ for each individual separate q.

Thank you for the suggestion but the equality $K_q = H\ q$ is what we want to evaluate with slope test, while the MSA regression should not assume this linear relationship: as explained at **Lines 7 to 8, Page 4**, SS models can be considered valid only if $K_q \approx Hq$.

As you suggested, we added "each" at **Line 15, Page 8** which now reads: *"for each $q = 0.2, 0.4, \ldots, 2.8, 3$, the slopes $K_q$ of the log-log linear relationships between the empirical $q$−moments ...."*

c) In the slope test text (page 8, lines 12-16) you need first to say that you aim to find the scaling coefficient H, which is the slope of the line you obtain while regressing Kq versus q. Then you say that you use a OLS regression and estimate H, and you call such estimate $\beta_1$ (I would call it $\hat{H}$). Finally you can say that you perform a t-test on the regression. I suggest stating that the null hypothesis was $\hat{H} = K_1$, and that if this null hypothesis is not rejected then you set $H = \hat{H} = K_1$. I would avoid whiting the relation $Kq = \beta_0 + \beta_1 q$ (otherwise the reader will ask where $\beta_0$ is gone, for the equality of $K_1 = \beta_1$).

We apologize for this lack of clarity. To improve the description of this methodological step, which aim at testing the linearity of the $K_q$ scaling exponents estimated at the previous step, we modified **Lines 19 to 25, Page 8** to:

*"To verify the SS assumption that the estimated $K_q$ exponents vary linearly with the moment order q, i.e. $K_q \approx Hq$, an OLS regression between the MSA slopes $K_q$ and q was applied [see Fig. 1 (d)]. For the regression line $K_q = \hat{h}_0 + \hat{h}_1 q$, a Student's t-test was then used to test the null hypothesis $\mathbf{H}_0$: $\hat{h}_1 = K_1$. If $\mathbf{H}_0$ was not rejected at the significance level $\alpha = 0.05$, the SS assumption was considered appropriate for the scaling interval and the simple scaling exponent $H = K_1$ was retained."*

d) The Goodness of Fit test (page 8 lines 19-26) can be explained more clearly, following the cronologiocal order of the calculations you perform: first, you consider the AMS for each duration $d_j$, AMS $d_j = \{xd_j, 1, x_{d_j,2}, \ldots x_{d_j,n}\}$, and you rescale it as $d_j H \cdot$ AMS $d_j$, which can be considered as a sample of the reference duration (the choice of notation $x'_{d_j}$ is confusing, since one would think it is a sample for the duration $d_j$, whereas it is a sample for the reference duration). You then pool together these rescales samples for all dj. Then you invert equation 2 and obtain, from this large sample, a sample for each duration dj, under SS hypohesis.

Following your suggestion we changed notation for the description of the SS sample construction and we rephrased **Lines 28 to 13, Page 8** to:

*"To this end, each AMS, $\boldsymbol{x}_{d_j} = \left( x_{d_j,1}, x_{d_j,2}, \ldots, x_{d_j,i}, \ldots x_{d_j,n} \right)$, recorded at duration $d_j$ was rescaled at the reference duration d\* by inverting Eq. (2):*

$$\boldsymbol{x^*}_{d_j} = \left( d_j{}^H x_{d_j,1}, d_j{}^H x_{d_j,2}, \ldots, d_j{}^H x_{d_j,i}, \ldots d_j{}^H x_{d_j,n} \right) \tag{8}$$

*where $n$ represents the number of observations (years) in $\boldsymbol{x}_{d_j}$. Then, the pooled sample, $\boldsymbol{x}_{d^*}$, of the $D$ rescaled AMS, $\boldsymbol{x}^{\boldsymbol{*}}_{d_j}$, was used to define $X_{d^*}$ under the SS assumption:*

$$\boldsymbol{x}_{d^*} = \left(\boldsymbol{x}^{\boldsymbol{*}}_{d_1}, \ldots, \boldsymbol{x}^{\boldsymbol{*}}_{d_j}, \ldots, \boldsymbol{x}^{\boldsymbol{*}}_{d_D}.\right) \tag{9}$$

*Since, in Eq. (9), $D$ represents the number of durations $d_j$ in the scaling interval, $n \times D$ rescaled observations were included in $\boldsymbol{x}_{d^*}$."*

e) I suggest performing the GOF test also in a cross validation way (e.g. for the ID dataset, when you test the duraton of 3h, you rescale to the duration of 1h all durations excluded the 3h; then you invert Eq.2 and you apply the test to the obtained -slightly smaller- samples).

Thank you for pointing out our omission. The GOF and Slope test were also applied during the cross-validation experiment, which consists of repeating steps 1 to 3 described on **Page** 8 [see below], but we did not present the relative results since these are not substantially different to those of Fig. 2 [Fig. 1 in the previous version]. For completeness, we added these results (Slope and GOF tests in cross-validation) in Fig. S4 of the supplementary material.

To clarify that the steps 1 to 3 were repeated in a cross-validation setting, **Lines 24 to 29, Page 9** were modified to:

*"The SS model validity and the mean error resulting from approximating the $X_d$ distribution by the SS model were then evaluated in a cross-validation setting. For this analysis, each duration was iteratively excluded from each scaling interval and the scaling model re-estimated at each station by repeating steps 1 to 3 [MSA regression, Slope test, and GOF tests]".*

Then, we added the following reference to cross-validation results presented in Fig. S4 [**Line 13, Page 11**]:

*"These findings were also confirmed by cross-validation experiments. The proportion of valid SS stations resulting from cross-validation Slope and GOF tests were similar, event if slightly lower, to proportions displayed in Fig. 2 [see Fig. S4 of the supplementary material].*

Moreover, the following explanation has been added in the introduction of the supplementary material:

*"Figure S4 presents the results for the cross-validation experiment for the SS model (Slope and GOF tests) for each duration and scaling interval. "*

f) Page 9, lines 21-23: this is not entirely clear, do you repeat all steps associated to the GOF part only (page 8, line 7 onwards), or also for the MSA regression and slope test? Only after reading page 11 line 1 I understood that you repeat all (points 1,2,3 at page 8). Make the sentence clearer.

The sentence has been modified specifying that steps 1 to 3 of the methodology [MSA regression, Slope test, and GOF tests] were repeated in cross-validation experiments. Please, see our reply to the previous comment.

g) Text from page 9 line 27 to page 10, line 4: similar to what suggested for the GOF test, describe what you calculate in order of calculation, i.e. define first the normalized RMSE for each station (Eq 11) and after the average over all stations (Eq 10).

Following your suggestion **Lines 1 to 13, Page 10** were modified to:

*"For each station s, the normalized RMSE, $\overline{\epsilon}_{x_{d,s}}$, was estimated:*

$$\overline{\epsilon}_{x_{d,s}} = \frac{\epsilon_{x_{d,s}}}{\overline{x}_{d,s}} \qquad (8)$$

*where $\epsilon_{x_{d,s}}$ and $\overline{x}_{d,s}$ are, respectively, the RMSE and the mean value of all $X_d$ quantiles of order $p > 0.5$. Then, the average over all stations of the normalized RMSE, $\overline{\overline{\epsilon}}_{x_d}$, was computed for each scaling interval and duration:*

$$\overline{\overline{\epsilon}}_{x_d} = \frac{1}{n_s} \sum_{s=1}^{n_s} \overline{\epsilon}_{x_{d,s}} \qquad (9)$$

*where $n_s$ is the number of valid SS stations in the dataset. Note that $\overline{\overline{\epsilon}}_{x_d}$ is a measure of error, meaning that values of $\overline{\epsilon}_{x_{d,s}}$ closer to 0 correspond to a better fit than larger values. ".*

13. **Section 4.1:**

a) Figure 1 shows results of slope test and GOF test together. However a reader is intrigued in disentageling the two. You have actually looked at the results seperately, and decided to put them together, with the knowledge that solely the GOF test has a signal. The reader, however, does not know this and remains in the doubt up unti reading at the end of page 10 (lines 25-27). I suggest you to move the sentence at lines 25-27 at the very beginning of the description of these results, after line 10. After this sentence, you should state that the differences in station rejections for the separate durations as shown in Figure 1 are due to the results of GOF test only. And then you keep describing Figure 1 as in your lines 11-25 (which I assume refers solely to the GOF test: this should be made more explicit). Eliminate the [not shown] at line 13 (I think you show this, with the SD dataset)

We agree with the reviewer and we modify the first paragraph of section 4.1 to [see **Line 15, Page 10**]:

*"Figure 2 presents the results of steps 1 to 3 of the methodology for evaluating the SS validity. For all the three scaling datasets, no particular pattern was observed for slope test results, with at most 2% of the stations within each scaling interval displaying a non linear evolution of the scaling exponent with the moment order. For this reason, Fig. 2(a)-(c) show, for each scaling interval and duration, the proportion of valid SS stations without differentiating for slope or GOF test results. "*

b) Figure 2: Page 11, lines 2-4: rewite this as "On the other hand, the extrapolation under SS of the Xd distribution is generally less accurate for durations at the boundaries of the scaling intervals (especially for the short durations)". You do not show/perform extrapolations of Xd by estimating H with durations outside the scaling interval (as far as I can see), and I believe you meant to rewrite the sentence deleted at line 5.

Thank you fo the suggestion. Apply the cross-validation for durations at the boundaries of the scaling intervals results in the estimation of $H$ and $X_d$ using non-recorded durations (i.e. durations outside the scaling interval since one boundary is moved). For instance, the first cross-validation for the interval 1h-6h in the ID dataset consists in the estimation of the SS model on durations 2h, 3h, 4h, 5h, and 6h. Then, this model is evaluated for 1h which technically is outside of the scaling interval "2h-6h" used for the estimation. For this reason, we modified **Line 20, Page 11** to:

*" Conversely, the extrapolation under SS of the $X_d$ distribution is generally less accurate for durations at the boundaries or outside the scaling interval used to estimate $H$. "*

c) Figure 2: I would be curious to see the slope and GOF test results also for the cross validation experiment (as in Figure 1, to be able to compare them). This actually is related also to my previous comments e) and f) for Section 4. In alternative, you could reproduce Figure 2 for the non-cross- validation calculation.

Please, refer to reply to comment 12 e): Fig. S4 showing Slope and GOF test results for cross-validation experiments has been added to the supplementary material.

14. **Section 4.2:** After your introductory sentence (page 11, lines 10-12) I suggest describing first the results pertaining to $\Delta H$ (Figure 3 ii, iii, iv) and after the results pertaining to the spatial variability of H (Figure 3 i, and Figures 4 and 5). My suggestions for improvement are:

a) Join the paragraph at lines 13-16 with the paragraph at lines 20-25 (page 11).

The paragraphs are now contiguous.

b) Eliminate (move) the sentence at page 11 lines 26-27 to page 12 line 11, where you will start the description of the spatial variability of H.

Done. **Lines 7 to 10, Page 12** now reads:

*"Figures 4(ii)-(iv) show the median, Interquantile Range (IQR), and quantiles of order 0.1 and 0.9 of the $\Delta_{H_{(j)}}$ distribution over valid SS stations for all relevant scaling intervals."*

c) The text at lines 29-33 is difficult to follow, give (for each sentence) a precise reference to the figure panels.

Thank you for the suggestion. The references to figures have been added and the paragraphs has been rewritten. Please, see our reply to your following comment.

d) You might want to discuss first the results at page 12 lines 3-10 (which are positive, showing near zero $DeltaH$ ) and after the results at page 11, line 29 to page 12 line 2 (which are more detailed and negative).

The paragraphs have been inverted [**Lines 10 to ??, Page 12**]:

*"Adding new durations to the scaling intervals, median $\Delta_{H_{(j)}}$, as well as its IQR, increased for all $d_1$. Nonetheless the median scaling exponent variation was generally smaller than 0.05, except for a relatively small proportion of stations. Equally important, $|\Delta_{H_{(j)}}|$ was generally centered on 0 and for all $d_1 \geq 1$ h more than 50% of stations had $|\Delta_{H_{(12)}}| \leq 0.025$ (SD dataset) and $|\Delta_{H_{(18)}}| \leq 0.03$ (ID dataset) [Fig. 4 (ii)-(iii)].*
*For some stations, a dramatic difference could exist in IDF estimations obtained with the different definitions of the scaling interval. For instance, for the 24-duration scaling interval "1h - 24h" (ID dataset), the median $\Delta_{H_{(24)}}$ was equal to 0.047 [Fig. 4(iv) b)]. For the interval "15min - 6h" (SD dataset), $\Delta_{H_{(24)}}$ was even larger, with a median scaling exponent variation approximately equal to $0.087$ and with 25% of stations having $\Delta_{H_{(24)}} \geq 0.11$ [Fig. 4(iv) a)]. Finally, changes in H values were also important when comparing 6- and 12-duration scaling intervals when $d_1 \leq 1$ h (SD and ID datasets) and in LD dataset [Fig. 4 (ii)]."*

e) Eliminate lines 13-17 and the related sentence at lines 30-31 of page 12 (this is too technical and does not add meat to the article, but rather distracts the reader).

Thank you for the suggestion. The sentence needed to be rephrased as also pointed out by Reviewer 3 [see comment

3 of the 3rd Reviewer]. However, we consider that the paragraph is important for the interpretation of Fig. 4 and 5 and following results since it highlights an important issue on the uncertainty of the $H$ estimator. In this regard, note that Reviewer 3 did not suggest to eliminate the sentence but to rephrase it. To simplify the text without eliminating all the information, **Lines 34 to 12, Page 12** have been modified to:

*"This result could be partially explained by the use of scaling intervals having equally spaced durations. This implies that the mean distance between the logarithms of durations in the scaling interval decreases as $d_1$ increases. Hence, the OLS estimator of $H$ used in the MSA regression may have larger variance for longer $d_1$, especially when scaling intervals include few durations. Larger uncertainty may thus have an impact on the $H$ estimation for the longest $d_1$ scaling intervals of SD. However, as showed in next sections, $H$ spatial distribution may also explain the greater variability of the scaling exponent for $d_1$ greater than a few hours."*

15. **Section 5:**

a) Figure 4 and 5 (and Figure S1) show clear spatial clustering in the behaviour of H (as commented in my previous revisions): I still think that the authors should apply a cluster analysis to their own data. Climatology and extremes are different, and extremes might not follow the Bukovsky regions. You can develop a spatial model for SS and IDF estimation (as you state at page 15, lines 16-18) solely considering a regionalization based on the extreme behavior (rather than pre-set regions).

The reviewer raised an interesting question regarding the construction of homogeneous region for extremes, which, as underlined by the reviewer, generally display a different spatial distribution than climatology. Two distinct issues must be discussed to correctly address the reviewer's comment: on the one hand, the choice of a methodology for the definition of geographical regions which is consistent with our analysis, and, on the other hand, the difference between "regionalization and regional estimation" evoked by the reviewer and the more general concept of "spatial model" for IDF parameters mentioned in our text.

To address the problem of the identification of homogeneous regions for extremes, a panoply of algorithms based on different approaches has been developed in the literature [e.g., Grimaldi et al, 2011; Hosking and Wallis, 1997; among others]. According to several authors [e.g., Wazneh et al., 2015; and references therein], the definition of homogeneous regions involves a great amount of subjectivity due to the choice of :

i) the definition of *'homogeneity'* and the subsequent choice of the site *characteristics* or *statistics* to be used for the classification [for instance, should one use the geographical location, elevation, climatological features, and/or precipitation extreme indexes for the observed sites?];

ii) the number of regions and some basic classification criteria [e.g., should be the regions be geographically contiguous?] which depend on several factors such as the spatial scale of the analysis and on the subsequent use of the classification [e.g., should one use the regionalization for approximating unknown quantities at un-gaged sites or it is only a descriptive tool?];

iii) the algorithm performing the classification based on the selected classification variables [site characteristic(s) and/or site statistic(s)];

Points i)-iii) are also closely interconnected. For instance, if one uses a cluster analysis based on site extreme precipitation statistics, stations from different geographical areas could be pooled in the same group [see figure below]. This non-spatially-contiguous classification may lack of physical consistency and may be hardly interpreted in term of climatology.

An example is provided in Figure 1 which shows the results of a basic *k-means cluster analysis* performed on $H$ (scaling exponent) and $\bar{P}_{24h}$ (climatic median of station AMS for the duration $d = 24h$) for the scaling interval presented in Fig. 5 [Fig. 4 of the previous version of the paper mentioned by the reviewer]. The number of groups

[Figure]

**Figure 1.** Example of regions obtained after applying a k-mean cluster analysis applied on $H$ (scaling exponent) and $\bar{P}_{24h}$ (climatic median of station AMS for the duration $d = 24h$) for scaling interval presented in Fig. 5 of the paper. Stations of the same color belong to the same cluster.

has been arbitrarily set to $N_{reg} = 6$, since six geoclimatic regions were used in the paper based on Bukovsky (2012) classification [see Fig. 7]. However, more objective criteria should be defined to guide the selection of an appropriate number of clusters.

Even if, at a first sigh, the clustering in Fig 1 does not seem to significantly differ with the one used in our analysis, it is important to highlight that:

  a) As observed by the reviewer in its previous revision, some regions used in the paper may be split in various subregions; however, these differences seem to be mainly driven by the NW-to-SE gradient of $\bar{P}_{24h}$ (see Fig. S1 of the supplementary material).

  b) As previously noted, the method produces non-contiguous regions which limits the possibility of interpreting the physical meaning of the values taken by the scaling exponent at a regional scale in terms of geoclimatic characteristics of the regions. To improve the performances of the clustering method and allow for the definition of physically sound regions, one should probably include other classification variables, such as the

geographical coordinates, the mean Total Annual Precipitation and Temperature, or other climatological indexes.

c) The classification is based on the $H$ and $\bar{P}_{24h}$ which are at-site statistics of rainfall extremes estimated on available AMS. The estimation of this unknown quantities entails an amount of uncertainty which may impact the classification in an unpredictable way, especially if observed series are short. Conversely, climatic statistics such as the mean Total Annual Precipitation and Temperature can be generally estimated with lower uncertainty.

Essentially, more sophisticated methods should be used to provide an accurate and physically consistent clustering in order to rigorously define the spatial structure of $H$ and, eventually, to approximate AMS distributions with a regional approach, as recommended by the reviewer. Although interesting, we sincerely believe that these analyses deserve to be comprehensively done and are out of scope of this study. In that perspective, it is important to remind that :

Firstly, our study did not intend, at this point, to use a regional approach to estimate SS models and IDF curves; the main objective of Section 5, is to investigate the relationship between the spatial distribution of $H$ and the geographical and climatological features of the study region. This qualitative analysis was merely descriptive and did not aim at defining a regional and strictly homogeneous value of $H$.

Secondly, to our opinion, $H$ can be realistically expected to be a climatological characteristic of extremes with a smoother behavior in space than other extreme statistics [e.g.,high order moments or quantiles of extreme precipitation distributions]. This is also confirmed by maps in Fig. 5 [Fig. 4 in the previous version].

Finally, the Bukovsky (2012) regions correspond to a subdivision of North America territory into homogeneous climatological regions that was already been used in the literature [e.g., Separovic et al., 2013; Prein et al., 2016;]. Our analysis could hardly reproduce the accuracy of this classification with the available data. Hence, Bukovsky regions appeared to be a convenient and reasonable choice for our study.

We realized that some confusion may arise by the expression "spatial model" used in the following sentence [**Line 12, Page 16**]: *"Even more important, these results could help for the definition of IDF relationships at non-sampled locations by the construction of spatial models for the IDF parameter H"*. Note that this sentence did not refer to to regional estimation approaches such as RFA (Regional Frequency Analysis). These regionalization approaches effectively rely on an accurate definition of homogeneous regions and the validation of homogeneous regions in order to pool the series from stations with similar characteristics. However, **Line 12, Page 16** referred to "spatial models" aiming at explicitly modeling the spatial distribution of the GEV and/or IDF parameters such as in Begueria, and Vicente-Serrano (2006), Blanchet and Lehning (2010), Panthou et al (2012), and Davison et al. (2012). To avoid any confusion we rephrased the sentence at **Line 12, Page 16** as:

*"Even more important, these results give useful guidelines for modeling the spatial distribution of H, which could help for the definition of IDF relationships at non-sampled locations."*

– *Begueria, S and SM Vicente-Serrano (2006). Mapping The Hazard Of Extreme Rainfall By Peaks Over Threshold Extreme Value Analysis And Spatial Regression Techniques. J Appl Meteorol. Vol. 45. no. 1, pp. 108–124.*

– *Blanchet, J, Marty, C, and M Lehning (2009). Extreme value statistics of snowfall in the Swiss Alpine region. Water Resour Res. Vol. 45. no. 5.*

– *Blanchet, J. and M. Lehning (2010). Mapping snow depth return levels : smooth spatial modeling versus station interpolation. Hydrol Earth Syst Sc. Vol. 14. no. 12, pp. 2527–2544.*

– *Davison, Anthony C, SA Padoan, M Ribatet, et al. (2012). Statistical modeling of spatial extremes.*

– *Grimaldi S, Kao S-C, Castellarin A, Papalexiou S-M, Viglione A, Laio F, Aksoy H and Gedikli A (2011) Statistical Hydrology. In: Peter Wilderer (ed.) Treatise on Water Science, vol. 2, pp. 479–517 Oxford: Academic Press.*

– *Hosking, J.R.M., Wallis, J.R., 1997. Regional Frequency Analysis: An Approach Based on L-moments. Cambridge, UK, 244pp*

– *Panthou, G., T. Vischel, T. Lebel, J. Blanchet, G. Quantin, and A. Ali (2012). Extreme rainfall in West Africa : A regional modeling. Water Resour Res. Vol. 48. no. 8, n/a–n/a.*

– *Prein, A. F., Holland, G. J., Rasmussen, R. M., Clark, M. P., and Tye, M. R. 2016. Running dry: The US Southwest's drift into a drier climate state. Geophysical Research Letters, 43(3), 1272-1279.*

– *Separovic L, A Alexandru, R Laprise, A Martynov, L Sushama, K Winger, K Tete, M Valin. 2013. Present climate and climate change over North America as simulated by the fifth-generation Canadian regional climate model. Clim Dyn 41:3167-3201. DOI 10.1007/s00382-013-1737-5.*

– H. Wazneh, F. Chebana, T.B.M.J. Ouarda, *Delineation of homogeneous regions for regional frequency analysis using statistical depth function, Journal of Hydrology, Volume 521, February 2015, Pages 232-244, ISSN 0022-1694, https://doi.org/10.1016/j.jhydrol.2014.11.068.*

b) page 13, lines 9-23: I am not sure the two statistics described here add too much to the article (unless they help the physical interpretation of Section 5.1, which is obscure to me -see following comment-): in fact, I believe that their behaviour is expected, from their definition: shorter duration have a larger number of events, which decay the longer is $d_1$ (S4); conversely, the mean wet time per event decay as d1 grows (averaged on longer duration) ... I suggest moving all this to the supplementary material (also the related text at page 14 lines 7-8, 10-13, 23-32). The implications at page 14, line 13 is not clear.

Thank you for the suggestion but we consider that $\bar{N}_{eve}$ and $\bar{T}_{wet}$ concretely show some important features of the extreme events sampled by the AMS observed for the set of durations included in the scaling intervals. Therefore, we didn't move this paragraph to the supplemtary material. We agree that their overall behavior and patterns over $d_1$ are generally expected. However, the differences among panels of Fig. S5 and S6 (Fig. S4 and S5 of the previous version of the supplementary material) are crucial and support the physical interpretation of $H$ and the regional analysis given at **Page 14**.

16. **Section 5.1:**

a) From page 13 line 30 to page 14 line 3: The link between the behaviour of H and the physical characteristics of the precipitation in the region is missing / not clear. Similarly, at page 14 lines 4-7, the implication is not clear at all.

In order to improve the discussion of Fig. 8 we recalled that *"higher $H$ values are associated with larger variations in moment values as the scale is changed (i.e. a stronger scaling), while $H$ close to zero means that the $X_d$ distributions for different durations $d$ more closely match each other."* [**Line 28, Page 4**] This was also stated at **Line 29, Page 15**, **Line 7, Page 16**, and **Line 27, Page 19** ,and at **Line 3, Page 15** for $H_{depth} = 1 - H$. In this regard, note that, as reported in the literature, higher $H$ values have been generally observed for shorter-duration intervals and regions dominated by convective precipitation (e.g., Borga et al., 2005; Nhat et al., 2007; Ceresetti et al., 2010; Panthou et al., 2014, and references therein); this was mentioned at **Line 20, Page 4**.

Hence, we modified **Lines 17 to 25, Page 14** to:

*"For $d_1 \leq 24$ h, Fig. 8 (a) displays lower values of $H$ than Fig. 8 (e)-(f), meaning that smaller variation in AMS moments are observed in A1 and A2 when the scale is changed. This difference can be partially explained by the weaker impact of convection processes in generating very short duration extremes in North-West coastal regions with respect to southern areas (regions E and F). For northern regions, in fact, the transition between short and long duration precipitation regimes may be smoothed out by cold temperatures which moderate short-duration convective activity, especially for $W\_Tun$ (region A1). The topography characterizing the northern pacific coast may then explain the smoothing effect for the curve of region $NW\_Pac$ (A2). In this case, in fact, the precipitation rates at daily and longer scales are enhanced by the orographic effect acting on synoptic weather systems coming from the Pacific Ocean (Wallis et al., 2007)."*

b) Maybe you need to explain beforehand what does it means (physically) when H is small and when H is large (as at page 15, lines 13-15), when H increase and when H decreases with $d_1$.

Please, see our reply to the previous comment. As you mentioned, **Lines 15 to 29, Page 4** already list the results in the literature that reports the physical interpretation of $H$ and gives details about its spatial distribution over several regions. In the initial version of the paper this explanation was part of the introduction of Sect. 5, just before the interpretation of our results. However, they have been moved following Editor's suggestion to improve the separation between the methods, results and discussions. In this way, the practical interpretation of high and low values

of the scaling exponent had been logically linked to its statistical meaning [Eq. (2) and lines below].

In the actual version of the manuscript, Section 5.1 is structured as follow: the section first describes the regional distribution of $H$; then the connection between $H$ spatial patterns to the geographical and climatological characteristics of each region is made; finally a general interpretation of the results fis given at **Lines 2 to 11, Page 16**.

c) Page 14, lines 14-22: good interpretation of the behaviour in Region D.

Thank you.

d) Page 15 lines 2-5: clear, whereas you loose me at lines 6-9.

Thank you for pointing out our lack of clarity. **Lines 24 to 1, Page 15** have been rephrased to:

*"However, $H$ shows a smoother increase in Fig. 7 (f) with respect to Fig. 7(e). This may indicate that in eastern areas [region F] sub-daily duration extremes are more likely associated to embedded convective and stratiform systems or to mesoscale convective systems, which are less active in western dry areas of region E (Kunkel et al., 2012). On the contrary, differences between short- and long-duration extreme precipitation intensity seem stronger for south-western dry regions [Fig. 8 (e)], where less intense summer extremes are expected compared to eastern areas [see supplementary material, Fig. S1]. In particular, $H$ tended to scatter in a range of higher values for approximately 1 h $\leq d_1 \leq$ 12 h indicating that precipitation intensity moments strongly decrease as the duration increases."*

e) page 15, lines 10-11: I disagree with this sentence, I have not seen any evidence that the material illustrated here supports these results (you have not proven this).

In Section 5.1 we showed that different geographical regions generally displayed distinct distribution of H for different scaling intervals [panels (a)-(f) of Fig. 8 show different curves]. These regions are characterized by different climates and precipitation regimes [please, see also our reply to comment 15 a) for a discussion of the importance of choosing geo-climatological regions for this analysis]. The suggestion we made at **Line 2, Page 16** is thus that one can suppose a link (even qualitative) between the difference observed for $H$ values [e.g, for the black lines in Fig. 8 representing the median of $H$ in each region] and the differences between the specific climates and weather regimes characterizing each region. We think that the results presented in the paper qualitatively support the hypothesis that there exist a link between scaling and climatology even if it is not proven in a statistical or mathematical way. That was we use the term "suggest". However, to prevent any confusion we rephrased this sentence as:

*"In summary, these results suggest that both local geographical characteristics, such as topography or coastal effects, and general circulation patterns may influence precipitation scaling at a regional scale."*

17. **Section 6:**

a) From page 15 line 30 to page 16 line 7: since you are assessing a distribution, why don't you use a KS statistics, rather than inventing an engeneered metric which compares the quantiles? Recall Equation 11 for clarity, please.

Thank you for the suggestion but we preferred to limit the used of the GOF test (AD and KS) to the non-parametric estimation of SS models (Section 4) for several reasons. Primarily, these tests have a low statistical power so it is always preferable to limit their use to situations in which no alternative exists; this was the case in Section 4. Secondly, the use of a discrepancy measure such as RMSE allows to numerically evaluate the mean errors on quantile

estimates; conversely, GOF tests can only state the statistical significance of the results according to a specified significance level. For this reason, in Sect. 4 we also estimated the Normalized RMSE after the evaluation of SS validity via the use of the KS and AD tests. Finally, the use of GOF tests is not strictly necessary to validate the use of the SS-GEV models considering that i) the use of the SS hypothesis for approximating the AMS distributions has been legitimated by results in Section 4 and that ii) the GEV distribution is the only natural distribution to use for Annual Maxima Series. Hence, no direct logical alternative exists.

As you suggested, the following reference to Eq. (11) has been added at **Line 4, Page 17**:

*"See Eq. 10 for the definition of $\epsilon_{d,mod}$ for each station."*

b) Page 16, line 13: for the ID and LD datasets, the behaviour of the SS parameters versus non-SS parameters is opposite, they cannot be both more right skewed for the SS estimation.

Thank you for pointing out this error. We agree with the reviewer that, while for the ID dataset "both $\mu_*$ and $\sigma_*$ distributions were more positively skewed than the corresponding non-SS distributions", this is not true for LD. Hence, we corrected the sentence to [**Lines 9 to 13, Page 17**]:

*"Similarly, for 6 h $\leq d_1 \leq$ 2 days in the LD dataset, the SS location and scale parameter distributions are in relatively close agreement with the corresponding non-SS parameter distributions. Conversely, in the ID dataset, both $\mu_*$ and $\sigma_*$ distributions are more positively skewed than the corresponding non-SS distributions. Finally, for $d_1 \geq$ 2 days in the LD dataset, $\mu_*$ and $\sigma_*$ had distributions shifted toward lower values than $\mu_{24h}$ and $\sigma_{24h}$."*

c) Page 16, line 16: it seems to me that the discerepancies between SS and non-SS parameters for the LD dataset and long durations (the $\Delta\mu$ and $\Delta\sigma$ in the supporting material) and quite big.

Thank you for pointing out our mistake: the colorbar label was expressed in percent scale, while $\Delta\mu$ and $\Delta\sigma$ where defined in relative scale. The error has been corrected by changing the colorbar labels of Fig. S12 and S13 to, respectively, $\Delta\mu\%$ and $\Delta\sigma\%$.

Once these corrections made, one can observed that the relative biases in $\mu$ and $\sigma$ estimates were small in most cases since the *"median values of $\Delta_\mu$ and $\Delta_\sigma$ were generally smaller than $\pm5\%$ and $\pm10\%$"* [as stated at **Line 15, Page 17**].

d) Page 16, lines 14-21: there is something strange about your results: you state that the estimation of the scale parameter has a small uncertainty when the shape parameter is correctly estimated. However, from Figure 8. I can see that the shape parameter is not correctly estimated (red and black curves do not match). What is the implication of the mismatch of the shape parameter for the non-SS and SS estimation on your results? The shape parameter is usually set to zero because it is all over the places, and often not-significantly different from zero (as you find in your Figure 9c: the estimated shape parameter is quite noisy)!!!

Figure 9 presents the distributions of SS and non-SS GEV parameters across all stations (these are not at-site differences). For the shape parameter (Fig. 9, $3^{rd}$ col.), one has to remind (as specify in the caption) that the case $\xi = 0$ was not considered [see also our reply to comment 17 h]. Hence, the fact that the red and black curves don't match simply indicate that the shape parameters estimated using the SS hypothesis are different than the non-SS estimates when the shape parameter is significantly different from zero.

We argue that the SS model allows a better assessment of the the shape parameter values than the non-SS model.

In our application, in fact, the majority of stations have non-SS shape parameters $\xi_d$ which are non-significantly different from zero while these fractions are substantially reduced under the SS model [see Fig. 11]. In this regard, note that many authors have shown that the fact of setting $\xi = 0$ *"may lead to important underestimations of the extreme quantiles quantiles (e.g., Koutsoyiannis, 2004a, b; Overeem et al., 2008; Papalexiou et al., 2013; Papalexiou and Koutsoyiannis, 2013)"* [see **Line 17, Page 5**]. Moreover, we observed more evidence of heavy tailed AMS distributions for SS GEV models ($\xi_* > 0$) with values of $\xi_*$ mostly in the $(0.10, 0.25]$ interval [please, see **Line 8, Page 18**, Fig. 11, and Fig. 9, 3rd col.]. These results are consistent with previous studies, which reported typical values of $\xi \approx 0.15$ for cases in which AMS are long enough to reduce estimation uncertainty and provide non-zero estimates of the GEV shape parameter (e.g., Koutsoyiannis, 2004b). [see **Line 9, Page 18**]. Hence, the shape parameter $\xi_*$ seemed to be better assessed by SS GEV models, since under the SS hypothesis, the statistical information from several durations can be pooled.

Concerning the scale parameter estimation, we observed that the estimation of the scale parameter, $\sigma$, may be biased when the $\xi$ is spuriously set to zero [**Line 17, Page 17**], since the $\sigma$ may tends in this case to increase to fit the distribution to the more extreme events. Hence, large uncertainties in $\xi$ could imply large uncertainties and biases in $\sigma$.

In other words, our interpretation of Fig. 9 [Fig. 8 in the previous version] is that SS-hypothesis provides a framework under which the GEV parameters (especially the shape parameter) are more accurately estimated than the non-SS estimates, even if we recognize that strong uncertainties still affect the $\xi_*$ estimation [**Line 14, Page 18**]. Furthermore all our results are consistent with what has been previously reported in the literature.

To better stress these points, several changes have been made in Section 6.1 [**Page** 17]:

–  We moved the sentence *"the scale parameter $\sigma_d$ may be biased when the shape parameter is spuriously set to zero ($\xi_d = 0$)"* from the end of the paragraph to **Line 17, Page 17**.

–  **Lines 25 to 1, Page 17** has been rewritten as:

> *"Notable differences between SS GEV and non-SS GEV estimates were observed for the shape parameter [Fig. 9, third col., and Fig. 11] Firstly, for cases having shape parameters strictly different from zero [third column of Fig. 9], $\xi_*$ absolute values were smaller than non-SS $\xi_d$ absolute values. Secondly, the distributions of $\xi_*$ across stations were generally more peaked around their median value than the corresponding non-SS distributions. Finally, for the non SS model, the majority of stations had shape parameter $\xi_d$ non-significantly different from zero, while the fraction of SS GEV shape parameters $\xi_* \neq 0$ was always greater than 39% [asymptotic test for PWM GEV estimators applied at level 0.05; Hosking et al., 1985]. "*

e) Page 16, lines 22-23: I agree with this sentence, nice spatial coherence.

f) Page 16, lines 25-26: why SS shape parameter is nearer to zero and exhibit less spread? Is this an effect of the assumptions of SS?

Yes, as stated at **Line 7, Page 18**, the fact that $\xi_*$ show less variability than $\xi_d$ is one of the results which suggest that *"pooling data from several durations may effectively reduce the sampling effects impacting the estimation of $\xi$, allowing more evidence of non-zero shape parameters, and, in many cases, of heavy tailed ($\xi > 0$) AMS distributions."*. In other words, the improvement in GEV estimation induced by the SS models allows to 1) decrease the number of $\xi = 0$ cases, and, 2) reduce the dispersion of shape parameters among the station which manifest itself into shape parameter values closer to zero.

g) Page 16, line 27: very difficult sentence to read (essentially the shape parameter is zero).

As reported in reply to comment 16 d), the sentence has been rewritten as [**Line 30, Page 17**]:

*"Finally, for the non SS model the majority of stations had shape parameter $\xi_d$ non-significantly different from zero, while the fraction of SS GEV shape parameters $\xi_* \neq 0$ was always greater than 39% [asymptotic test for PWM GEV estimators applied at level 0.05; Hosking et al., 1985]."*

h) Page 16, lines 27-31: these results are not intuitive. Figure 10 suggests that the shape parameter for the non-SS estimates is predominantely zero, whereas for the SS estimates there is a large proportions of positive and negative shape parameters. However, the right column of Figure 8 shows exactly the opposite (SS estimate are nearer to zero than non-SS estimates). I assume the difference is due to the very different width of confidence interval associated to the estimate of the shape parameter, for non- SS and SS samples. Are the sample sizes very different (I believe so ... given that x* in equation 8 is obtained from a very large sample). Then maybe these results are artificial ... (as you conclude afterwards, at page 17, lines 3-5: but this link is not explicit)!!!

The reviewer must remind that Fig. 9, $3^{rd}$ col, excludes cases for which $\xi = 0$ (Gumbel distribution) [see the caption of Fig. 9]. Therefore, Figures 9 and 11 are not inconsistent since a larger fraction of stations have significant non-zero shape parameters for the SS-model. The reviewer might be right when he mentioned that the difference between shape parameter distributions is due to the very different width of confidence interval associated to the estimate for the SS- and non-SS models. Sample sizes are indeed very different since the SS model pools the AMS from all the durations included in each scaling interval. These results are not "artificial" and reflect the fact that, under SS model, the sampling error is reduced. Corrections made due to previous comments should have clarified these points [for instance, see also our reply to comment 16 d)].

d) Page 17, lines 7: eliminate (not clear what this refer too).

To make the text clearer, we modify the sentence to [**Line 11, Page 18**]:

*"These studies typically reported values of $\xi \approx 0.15$ (e.g., Koutsoyiannis, 2004b), which are close to $\xi_*$ values estimated in the present analysis for cases with $\xi_* > 0$. "*

e) Overall, the text starting at page 16 line 27 and ending at page 17 line 10 should be all reorganized ina more coherent single paragraph.

We agree with the reviewer and corrections made in response to previous comments should have improved the text between **Line 30, Page 17** and **Line 15, Page 18**.

f) Page 17, lines 12-15: Figure S13 shows that for shape parameter equal to 0 (the majority of the stations) the error of quantiles estimated by the non-SS is smaller than that for SS estimates.

Yes, as reported at **Line 3, Page 18**, for many stations (55% to 60% for 6-duration scaling intervals) the SS shape parameter is not-significantly different from zero ($\xi_* = 0$). In these cases, the SS GEV model allows a reduction of the mean error on quantiles only for a small proportion of stations (generally lower than 0.4), as showed in Fig. S14 [Fig. S13 in the previous submission]. Conversely, for cases with non-zero $\xi_* = 0$, the fraction of stations with decreasing errors was higher than 60% for most of the scaling intervals and durations.

To better describe these results **Lines 18 to 23, Page 18** have been modified to:

*"For cases with non-zero $\xi_*$, more than $60\%$ of stations had $\epsilon_{d,ss} < \epsilon_{d,non-ss}$ over most scaling intervals and durations. The 6-duration scaling intervals "15 min - 1 h 30 min" (SD dataset) and "1 h - 6 sih" (ID dataset) showed the largest fractions of stations with increasing errors. On the contrary, increasing errors ($\epsilon_{d,ss} > \epsilon_{d,non-ss}$) were observed for all scaling intervals and durations for most stations (generally more than $70\%$) having $\xi_* = 0$. "*

**Authors' response to $2^{nd}$ referee's comments**

I checked the article and author's responses. I think the authors did a good job to account for both reviews. A few technical points need to be addressed but this is minor.

5 **Minor comments:**

1. p 3 l 23 : the fact that $X_d^q$ and $\lambda^{Hq} X_{\lambda d}^q$ have the same distribution comes directly from (2). You don't need for that to have finite moments (please note by the way that exponent 'q' is missing in lambda). However (3) needs finite moments.

   Corrected, thank you. The paragraph now reads [**Lines 26 to 29, Page 3**]:

10 *"An important consequence of the SS assumption is that $X_d$ and $\lambda^H X_{\lambda d}$ have the same distribution. Hence, if $X_d$ and $X_{\lambda d}$ have finite moments of order q, $E[X_d^q]$ and $E[X_{\lambda d}^q]$, these moments are thus linked by the following relationship ... "*

2. p7 l 14 : I don't think the relation $x_{d_2} \leq x_{d_1}$ for $d1 < d2$ is always valid. For example, let consider the hourly series with values 10-2-10 mm/h. Then the maximum 2h-intensity is 12/2=6mm/h, while the maximum 3h-intensity is 22/3>6
15 mm/h. So $x_{d_2} > x_{d_1}$ for $d_2$=3h and $d_1$=2h. Also $x_{d_2}/x_{d_1} < d_1/d_2$.

   We apologize for the error made in the sentence at **Line 20, Page 7** and we agree that relationship $x_{d_2} \leq x_{d_1}$ is not always valid. The numerical algorithm used for data screening and AMS construction considered the following criterion for *precipitation depth [mm]*:

   $$\frac{x_{d_2}}{x_{d_1}} \geq \frac{d_2}{d_1}$$

20 When the rainfall depth is used, in fact, the relationship must be respected for each couple of durations $d_1 < d_2$ (for your example we have: $x_{d_1} = 12 < 22 = x_{d_2}$ and $\frac{x_{d_2}}{x_{d_1}} = \frac{22}{11} \geq \frac{3}{2} \frac{d_2}{d_1}$). The confusion arose when converting this relationship in term of rainfall intensity. The error has been corrected by changing **Line 20, Page 7** to:

   *"For instance, each pair of DMPD rainfall intensities [mm/h] $(x_{d_1}, x_{d_2})$ observed at durations $d_1 < d_2$ must respect the condition $x_{d_2}/x_{d_1} \geq d_1/d_2$ derived from the definitions of daily maximummaxima rainfall intensity and depth; "*

3. p 7 l 29 : IS $\rightarrow$ ID ?

   Corrected.

4. p 7 l 30: the first matrix on the left of Fig. 1(a) $\rightarrow$ the top left matrix of Fig. 1(a)

30 Corrected, thank you.

5. p 8 : Does the SS sample $x_{d,ss}$ comprises the non-SS sample $x_d$ ? I guess it should not (for independence testing) but it is not clear to me on (8)

We agree that for GOF tests applied in Sect 4 it would be worth considering SS samples $x_{d,ss}$ that do not contain data from duration $d$. For this reason, the steps 1-3 of the methodology described at **Page 9** has been repeated in a cross validation settings [see also the replies to comments 11 e), 11 f), and 12 c) of the $1^{st}$ referee].

However, in the mathematical definition of SS sample, $x_{d,ss}$, [**Line 15, Page 9**] one should include the observations of the rescaled AMS for duration $d$ (i.e., observations from sample sample $x_d$). This sample, in fact, is not only used for the SS validation presented in Section 4, but also for other analyses which require to include the rescaled $x_d$ observations in $x_{d,ss}$. For instance, for the estimation of the SS GEV models [Sect. 6, **Lines 24 to 26, Page 16**] $x_{d,ss}$ should contain observations from AMS observed at $d$ and rescaled at $d^*$.

6. p 15 l 25 : obtained 12 $\rightarrow$ obtained for 12

Corrected.

**Authors' response to 3$^{rd}$ referee's comments**

Thank you for the opportunity to review this manuscript. I agree with the two reviewers on that the first two sections of the paper are very well written and the rest of the sections need improvement in order to clearly deliver your messages. Although still a little bit difficult to understand (I had to read it a few times. But it could be due to my lack of background.), I can see that the manuscript has improved significantly by addressing the comments from previous reviewers. I recommend the manuscript to be accepted with minor revisions.

1. There are a lot of symbols and abbreviations in this manuscript. Maybe the authors can redefine them when first used in each major section (as well as in figure captions) to help readers to understand the methods and results better. Or adding a glossary table in the supplementary documents may help.

   Following the reviewer suggestion we added Table S1 to the supplementary material; this table lists the relevant and recurrent acronyms. A reference to Table S1 has then been added in the introduction [**Line 17, Page 3**]. Then, additional definitions of some acronyms have also been added to the text [see, for instance, acronym added at **Line 21, Page 3**, AMS definition added at **Line 13, Page 6**, section title of Sect. 6, and reference to Figure 1 a) added at **Line 9, Page 7**].

2. What is the colour scheme in Figure 1? Can you please add a legend of the colour scheme?

   We apologize for the technical problem, the color bar has been added to Fig. 2 [corresponding to Fig. 1 in the previous version of the manuscript].

3. To improve readability, this reviewer recommends the authors to break some long sentences into shorter ones. For example, the sentence starting on Line 11 on Page 12 includes at least three messages, which should be broken down into shorter sentences. There are a number of similar sentences in the results and discussion sections.

   We agree with the reviewer and we rephrase various paragraphs of the manuscript in order to increase the readability. See, for instance, the following paragraphs :

   – Sect. 4: **Line 27, Page 10**, **Line 7, Page 8**, and **Line 34, Page 12** [cited by the reviewer];
   – Sect. 5: **Line 24, Page 15**, **Line 1, Page 14**, **Line 14, Page 14**, and **Line 7, Page 16**;
   – Sect. 6: **Line 18, Page 16**, **Line 22, Page 17**, and **Line 30, Page 17**;
   – Conclusion: **Line 8, Page 20**

4. Section 7 on discussion and conclusion seems to be a summary of what you have written so far. Can you please add some discussion on why this research is important? How can the outcomes be used in practical sense? For example, do they imply any changes to current flood risk estimation/infrastructure design guidelines? What are the limitations of this study and recommended future research?

   Several part of Sect. 7 introduce the mentioned issues. In particular recommended extensions of the research or study limitations were briefly discussed at **Line 6, Page 20**, **Line 29, Page 20**, and **Line 11, Page 20**. However, to make more explicit references to practical implications and limitations of our study, the following modifications have been made:

   – We modified **Lines 6 to 8, Page 20** to :

[revised manuscript text omitted]

---

## Author Response (AR3)

Dear Editor,

Please find enclosed the third revision of the manuscript "Simple Scaling of Extreme Precipitation in North America" by Innocenti et al. to be considered for publication in HESS.

According to the referee's comment, we added a discussion of recent results concerning the evidences of extreme rainfall modification with climate change and the implications of these results in terms of the temporal scaling presented in our paper

**5 modification with climate change an [Lines 23 to 4, Page 18].**

Following HESS editorial instructions, we provide a detailed response to the referee's comment (which is reported in blue in the following text). Line and page numbering (**in bold**) refers to the revised manuscript in "track changes" mode attached to this reply.

10 Sincerely,

Silvia Innocenti, on behalf of the co-authors.

**Referee's comments**

This paper has been reviewed in the past and I am coming new to the review process here. Hence I will try to keep my comments constructive and general rather than heavy on details, as I feel other reviewers have addressed those well.

- 5 My main problem with the paper is assumption of SS in a warming climate. There have been too many papers that show (a) that rainfall extremes are scaling with rising temperature [Westra, S., L. V. Alexander, and F. W. Zwiers (2013), Global increasing trends in annual maximum daily precipitation, Journal of Climate, 26(11), 3904-3918.], and (b) that the scaling is greater with short duration rainfall than long duration rainfall [Hardwick-Jones, R., S. Westra, and A. Sharma (2010), Observed relation-ships between extreme sub-daily precipitation, surface temperature, and relative humidity, Geophysical Research Letters, 37, doi:10.1020/2010/CL0450211. Circum this are needed to experime the there for a sub-daily precipitation of the sub-daily precipitation.
- 10 doi:10.1029/2010GL045081.]. Given this, one needs to question whether the SS assumption the authors use is justifiable or not.

However, there is considerable uncertainty in the conclusions drawn in the above papers, which allows for this paper to make a contribution to the overall discussion. To add to the discussion already there in the paper, I suggest the authors add a section discussing this issue, and the need for further developing of SS models that possibly consider temperature as a covariate in

15 some form or the other. If there is no such variation, one is implicitly assuming stationary co-variability of rainfall with temperature irrespective of rainfall duration, which I believe does not make sense to do anymore.

I also suggest the authors look at [doi:10.1016/j.jhydrol.2016.12.002.] which makes a similar case, and actually presents an approach for generating rainfall sequences such that derived design intensities do observe some type of scaling with temperature differently at different durations. I believe, though, this is an area of evolving research and a lot more needs to be done - but a discussion of all that is happening and what more holes need to be filled will help other researchers who follow in this space.

I recommend publication - but with a good discussion of these issues included in.

**Authors' response to referee's comments**

20

We definitely agree with the reviewer that well known theoretical and empirical evidences support the ongoing changes in precipitation in a warmer climate [*Trenberth et al. (2003), Westra and al. (2014)*]. In particular, the literature has extensively shown evidences of the extreme rainfall intensification as a result of climate change, with studies particularly analyzing i) changes in the intensity, frequency, and spatial patterns of extreme rainfall at global, regional, and local scales [e.g., *Trenberth (2005), Alexander et al. (2006), Westra et al. (2013), Hartmann et al. (2013), Donat et al. (2016)*], and ii) the relationship between increases in daily and sub-daily rainfall extremes with increasing temperature and changes in humidity conditions
[e.g., *Trenberth et al. (2003), Westra and al. (2014), Panthou et al. (2014), Barbero et al. (2017), Wasko and Sharma (2017)*].

As suggested by the reviewer, two common major conclusions resulted from these studies. Firstly, with global warming, rainfall extremes are expected to change at a rate equal to or higher than the moisture holding capacity of the atmosphere. The use of the Clausius-Clapeyron relationship [*CC scaling*] leads to an expected global sensitivity of extreme rainfall equal to  $\sim 7\%$  per °C; however the actual scaling between extreme rainfall and temperature (or dew point temperature) estimated in empirical studies was generally higher that the theoretical CC value and greatly varied in space and with the chosen rainfall exceedance

35 studies was generally higher that the theoretical CC value and greatly varied in space and with the chosen rainfall exceedance probability [e.g., *Westra and al. (2014)* and references therein]. Secondly, higher temperatures are expected to induce greater CC scaling rates for short duration extreme rainfall [e.g., up to a few hours] than those at daily or longer time scales [e.g., *Lenderink and Attema (2015), Barbero et al. (2017), Wasko and Sharma (2017)*].

As a result, the SS exponent, *H*, describing the link between short and long duration AMS distributions, may effectively change
in a changing climate and the hypothesis of a stationary temporal scaling used in our analysis does not hold over long time horizons. As suggested by the reviewer, one possible way to incorporate this non-stationarity in temporal scaling models could be to allow SS and GEV parameters to vary in time as a function of temperature, rather than assuming a constant value for *H* and

the same AMS probability distribution over the whole observational period. A comparable approach has been used in *Wasko* and Sharma (2017) for conditioning precipitation distribution parameters on monthly temperature when using a Neyman-Scott rectangular pulses (RPLS) process for the simulation of continuous rainfall series.

Although interesting, however, the possibility of applying this approach for our study remains difficult. In particular, the esti-

- 5 mation of temporal trends for the SS scaling-temperature relationships would be challenging considering the relatively short historical records available in our analysis [see Barbero et al. (2017) for a discussion on the estimation of trends for daily and sub-daily precipitation extremes in North America]. Moreover, a comprehensive separate study would be needed to investigate the complex relationship between H and the increasing temperature in order to correctly identify the suitable covariate(s) for a non-stationary SS model.
- 10 Notwithstanding, considering the importance of the issue raised by the reviewer, we added the following paragraph at Lines 13 to 17, Page 3:

"Note that, although modifications in precipitation distributions are expected as a result of climate changes [e.g., Trenberth (2003), Hartmann et al. (2013), Westra et al. (2014)], the proposed approach implicitly relies on the assumption of stationarity for extreme rainfall. This choice has been motivated by both the limited evidence for changes in rainfall intensities for North

15 America extremes during last decades, and the difficulties of assessing distribution changes from short recorded series, especially for sub-daily extremes [Barbero et al. (2017) and references therein]."

Moreover Lines 23 to 4, Page 18 [Sect. 7 Discussion and conclusion] have been modified to:

"Moreover, two important limitations of the presented SS approach must be stressed. Firstly, a more comprehensive assessment of the scaling exponent uncertainty and of the influence of dataset characteristics on the estimation of AMS simple scaling is

- 20 recommended for a reliable estimation of SS IDF curves. Secondly, the proposed model relies on the implicit hypothesis of stationarity of AMS over the observed period while growing evidence supports the ongoing changes in extreme precipitation intensity, frequency, duration, and spatial patterns as a result of climate change [e.g., Hartmann et al. (2013), Westra et al. (2014), Donat et al. (2016)]. In particular, short duration extreme rainfall is expected to respond to global warming with a different sensitivity to temperature than those expected at daily or longer time scales [e.g., Westra et al. (2014), Lenderink and
- 25 Attema (2015), Wasko and Sharma (2017), Barbero et al. (2017)] which implies a change in the temporal scaling properties of precipitation over time.

Hence, considering these limitations and our general results, any future extension of this study should investigate the possibility of introducing spatial information in scaling models as well as the characterization of possible evolution of the scaling exponent in a warmer climate in order to identifying valuable approaches allowing non-stationarity of SS model parameters."

30

35

- Alexander, L.V., Zhang, X., Peterson, T.C., Caesar, J., Gleason, B., Klein Tank, A.M.G., Haylock, M., Collins, D., Trewin, B., Rahimzadeh, F. and Tagipour, A., 2006. Global observed changes in daily climate extremes of temperature and precipitation. Journal of Geophysical Research: Atmospheres, 111(D5).
- Barbero, R., Fowler, H.J., Lenderink, G. and Blenkinsop, S., 2017. Is the intensification of precipitation extremes with global warming better detected at hourly than daily resolutions?. Geophysical Research Letters, 44(2), pp.974-983.
  - Berg, P., Moseley, C. and Haerter, J.O., 2013. Strong increase in convective precipitation in response to higher temperatures. Nature Geoscience, 6(3), p.181.

- Donat, M.G., Alexander, L.V., Herold, N. and Dittus, A.J., 2016. Temperature and precipitation extremes in century-long gridded observations, reanalyses, and atmospheric model simulations. Journal of Geophysical Research: Atmospheres, 121(19).

- 40 Hartmann, D.L., Tank, A.M.K., Rusticucci, M., Alexander, L.V., Brönnimann, S., Charabi, Y.A.R., Dentener, F.J., Dlugokencky, E.J., Easterling, D.R., Kaplan, A. and Soden, B.J., 2013. Observations: atmosphere and surface. In Climate Change 2013 the Physical Science Basis: Working Group I Contribution to the Fifth Assessment Report of the Intergovernmental Panel on Climate Change. Cambridge University Press.
  - Lenderink, G. and Attema, J., 2015. A simple scaling approach to produce climate scenarios of local precipitation extremes for the Netherlands. Environmental Research Letters, 10(8), p.085001.

- Panthou, G., Mailhot, A., Laurence, E. and Talbot, G., 2014. Relationship between surface temperature and extreme rainfalls: A multi-time-scale and event-based analysis. Journal of Hydrometeorology, 15(5), pp.1999-2011.
- Trenberth, K.E., 2005. The impact of climate change and variability on heavy precipitation, floods, and droughts. Encyclopedia of hydrological sciences.

[revised manuscript text omitted]